



# A coupled pelagic-benthic-sympagic biogeochemical model for the Bering Sea: documentation and validation of the BESTNPZ model (v2019.08.23) within a high-resolution regional ocean model

Kelly Kearney[1,2], Albert Hermann[1,3], Wei Cheng[1,3], Ivonne Ortiz[1,2], and Kerim Aydin[2]

[1]University of Washington, Joint Institute for the Study of the Atmosphere and Oceans (JISAO), Seattle, WA
[2]NOAA Alaska Fisheries Science Center, Seattle, WA
[3]NOAA Pacific Marine Environmental Laboratory, Seattle, WA

**Correspondence:** Kelly Kearney (kelly.kearney@noaa.gov)

**Abstract.** The Bering Sea is a highly productive ecosystem, supporting a variety of fish, seabird, and marine mammal populations as well as large commercial fisheries. Due to its unique shelf geometry and the presence of seasonal sea ice, the processes controlling productivity in the Bering Sea ecosystem span the pelagic water column, the benthic sea floor, and the sympagic sea ice environments. The BESTNPZ model has been developed to simulate the lower trophic level processes throughout this region. Here, we present a version of this lower trophic level model coupled to a three-dimensional regional ocean model for the Bering Sea. We quantify the model's ability to reproduce key physical features of biological importance as well as its skill in capturing the seasonal and interannual variations in primary and secondary productivity. We find that the ocean model demonstrates considerable skill in replicating observed horizontal and vertical patterns of water movement, mixing, and stratification, as well as the temperature and salinity signatures of various water masses throughout the Bering Sea. It is also able to capture the mean seasonal cycle of primary production observed on the data-rich eastern middle shelf. However, its ability to replicate domain-wide patterns in nutrient cycling, primary production, and zooplankton community composition, particularly with respect to the interannual variations that are important in a fisheries management context, remains limited.

## 1 Introduction

The Bering Sea is a highly productive ecosystem. Its broad, shallow eastern shelf reaches widths of over 500 km, with an average depth of only 70 m, leading to a long growing season and high annual primary production (Rho and Whitledge, 2007). Tidal mixing along the continental shelf break also leads to a highly productive off-shelf region, often referred to as the "green belt", where the confluence of nitrate from the deep basin and iron from the shelf are mixed into the euphotic zone (Springer et al., 1996). This high primary productivity across the shelf and slope in turn supports a wide variety of pelagic and benthic predators, which support fisheries that land nearly half of the U.S. annual catch (National Marine Fisheries Service, 2017; Fissel et al., 2017).

Because of the Bering Sea's economic and cultural importance, changes in its ecosystem have prompted a series of contemporary research efforts, including the Bering Ecosystem Study (BEST) and Bering Sea Integrated Ecosystem Research Program





(BSIERP), that aimed to advance understanding of ecosystem processes and their relationship to the physical environment in the Bering Sea (Sigler et al., 2010). As part of these efforts, the Bering Ecosystem Study Nutrient-Phytoplankton-Zooplankton (BESTNPZ) biogeochemical model (Gibson and Spitz, 2011) was developed to simulate the key processes and features of the Bering Sea lower trophic level ecosystem, including primary and secondary production in the pelagic environment as well as
benthic-pelagic and ice-pelagic interactions.

A regional ocean model that includes the BESTNPZ biological model has been used to investigate a variety of topics, including historical and future biophysical variability (Hermann et al., 2013, 2016, 2019), ecosystem status and variability (Ortiz et al., 2016), fish advection and recruitment (Wilderbuer et al., 2016), and community connectivity within crab populations (Richar et al., 2015); at least a dozen ongoing projects continue to rely on this model for retrospective and future analyses.
Previous studies have examined the model's skill and sensitivity broadly. Hermann et al. (2013) demonstrated that the physical model shows skill in replicating observed patterns in temperature, salinity, and circulation, while Gibson and Spitz (2011) performed a thorough sensitivity analysis of the biogeochemical model in a one-dimensional environment representing the mid-shelf portion of the southeast Bering Sea shelf. However, a comprehensive evaluation of the biogeochemical skill of the BESTNPZ model in the three-dimensional ocean modeling context has been lacking.

Here, we present a thorough documentation of the BESTNPZ biogeochemical model in its current state. We also provide context and history for the various versions of the code that were used in the aforementioned publications, and the changes that were made between publications and since. Finally, we evaluate several aspects of lower trophic level output of the BEST-NPZ model within a high-resolution regional ocean model, focusing on whether the model properly captures key biophysical and biogeochemical processes necessary to realistically simulate primary production in the Bering Sea. This skill assessment
reveals the model's strengths and weaknesses in reproducing historical patterns across the entire Bering Sea domain, and also serves as a baseline to which further model improvements can be compared.

## 2   An overview of the BESTNPZ model

### 2.1   Model structure

The biogeochemical and ecosystem model underlying this study uses a traditional nutrient-phytoplankton-zooplankton structure.
ture. It tracks a total of 19 state variables: 14 pelagic variables, 2 benthic variables, and 3 sympagic (ice) variables.

The 14 pelagic state variables are resolved as tracer variables within the physical model, and are therefore subject to advection and diffusion. The nutrients include nitrate, ammonium, and iron. Two size classes of phytoplankton (small and large) and five zooplankton groups (microzooplankton, small copepods, large copepods, euphausiids, and jellyfish) comprise the living state variables in the model. Both the large copepods and euphausiids groups are further subdivided into two state vari-
ables each, with parameterizations tailored to on-shelf and off-shelf populations; at present the parameterizations for these two groups are very similar. Two detrital state variables, representing fast- and slow-sinking detritus, are also included.

The benthic submodel includes a living benthos group, encompassing all live infauna, and a single benthic detritus group. These state variables do not include any horizontal or vertical movement.





The three sympagic state variables are associated with seasonal sea ice, and include nitrate, ammonium, and ice algae. These variables are assumed to occupy a thin skeletal layer on the underside of sea ice (when present); their horizontal movement is determined by the movement of the ice in which they reside, and they exchange material with the top layer of the ocean model. The exact thickness of the skeletal ice layer is specified via a user input parameter (see subsection A3 for further details); for

all simulations to date, a value of $0.02\,\mathrm{m}$ was used.

Nitrogen is used as the primary currency throughout the model, with all living and detrital state variables assumed to have a constant stoichiometry of 106 moles carbon per 16 moles nitrogen. While iron is also included as a state variable for primary production limitation purposes, its flux through the ecosystem is not tracked beyond its uptake during primary production, and water column iron is restored to a simple empirical distribution based on water depth on an annual timescale. Due to a quirk

inherited from its predecessor model (Hinckley et al., 2009), many of the model's output variables, including phytoplankton, zooplankton, and detrital biomass variables as well as all flux rate diagnostic variables, are reported in carbon-based units; this conversion uses a constant N:C ratio across all state variables.

For a full description of all state variables, process equations, and input parameters in the BESTNPZ model, please see Appendix A.

In its three-dimensional setup, the BESTNPZ ecosystem model is coupled to an implementation of the Regional Ocean Modeling System (ROMS), a free-surface, primitive equation hydrographic model (Shchepetkin and McWilliams, 2005; Haidvogel et al., 2008). The Bering10K ROMS domain spans the Bering Sea and northern Gulf of Alaska, with 10-km horizontal resolution (Hermann et al., 2013). To date it has been run with either 10 (previous studies) or 30 (this study) terrain-following depth levels (Figure 1); we discuss this increase in vertical resolution in the next section. This horizontal domain is a subsec-

tion of the larger Northeast Pacific (NEP5) domain (Danielson et al., 2011); both domains have been shown to capture the primary physical features important to biological processes, including circulation patterns, temperature, and salinity, as well as the seasonal advance and retreat of sea ice.

## 2.2 History and modifications

The BESTNPZ model has undergone several phases of development over the past several years. The code for the pelagic system

originated from a Gulf of Alaska NPZ model known as GOANPZ (Hinckley et al., 2009). Model parameters and equations were tailored to the Bering Sea ecosystem during the Bering Ecosystem Study and Bering Sea Integrated Ecosystem Research Project, and the benthic and sympagic modules were added to the code during this phase. In the earliest publication of the model, Gibson and Spitz (2011) analyzed its sensitivity to input parameter uncertainty when coupled to a one-dimensional ocean model representing the Bering Sea M2 mooring location (56.87°N, 164.06°W). While this study confirmed that large-

scale properties, such as modeled annual net primary production, were in line with observations from the eastern shelf region of the Bering Sea, it did not present any detailed skill analysis of the biological state variables against observations.

Following the Gibson and Spitz (2011) study, the BESTNPZ model was embedded within a realistic three-dimensional ocean model for the Bering Sea, referred to herein as the Bering10K ROMS domain. In a study focusing on physical and biological modes of variability with the Bering10K+BESTNPZ model output, Hermann et al. (2013) demonstrated that the

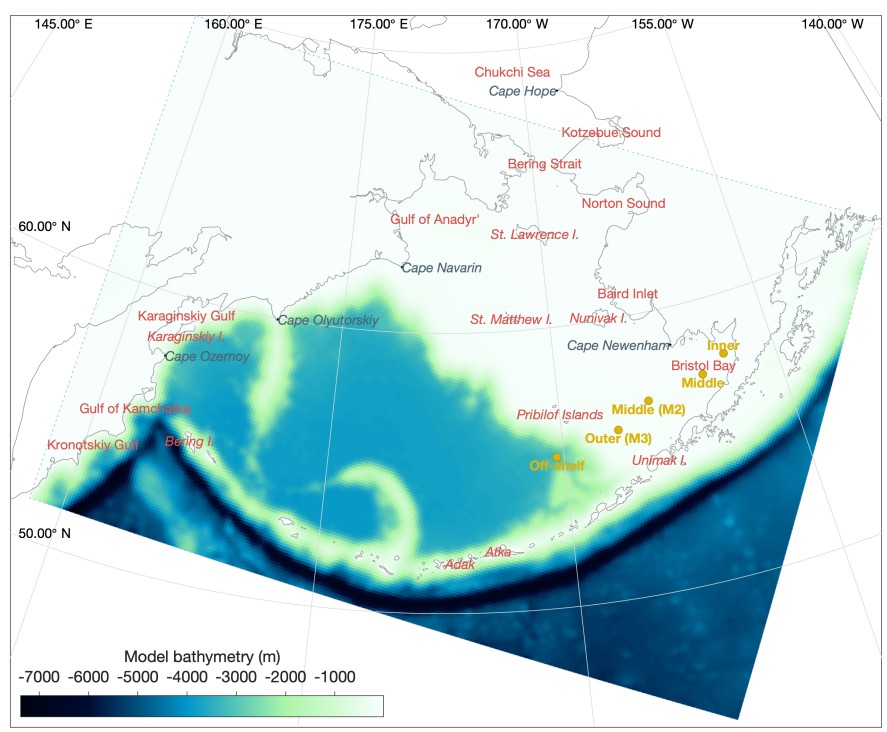

**Figure 1.** The Bering10K ROMS domain, including model bathymetry and a variety of place names within and near the domain.

model physics skillfully replicated observed patterns in temperature and salinity on the Eastern Bering Sea shelf, and that nutrients, phytoplankton, and zooplankton in this shelf region covaried with each other and with physical properties in a manner that supported the existing hypotheses of energy flow in the Eastern Bering Sea. However, Hermann et al. (2013)'s analysis focused on interannual variability, and did not specifically assess the skill of the model to replicate seasonal patterns in primary and secondary productivity, nor did it address the behavior or skill of the model away from the eastern shelf. A few minor input parameter adjustments were made for the Hermann et al. (2013) study versus the earlier Gibson and Spitz (2011) study; apart from this and their differing one-dimensional versus three-dimensional physical environments, the versions of the BESTNPZ model used in these two publications are the same.

A third modification of the BESTNPZ model appears in Hermann et al. (2016), where an updated version of the Bering10K + BESTNPZ model complex seen in Hermann et al. (2013) was used to investigate long-term change in the Bering Sea under various climate change scenarios. In this version of the model, the light attenuation scheme was modified, and the light-related input parameters were adjusted. Several subsequent studies have used and continue to make use of the hindcast and/or climate forecast scenario output that was produced with this version of the Bering10K+BESTNPZ model (Ortiz et al., 2016; Hermann et al., 2019). Pilcher et al. (2019) also used this version of the model, with the addition of carbon variables, to support their analysis of ocean acidification in the Bering Sea.





Following the Hermann et al. (2016) publication, the BESTNPZ source code underwent a thorough revision. The revision process revealed a handful of small but consequential issues in the implementation of the BESTNPZ process equations that have led to a new version of the BESTNPZ model. These issues included the following:

1. Lack of conservation of biomass in large copepod biomass: An error in the code governing vertical migration of large copepods led to a slow but steady post-diapause accumulation of offshore large copepod (NCaO) biomass below their diapausing depth of $400\,\mathrm{m}$. Over the course of a multi-year simulation, this deep biomass manifested itself as a "fall bloom" at depth, and reached levels surpassing that of surface large copepod biomass; this artifact is visible in the depth-integrated biomass results presented in Ortiz et al. (2016). This error has now been corrected.

2. High light limitation: Preliminary validation of simulated phytoplankton production revealed that light was strongly limiting in certain regions of the model domain even under noon summer conditions. In deep water, the 10-layer physical model setup was too coarse to resolve variations in light within the mixed layer, leading to light-limited conditions year-round. In the shallower regions of the inner shelf, an overly-aggressive sediment attenuation term (introduced between the Hermann et al. (2013) and Hermann et al. (2016) versions of the model) also led to year-round light limitation. This high light limitation masked problems with micronutrient limitation in the deep basin and macronutrient limitation in the inner coastal domain. To remove this excessive light limitation, new runs of the Bering10K domain now use 30 vertical layers rather than 10, and a new equation for light attenuation was implemented. See Appendix B for further discussion of these changes to light attenuation.

3. Non-conservative behavior of macronutrients related to nudging: Gibson and Spitz (2011)'s version of the BESTNPZ model implemented empirical relaxation of iron, ammonium, and nitrate throughout its one-dimensional domain; both nitrate and iron were relaxed towards seasonal climatological profiles, while ammonium was relaxed toward zero, all on annual timescales. The relaxation was appropriate in the one-dimensional context, given the use of periodic lateral boundary conditions that did not allow for any advective transport of nutrients into or out of the domain. However, when moved to the three-dimensional Bering10K domain, this nudging becomes inappropriate, as all processes controlling nutrient distribution should conserve biomass (total nitrogen in the system is not constant, due to the open lateral boundary conditions and out-of-system fluxes from burial and benthic denitrification, but these changes to the nitrogen budget can be attributed to known processes and quantified). Particularly in the case of ammonium relaxation, the nudging applied to the three-dimensional domain introduced a phantom process that "scavenged" ammonium from the water column across most of the shelf regions. In the most recent versions of the model, nudging has been removed from the NO3 and NH4 state variables; it remains in place for Fe, since the simplified iron dynamics in the current model do not supply any explicit sources of bioavailable iron.

4. Preferential uptake of nitrate in high-nitrate, high-ammonium conditions: Under the previous governing equations for macronutrient limitation during the gross primary production calculations, the total nitrogen limitation factor (i.e. the factor applied to the maximum photosynthetic uptake rate to account for nutrient limitation, after Lomas and Glibert





(1999)) could exceed 1.0 under high nitrate, high ammonium conditions; the code implemented a cap to counter this by reducing the ammonium limitation factor so the sum of the nitrate and ammonium limitation factors was less than one. This approach led to reduced uptake of ammonium in favor of nitrate when concentrations of both macronutrients were high, and is counter to the assumption that ammonium uptake is usually energetically favored over nitrate uptake, and

5 that high levels of ammonium inhibit nitrate uptake (Glibert et al., 2016, and references therein); it is also counter to the ammonium inhibition encoded in the nitrate limitation factor itself in this model. While this quirk in nutrient uptake in unlikely to have made a large difference in uptake dynamics given that it was only relevant during nutrient-replete conditions, it may have exacerbated the accumulation of ammonium on the shelf seen when nudging was removed. In the most recent version of the code, nitrate and ammonium limitation factors were modified to use an equation (after Frost

and Franzen (1992)) that constrains the total nitrogen limitation factor to a range of 0–1 without the need for additional caps.

5. Euphausiid prey preferences: In previous versions of the BESTNPZ model, the euphausiid groups preyed upon large phytoplankton, ice algae, microzooplankton, and small copepods. However, the modeled euphausiid populations tended to drop precipitously in winter months when these prey were scarce, in contrast to fish diet data that indicate the continued

presence of euphausiids in fish diets on the southeastern shelf during the winter (Livingston et al., 2017). This observation is important to replicate when using the BESTNPZ model to look at the broader flow of energy to higher trophic level species (e.g. Ortiz et al., 2016). In an attempt to increase overwintering success of the on-shelf euphausiid group, feeding links to the two detrital groups and to small phytoplankton were added. The addition of these feeding links for the on-shore euphasiid group distinguishes the tendency of on-shelf Bering Sea euphausiids, dominated by *Thysanoessa raschii*,

to rely on detrital feeding for overwinter survival; in contrast, the shelf-edge population, dominated by *Thyanoessa inermis*, accumulates higher lipid stores to support winter survival (Hunt et al., 2016).

In addition the changes to the biogeochemical model listed above, significant biases in the ice fields produced by earlier versions of Bering10K and related models have been identified and addressed by ROMS colleagues (Durski and Kurapov, 2019, K. Hedstrom and A. Kurapov, personal communications). Late melting of ice was noted by Danielson et al. (2011) and

25 Cheng et al. (2014) for a Northeast Pacific model that utilizes nearly the same ice code as Bering10K, and by Ortiz et al. (2016) and Hermann et al. (2013, 2016) for the Bering10K model itself. Corrections to the longwave radiation terms of ice thermodynamics have been implemented in the latest version of Bering10K code to address the late ice melting bias. Additional improvements to the mechanical ice dynamics by Hedstrom (personal communication) corrected for a previous bias towards excessive ice thickness in some areas. Specifically, this was corrected through the addition of a quadratic ice strength term that

resists ice ridging, based on the work of Overland and Pease (1988).



## 3 Model validation: methods

### 3.1 Model configuration and forcing

We ran two simulations of the Bering10K-BESTNPZ model for this study. For both simulations, the model is driven by surface atmospheric forcing from either the Common Ocean Reference Experiment (CORE) (Large and Yeager, 2009) (1970-1994), the Climate Forecast System Reanalysis (CFSR) (Saha et al., 2010) (1995-March 2011), or the Climate Forecast System Operational Analysis (CFSv2-OA) (April 2011 - Sep 2018); bulk formulae were used to relate winds, air temperature, specific humidity, surface pressure, and shortwave and longwave radiation from these datasets to surface stress, freshwater and heat fluxes (Fairall et al., 1996). Comparison between overlapping years from the CORE and CFSR datasets revealed small differences in values in the radiation variables; the CORE shortwave and longwave radiation values were therefore divided by factors of 0.9 and 0.97, respectively, to align with the CFSR data (note that this is the opposite of the adjustment performed in previous studies, e.g. Hermann et al. (2013, 2016), where the CFSR values were adjusted downward by 10% and 3%, respectively). Lateral boundary conditions for the open southern and eastern boundaries of the model domain use a hybrid nudging/radiation scheme (Marchesiello et al., 2001). During the CORE period these boundary conditions derive from a simulation of the larger Northeast Pacific grid (Danielson et al., 2011), which relied on the Simple Ocean Data Assimilation (SODA) dataset (Carton and Giese, 2008) for its own lateral boundary conditions; the CFS periods use CFS ocean variable values. The northern boundary transport through the Bering Strait is relaxed to the observed value of 0.8 Sv (Woodgate and Aagaard, 2005); earlier sensitivity studies tested whether a seasonally-varying open boundary condition could better replicate the flow patterns in the northern portion of the domain, but the simple relaxation condition was found to perform equally well. Freshwater runoff due to river input was reconstructed from observed river discharge from Alaskan and Russian rivers (Kearney, 2019); river freshwater input is distributed across model grid points near the coast as a surface freshwater flux based on river mouth location, with an e-folding scale of 20 km.

The first model simulation, referred to hereafter as the spinup simulation, looped interannually invariant surface and lateral boundary conditions over a 30-year period. We chose to use boundary condition values from 2001, a year with close to average sea ice cover. Physical variables for this simulation were initialized from the January 1, 2001 values of a previous hindcast of the Bering10K domain. Nitrate was initialized to a constant value of $40\,\mathrm{mmol\,N\,m^{-3}}$ below $300\,\mathrm{m}$, transitioning to $0\,\mathrm{mmol\,N\,m^{-3}}$ at $100\,\mathrm{m}$. Iron was initialized to the same empirical profile used for annual nudging within the model, which sets surface and deep iron values based on bottom depth (see Appendix A). All living biological state variables (i.e. phytoplankton, zooplankton, and benthic infauna) were initialized using a tiny seed value to allow future growth, while all other state variables were initialized at zero. The purpose of this simulation was to allow the model to reach an internally-regulated state, and to verify that any accumulation of nitrogen outside the deep basin was a result of internal dynamics rather than overestimated initial conditions.

The second simulation, referred to as the hindcast simulation, was initialized using values from the final time step of the spinup simulation. The model was then run from 1970–2018 forced with the full time series of surface and lateral boundary conditions from the combined CORE/CFS-derived dataset described above.



## 3.2 Key features of biological importance

To systematically validate the BESTNPZ-Bering10K model complex, we focus on a few key features of the Bering Sea. We begin by looking at physical processes that are known to influence the primary production, and then compare our modeled patterns of primary production, phytoplankton biomass, and phytoplankton and zooplankton community composition to a

variety of measurements. We primarily focus this evaluation of the eastern Bering Sea shelf due to the availability of data in this region, but also qualitatively evaluate the patterns seen in the central basin and the northern shelf regions. While our physical model domain extends into the northern Gulf of Alaska, the biological model was never intended to simulate this region, and for this validation we assume that all regions south of the Aleutian Islands or east of the Alaska Peninsula are outside the biological domain of the BESTNPZ model.

### 3.2.1 Sea ice

Sea ice plays a key role in shaping the ecosystems of the Bering Sea. Ice advances southwestward through the Bering Strait into the Bering Sea, driven both by winds from the northeast and local ice formation, with much of the eastern shelf at least partially covered by ice beginning in early winter (Oct to Nov) through early spring (March to April). Variability in the timing of ice onset and retreat and extent of sea ice can be significant year to year, influenced by winds and air temperature related to

the position and strength of the Aleutian Low and Siberian High pressure systems, as well as ocean conditions (Stabeno et al., 2001).

To analyze the extent and timing of the updated seasonal sea ice, we collected estimates of sea ice concentration derived from Nimbus-7 SMMR and DMSP SSM/I-SSMIS Passive Microwave Data (Cavalieri et al., 1996) for the period of 1980-2018. For comparison with model output, satellite-derived fraction ice cover was interpolated via a nearest neighbor method

to the Bering10K model grid. For both model and observations, we calculate the location of the ice edge along 170°W (the approximate longitudinal center of the seasonal sea ice in the Bering Sea), defining the edge as the southern extent of a continuous block of grid cells where all cells have at least $15\%$ ice cover.

### 3.2.2 The cold pool

As ice advances southward along the Bering Sea shelf, the freezing process and resulting brine rejection leads to the formation

of cold, salty, dense bottom water underneath the ice (Stabeno et al., 2001); continuous freezing in the vicinity of the St. Lawrence polynya further intensifies the formation of this bottom water mass (Danielson et al., 2006). The resulting cold bottom water is referred to as the "cold pool", and is typically defined as waters colder than either $0\,°C$ or $2\,°C$. In the spring, warming of the surface waters coupled with melting of sea ice sharply stratifies the water column over the middle shelf region, isolating the cold pool waters from surface heating and mixing. As a result, this signature water mass can persist well into

the summer months (Stabeno et al., 2001). Due to the isolation of the cold pool, this water can serve as a nutrient reservoir to the mid-shelf region when mixed with nutrient-depleted surface waters during storm events following the initial spring bloom (Sambrotto et al., 1986; Aguilar-Islas et al., 2007). In addition, the location of the cold pool influences the spatial distribution





(Mueter and Litzow, 2008; Stabeno et al., 2012) and recruitment of higher trophic level predators (Mueter et al., 2011; Duffy-Anderson et al., 2016) through various mechanisms.

Measurements of bottom temperature are collected each summer as part of the Bering Sea Groundfish Bottom Trawl Survey. Net trawls are conducted at fixed survey stations located across the Eastern Bering Sea shelf at 20 nautical mile resolution.

From 1982-1989, temperature data was collected via expendable bathythermographs (XBTs). More recent surveys use digital bathythermograph recorders attached to the headrope of the bottom trawl net (BRANCKER RBR XL-200 Micro BTs recorded at 6-second intervals for the 1993-2001 surveys, and a Sea-Bird SBE-39 bathythermograph continuous data recorder at 3-second intervals for 2002-present). Bottom temperature is then averaged over the on-bottom portion of the trawl to produce a single value per station per year (see Buckley et al., 2009; Lauth et al., 2011, for details of data collection and postprocessing).

Here, we use bottom temperature measurements to verify that our model properly captures the characteristics of the Bering Sea cold pool. The cold pool is quantified by indices that represent the fraction of the survey area with bottom water less than $0\,°C$, $1\,°C$, or $2\,°C$ water. In the model, we define the Eastern Bering Sea shelf as all grid cells falling within the Eastern Bering Sea stratified sampling regions $10 - 62$ (Figure 2). We calculate the model cold pool index using two methods. First, we calculate the index value on July 1 of each year; choosing a fixed date allows us to compare a consistent summer snapshot of the cold

pool from year to year. The second index replicates the sampling scheme used in the groundfish survey, choosing bottom temperatures from the closest grid cell and time slice to each observation point; this index allows a better comparison to the observations, given the temporal spread in the observations between the first (typically southeast) and last (northwest) sampled station. For comparison, we also analyze bottom temperature extracted from the Climate Forecast System ocean model; this lower-resolution climate model is coupled to the same atmospheric forcings we use in our hindcast simulation for this time

period, and therefore allows comparison between the original climate model and our dynamically-downscaled representation.

### 3.2.3 Inner, middle, and outer shelf domains

The wide, shallow Eastern Bering Sea shelf is divided into three domains, each characterized by distinctive patterns of stratification and mixing (Coachman, 1986; Kachel et al., 2002; Stabeno et al., 2012). The inner domain stretches from the coast to approximately the $50\,m$ isobath, and is well-mixed year round by both tidal and wind energy. The middle domain reaches

from the $50\,m$ to $100\,m$ isobath; this domain is well-mixed during winter months but thermally stratified during the spring and summer, with a tidally-mixed bottom layer isolated from the surface waters. The outer shelf domain, reaching from the $100\,m$ isobath to the shelf break (approximately the $200\,m$ isobath), more closely reflects the seasonal stratification of an oceanic system, with a tidally-mixed bottom layer that is less sharply separated from the surface layer than in the middle domain (Coachman and Charnell, 1978). Tidal mixing dominates the energy across the entire shelf, with a very small net transport

northward (Coachman, 1986).

We estimate stratification in the model by calculating the potential energy required to mix the water column ($SI$, in units of $J\,m^{-2}$), after Simpson et al. (1977):



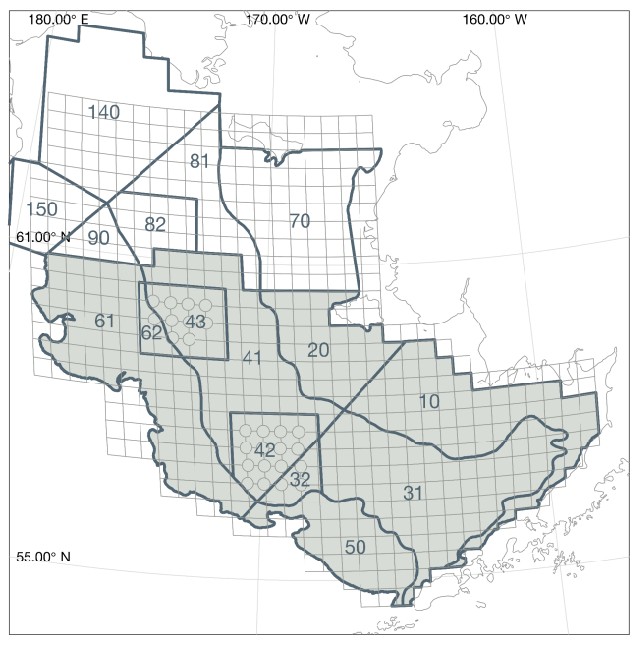

**Figure 2.** Bering Sea groundfish survey stratified sampling polygons. The area considered for calculation of the cold pool index in the model is shaded. The grid cells for each trawl, including bottom temperature measurement samples, are indicated by the light gray lines (circles on the corner of grid cells indicate crab resampling locations; data points from these locations were used in the spatial interpolations shown in Figure 4 but were removed from cold pool index calculations to avoid unequal weighting).

$$SI = \frac{1}{\zeta + h} \int_{-h}^{\zeta} g(\zeta - z)(\rho - \bar{\rho})dz \tag{1}$$

$$\bar{\rho} = \frac{1}{\zeta + h} \int_{-h}^{\zeta} \rho dz \tag{2}$$

where $\rho$ is density, $h$ is the depth of the water column, $\zeta$ is the height of the free surface, $g$ is gravitational acceleration, and $z$ is depth relative to mean surface height (i.e. $\zeta = 0$).

5     We also calculated the location of the structural front separating the well-mixed inner domain from the thermally-stratified middle domain, defining the front as the $0.5\,°\mathrm{C\,m^{-1}}$ contour in maximum vertical temperature gradient (after Schumacher et al., 1979; Kachel et al., 2002). We apply this calculation to years 2000–2010 in the hindcast simulation; this period spans multi-year cold and warm periods, and therefore encompasses a good deal of the variability one might expect to see in this property.





### 3.2.4 Spatial and temporal patterns in primary production

The physical geometry of the Bering Sea, along with the seasonal presence of sea ice, leads to a diverse set of controls on primary production, varying in both space and time. For validation purposes, we focused on a few features that best reflect this complex balance of macronutrient, micronutrient, light, and temperature control of bottom-up processes in this ecosystem.

The highest sustained productivity in the Bering Sea is seen near the edge of the shelf break. This region, referred to as the "Green Belt", coincides spatially with both the shelf break and the Bering Slope Current that carries water northward along the shelf slope (Springer et al., 1996). The shelf-break front and eddies drive high primary productivity both by supplying macronutrients (i.e. nitrate) from the deep basin and micronutrients (i.e. iron) from the shelf and by physically entraining phytoplankton (Okkonen et al., 2004). The Pribilof Islands provide an additional source of dissolved iron to the Green Belt region

(Aguilar-Islas et al., 2007). Variability in mesoscale eddies in the Bering Slope Current are a primary driver of productivity variability in the Green Belt (Okkonen et al., 2004; Mizobata and Saitoh, 2004). The initial spring bloom here is dominated by diatoms but transitions to smaller species in the late summer (Springer et al., 1996).

    Another hotspot of production in this region is the Pribilof Islands. Their location disrupts flow along the 100-m isobath, which leads to enhanced tidal mixing and introduces nutrients from the deep basin to this area, leading to high summer produc-

15 tivity. Productivity can be lower during years when mixing is decreased due to salinity-related frontal structures propagating from the inner shelf (Stabeno et al., 2008).

    While the Bering Sea is generally characterized as being very productive, this production is almost entirely driven by the on-shelf regions and the Green Belt. The deep basin, in contrast, is a high-nutrient, low-chlorophyll (HNLC) region, with low iron levels limiting primary productivity year round (Aguilar-Islas et al., 2007; Suzuki et al., 2002; Leblanc et al., 2005).

On the wide eastern shelf, primary productivity is mainly controlled by the balance of stratification-induced changes in light and nitrogen limitation. An initial spring bloom, dominated by diatoms, rapidly depletes the surface macronutrients along much of the shelf (Sambrotto et al., 1986). In the marginal ice zone, ice edge blooms can occur, accounting for a large fraction of the total spring bloom (Niebauer et al., 1995). This ice edge bloom occurs during years where ice lingers later over the shelf, protecting the underlying water from wind mixing and setting up stronger stratification; earlier-melting ice leads to more wind

mixing and a later spring bloom. Later summer and fall productivity can be driven by wind mixing events that mix nutrient-rich bottom water into the surface layer. Measurements along the middle and outer shelf regions indicate that nitrate drawdown accounts for 30-50% of observed carbon uptake, with the remaining 50-70% driven by ammonium (Sambrotto et al., 1986; Whitledge et al., 1986).

    To check for these patterns in primary production, we performed a visual comparison of modeled phytoplankton biomass

patterns with satellite chlorophyll estimates. The satellite chlorophyll values used were climatological monthly averages from MODIS Aqua OCI-algorithm 4-km resolution product, spanning the period of July 2002 – July 2015 (NASA Ocean Biology Processing Group, 2017). We compared these chlorophyll patterns to optically-weighted, depth-integrated phytoplankton chlorophyll from the model over the same time period, assuming an attenuation length scale of $45\,\mathrm{m}$. While chlorophyll is not a direct measurement of biomass, particularly when derived from satellite color in high-latitude locations, and our model-derived





estimate of satellite-visible chlorophyll is a rough one, it is in this case sufficient to allow large-scale comparison of general spatial patterns in biomass between the various biophysical regions of the Bering Sea (e.g. basin vs shelf, north versus south).

We also looked at chlorophyll data measured via fluorometer at the long-term biophysical mooring at station M2 (56.87°N, 164.05°W) (Stabeno et al., 2012). This mooring provides a more detailed look at both surface and subsurface chlorophyll over 5 several years, including during times of ice cover.

### 3.3 Plankton community composition

To evaluate plankton community composition, we focused on a few patterns of relative biomass seen in the Bering Sea. The spring bloom is typically dominated by diatoms, with small phytoplankton numbers increasing in summer and fall (Springer et al., 1996). Microzooplankon, consisting primarily of protists such as cilliates and dinoflagellates, are the primary grazers 10 on both large and small phytoplankton size classes; measurements of the biomass of microzooplankton vary on order of $10\,\mathrm{mg\,C\,m^{-3}}$ to $100\,\mathrm{mg\,C\,m^{-3}}$ (Olson and Strom, 2002; Howell-Kübler et al., 1996). The mesozooplankton community is dominated by larger zooplankton. Though numerically abundant, the small copepod species typically compose less than 10% of the zooplankton biomass (Vidal and Smith, 1986). Amongst the larger zooplankton groups, the dominant species vary on and off the shelf. The offshore community is dominated by oceanic copepod and euphausiid species, such as *Neocalanus* 15 *spp.* and *Thysanoessa inermis* (NCaO and EupO in the model, respectively), while on the shelf region *Calanus marshallae* and *Thysanoessa raschi* (NCaS and EupS, respectively) compose the majority of the mesozooplankton population. Biomass measurements for these larger mesozooplankton groups are on order of $10\,\mathrm{g\,C\,m^{-2}}$. The offshore and onshore euphausiid groups are distinguished by differing survival strategies. The shelf-edge *T. inermis* build up high lipid stores used for overwinter survival, and spawn early in the spring, without the need to feed on the spring bloom for spawning, while the shelf-dwelling *T.* 20 *raschii* rely more heavily on detrital feeding during the winter and spawn later, after feeding on the spring bloom (Hunt et al., 2016, and references therein).

## 4  Model validation: results

### 4.1  Sea ice

The improved sea ice model in this version of the Bering10K model demonstrates high skill in reproducing the advance of sea 25 ice across the domain, and in capturing the interannual variability of the location of maximum ice extent (Figure 3). Over the entire 1980-2018 time series covered by satellite observations, the ice edge location along 170°W shows a small southerly bias of 0.3° latitude (33.4 km) compared to the location measured via satellite; this is improved from the previous 0.56° (62.3 km) southerly bias seen in the Hermann et al. (2016) version of the ice model. However, despite improvements compared to previous versions of the model, ice retreat still lags observations by approximately two weeks in the early spring.



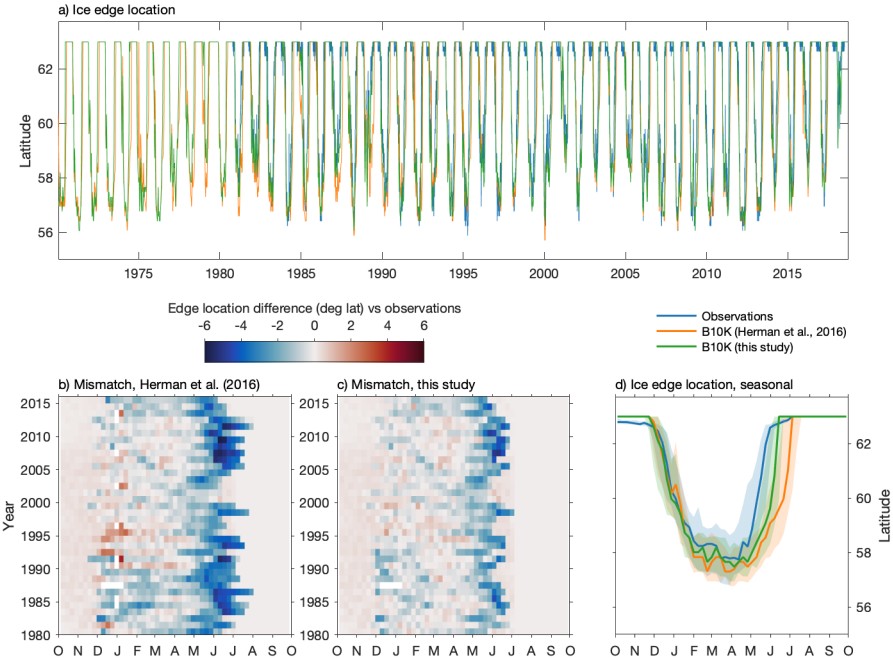

**Figure 3.** Skill assessment of the location of the sea ice edge along 170°W. Panel (a) depicts the entire time series in satellite observations, the Hermann et al. (2016) version of the Bering10K model, and the current version of the Bering10K model. Panels (b) and (c) indicate the seasonal and interannual mismatch between the location of the ice edge in the model versus the observations. Panel d shows the seasonal median (line) and 25th-75th percentile ranges (shaded region) of ice edge extent across the three datasets.

## 4.2 The cold pool

The Bering10K bottom temperature values clearly reproduce both spatial and temporal variability in the location and extent of the cold pool (Figure 4). During low-ice years, the cold pool is primarily located in the northwest portion of the eastern shelf, while in colder, high-ice years it extends throughout much of the middle domain and into Bristol Bay.

5   For a more quantitative assessment of skill, we looked at the correlation between annual time series of mean bottom temperature and cold pool index values in the groundfish survey dataset versus the models (Table 1, Figure 5). The Bering10K model values correlate very strongly with the observed values; correlation is highest when calculating the cold pool extent as defined by the 0 °C threshold ($R^2 = 0.940$), and $R^2$ values remain above 0.87 for all other metrics. To summarize model skill, we use a model efficiency metric after Stow et al. (2009), where a model efficiency value greater than 0 indicates more skill than a

10   simple average of the observed time series, and a value of one indicates a perfect match to the observed time series. The model efficiency for the Bering10K model is high across all metrics, ranging from 0.714–0.8. This is in sharp contrast to the mean bottom temperature and cold pool index metrics estimated by the coarser-resolution Climate Forecast System model. The cold pool produced by the CFS model lacks the detailed structure of colder waters seen in the observations, and fails to simulate

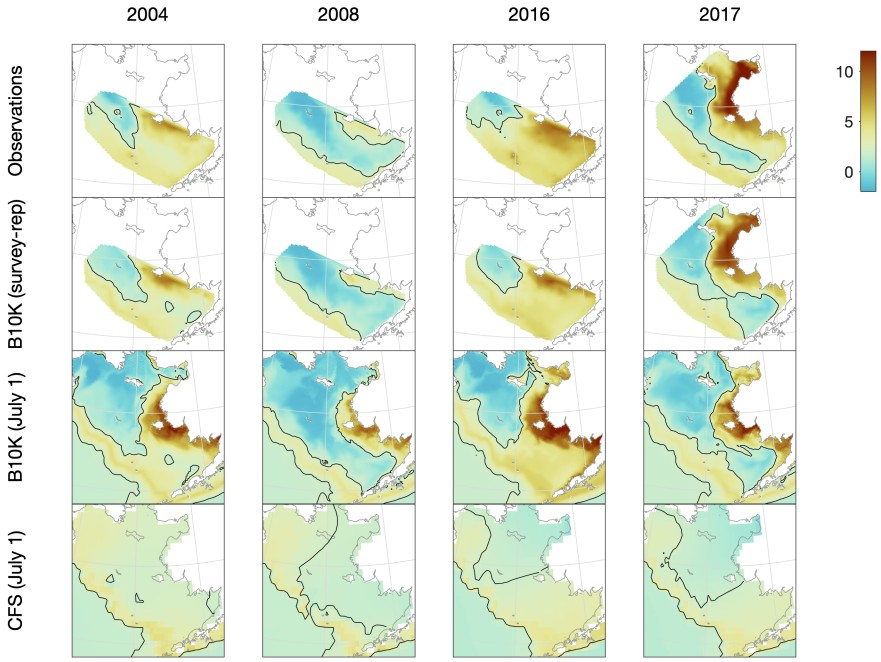

**Figure 4.** Comparison of observed, Bering10K-simulated, and CFS-simulated bottom temperature on the eastern shelf under a variety of conditions, including 2004 (a warm year), 2008 (a cold year), 2016 (an atypical warm year, where anomalously warm water from the Gulf of Alaska pushed the cold pool further northwest than usual) and 2017 (one of the few survey years when samples were collected from the northern Bering Sea). Discrete samples from the groundfish survey and the survey-replicated model datasets were interpolated to the model grid using natural neighbor interpolation; black contour line indicates the $2\,^\circ$C edge of the cold pool.

bottom waters below the $1\,^\circ$C threshold. The model efficiency metric suggests that the CFS model has much less skill in predicting mean interannual bottom temperature (MEF = 0.406), with only a marginal ability to capture the $2\,^\circ$C cold pool and no skill at all with respect to waters less than $1\,^\circ$C. This indicates that the dynamic downscaling offered by the higher-resolution Bering10K model is a necessary component in reproducing this feature.

## 4.3 Inner, middle, and outer shelf domains

Before analyzing whether the model properly reproduces the variations in vertical structure and mixing in the three shelf domains, it is important to note that the location of isobaths in our model are offset slightly compared to the real shelf bathymetry. Sigma-coordinate models such as the ROMS model used in this study are subject to computational errors in the horizontal pressure gradient along regions where topography is steep or the vertical gradient in a property is large (Shchepetkin, 2003); this issue often necessitates applying a smoothing filter to the bathymetry (Danielson et al., 2011). As a result, our modeled outer domain, as defined purely by location of isobaths, is generally narrower than that seen in any observations (Figure 6), particularly in the vicinity of the Pribilof Islands.



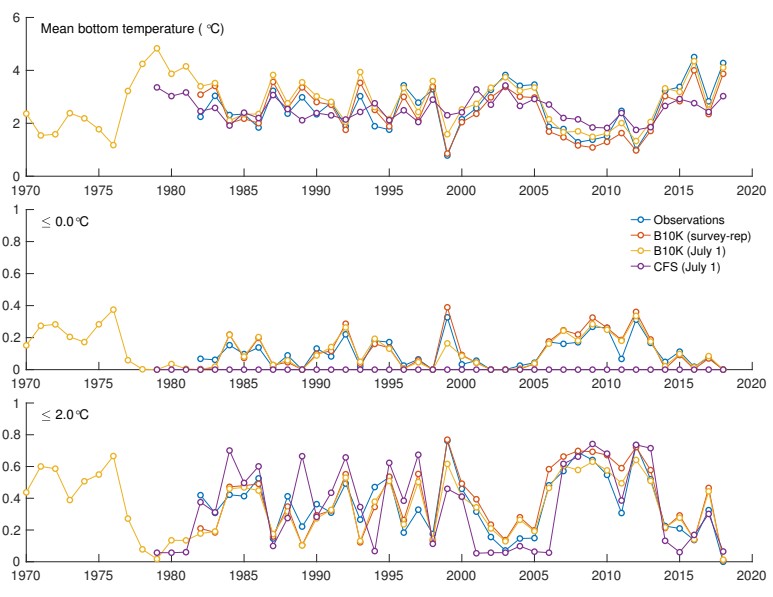

**Figure 5.** Annual indices of observed versus modeled bottom temperature, including average bottom temperature in the eastern shelf survey area, fraction of the survey area less than $0\,^{\circ}\mathrm{C}$, and fraction of the survey area less than $2\,^{\circ}\mathrm{C}$.

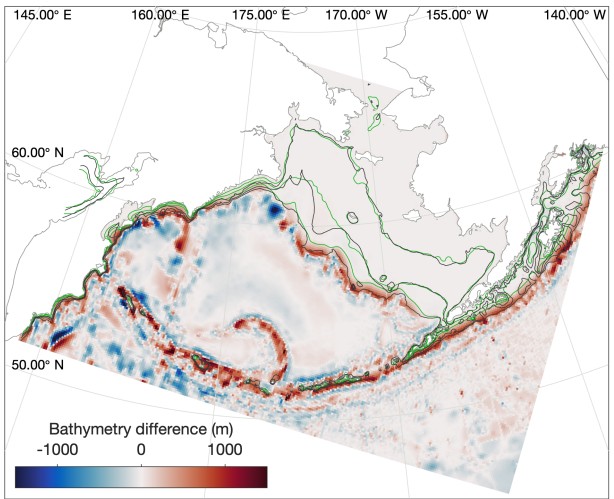

**Figure 6.** Difference in location between the real (black) and modeled (green) locations of the $50\,\mathrm{m}$, $100\,\mathrm{m}$, and $200\,\mathrm{m}$ isobaths on the eastern shelf. Colors indicate the adjustments to model bottom bathymetry (m) compared to the ETOPO5 dataset from which it was derived.





**Table 1.** Correlation, bias, root mean squared difference, and model efficiency for each model estimate of mean bottom temperature or cold pool index compared to the observed temperature/index. Statistics are applied to annual time series.

| Metric | Model | Correlation | Bias | RMSD | Efficiency |
|---|---|---|---|---|---|
| Mean bottom temperature | B10K (survey-rep) | 0.905 | −0.104 | 0.388 | 0.800 |
| Mean bottom temperature | B10K (July 1) | 0.903 | 0.201 | 0.424 | 0.761 |
| Mean bottom temperature | CFS (July 1) | 0.680 | −0.090 | 0.668 | 0.406 |
| 0 °C index | B10K (survey-rep) | 0.940 | 0.009 | 0.042 | 0.778 |
| 0 °C index | B10K (July 1) | 0.879 | 0.002 | 0.046 | 0.739 |
| 0 °C index | CFS (July 1) | NaN | −0.104 | 0.137 | −1.325 |
| 1 °C index | B10K (survey-rep) | 0.921 | 0.028 | 0.073 | 0.756 |
| 1 °C index | B10K (July 1) | 0.897 | 0.010 | 0.067 | 0.794 |
| 1 °C index | CFS (July 1) | 0.225 | −0.195 | 0.242 | −1.709 |
| 2 °C index | B10K (survey-rep) | 0.884 | 0.026 | 0.100 | 0.714 |
| 2 °C index | B10K (July 1) | 0.880 | −0.009 | 0.090 | 0.769 |
| 2 °C index | CFS (July 1) | 0.696 | 0.005 | 0.178 | 0.100 |

The simulated patterns in vertical stratification follow those expected across the three domains (Figure 7). During the winter, the majority of the shelf is well-mixed vertically. Stratification first appears in early spring along the outer domain, and by summer is also seen throughout the middle domain. Given the relatively coarse vertical resolution of our model, the distinction in the vertical structure between the middle and outer domains is not well-resolved. However, the model does reproduce the structural front expected between the unstratified inner domain and thermally-stratified middle domain during the summer months. The exact location of this front varies both seasonally and from one year to the next, but is generally located just inside the 50 m isobath (Figure 8). The front location is relatively consistent across the southern shelf region, though possibly further inshore than would be predicted by the 50 m isobath near Cape Newenham at the north end of Bristol Bay; it moves further offshore and its location becomes more temporally variable further north. The clear structural front begins to disappear north of Nunivak Island. Stratification in the northern Bering Sea and Norton Sound area is much more strongly influenced by salinity, especially near the large outflows of the Yukon River, and in this region the clear demarcation between inner, middle, and outer domains disappears.

Horizontal movement in the model, as expected, is dominated by tidal frequencies across the shelf domain, with low annually-averaged net velocities. There is a small net counterclockwise flow along the southern edge of the eastern shelf and then northward within the inner domain, with a small net transport from off-shelf to inner shelf waters (Figure 9). Cyclical circulation patterns seen near the southern and eastern boundaries of the model domain are likely an artifact of the boundary conditions.





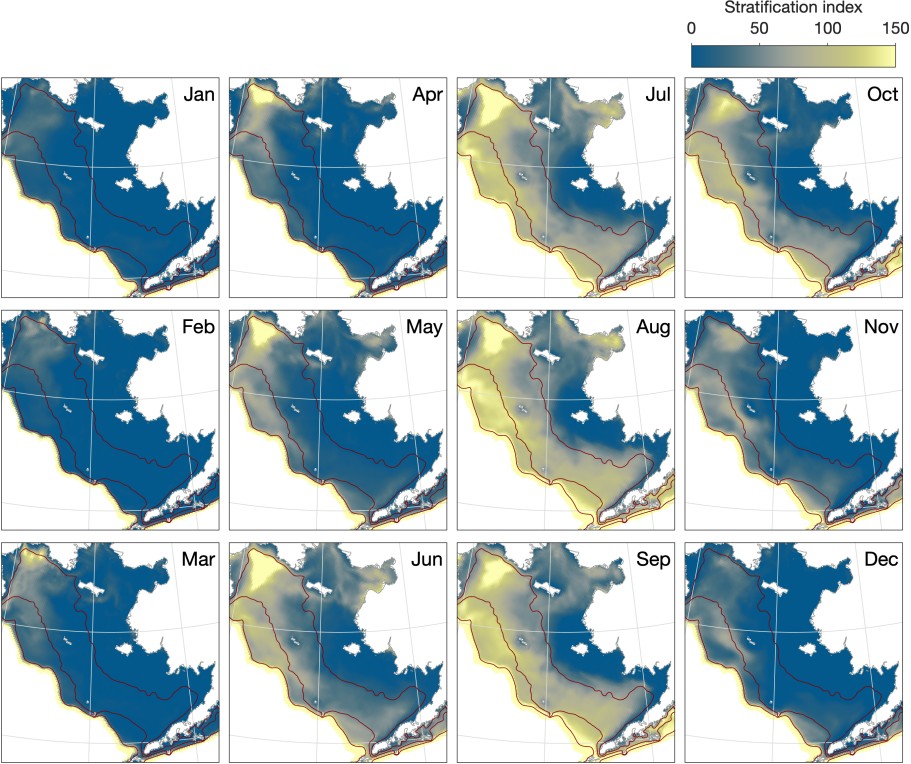

**Figure 7.** Monthly climatologically-averaged simulated stratification across the Eastern Bering Sea shelf. Dark red lines indicate the primary 50 m, 100 m, and 200 m contour lines based on the model shelf bathymetry.

## 4.4 Spatial and temporal patterns in primary production

Satellite ocean color measurements suggest that phytoplankton blooms in the Bering Sea first reach observable levels of chlorophyll in late February to early March, primarily on the eastern shelf in regions recently vacated of ice. As light levels and temperature increase throughout the domain in summer, chlorophyll levels increase both on the shelf and along the shelf slope, but remain low over the western side of the basin, where iron is limiting. The bloom peaks in late May to early June, then steadily decreases through September. A late fall bloom, smaller in magnitude than the earlier spring bloom, can be seen in October along the eastern shelf.

While the BESTNPZ model produces annual cycles of primary productivity of approximately the correct magnitude compared to these observations, it does not capture many of the nuances in spatiotemporal variability (Figure 10). In early spring, the model does not appear to capture the early ice-associated growth along the eastern shelf. Within the pelagic ecosystem, low growth rates governed by low temperature-mediated maximum production rates coupled with strong light limitation prevent any significant accumulation of phytoplankton. While concentrations within the ice algae state variable can reach approximately $70 \, \mathrm{mg \, m^{-3}}$ (monthly climatological average) within the thin skeletal ice layer, this biomass does not contribute significantly



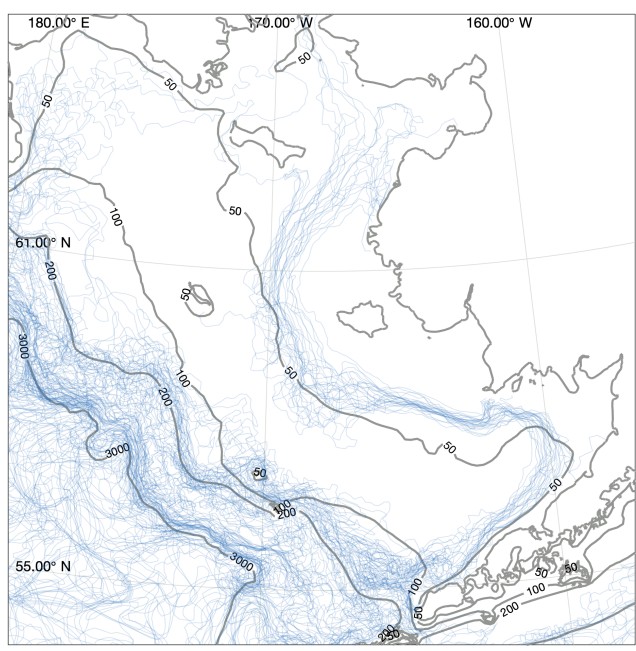

**Figure 8.** Location of structural fronts estimated by horizontal gradients in maximum vertical temperature gradient. Each blue line indicates a front location estimated from a single weekly-averaged time point taken from the August and September values between 2000 and 2010. Contours indicate model domain bathymetry. Note that fronts over water deeper than 200 m may reflect artifacts of the coarsening vertical resolution rather than true changes in vertical gradients.

to the pelagic large phytoplankton concentration once ice melts due to dilution coupled with unfavorable growth conditions in the underlying water. We do not include these ice algal numbers in the optically-weighted chlorophyll numbers used in our satellite comparison because these satellite measurements do not typically capture the spectral signals of ice algal pigments (Wang et al., 2018).

The spring bloom in the model begins once light and temperature levels increase in April. The first stages of the bloom resemble observations, with concentrations highest along the shelf slope and along the western shelf. However, rather than producing a short-lived bloom that drops once macronutrients are exhausted, the model allows for a sustained summer bloom. This bloom is driven by regenerated production; ammonium is produced from phytoplankton and zooplankton respiration, as well as quick remineralization of egested detritus, especially of the slow-sinking detritus group fed by small phytoplankton non-

predatory mortality and microzooplankton egestion and non-predatory mortality. This pattern is seen both on the eastern shelf and throughout much of the deep basin, where iron does not appear to play its expected role in limiting primary production. In the basin, production levels fall off as macronutrients are exhausted in early July; on the eastern shelf, however, high fluxes of ammonium from the benthos drive sustained production throughout the summer and into early fall. In late fall, modeled chlorophyll levels appear more similar to the satellite patterns, with production primarily limited to the middle domain of the

eastern shelf.



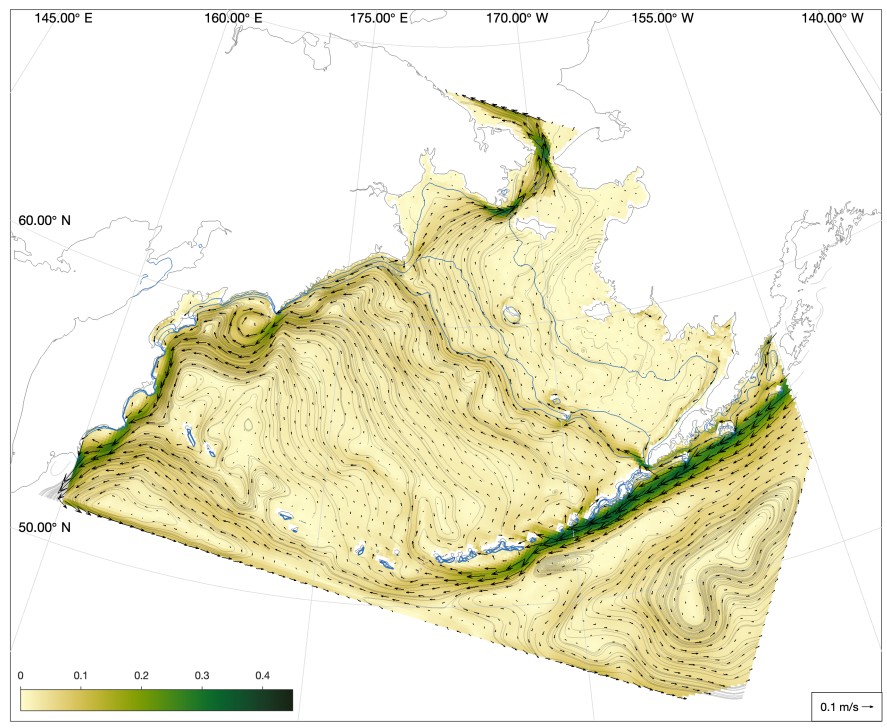

**Figure 9.** Mean currents at $40\,\text{m}$ depth (or in bottom layer, for locations shallower than $40\,\text{m}$) across the model domain, averaged over 2000-2010. Blue lines indicate bottom contours of $50\,\text{m}$, $100\,\text{m}$, and $200\,\text{m}$. Shading indicates the magnitude of the flow. Light gray lines indicate flow streamlines.

The late bias in the spring phytoplankton bloom is also evident when comparing model output to mooring measurements at the M2 mooring (Figure 11). The model captures the predominant bloom characteristics: the bloom begins with a large, diatom-dominated bloom starting in the surface waters and then migrating deeper as surface nutrients are depleted, then decreases during the summer months, with some subsurface chlorophyll remaining at the bottom of the mixed layer. However, the modeled bloom begins in mid-April, on average, later than the mid-March to early April start seen in the mooring data. Fall blooms in September and October are spurred by increased mixing and are short and localized in the observations; the modeled fall bloom matches the timing in the observations well.

## 4.5 Plankton community composition

The phytoplankton community composition in the model reflects the expected balance between the small and large functional groups (Figure 12). Throughout the majority of the domain, the spring bloom is heavily dominated by the large phytoplankton group. The one exception to this is along the shallow, well-mixed inner domain, where low macronutrient levels favor the small phytoplankton group for most of the year, with a small contribution of large phytoplankton in early summer when the bloom





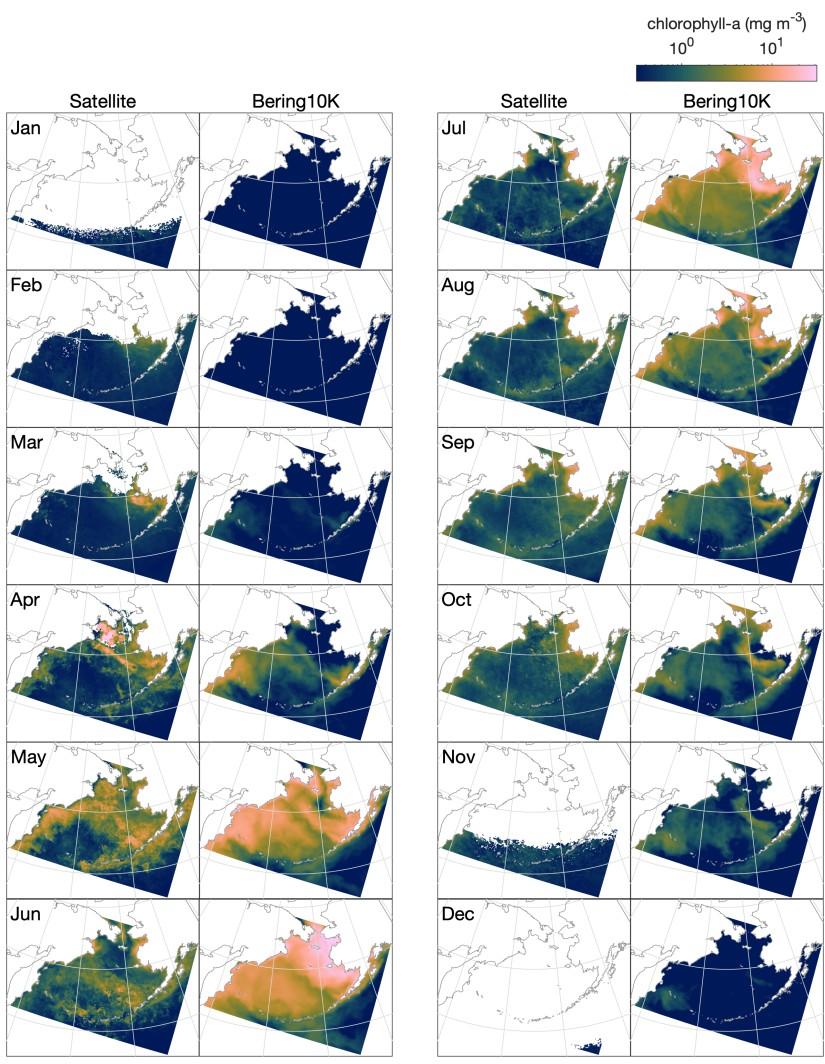

**Figure 10.** Monthly mean, optical-depth-averaged chlorophyll in the Bering10K BESTNPZ model versus satellite-estimated values.

begins there. In the inner part of the middle shelf, the large phytoplankton levels decrease following the initial bloom but small phytoplankton biomass continues to rise through the fall. Moving further outward along the shelf, late summer and early fall biomass is low across both functional groups.

Within the zooplankton community, we see little variation between the relative dominance of the functional groups across the shelf transect (indicated by gold circles and labels in Figure 1). In all locations, microzooplankton are the dominant group. However, their biomass is often only slightly higher than that of the summed mesozooplankton groups. Within the mesozooplankton groups, very little variation in their relative contribution to the biomass pool is seen either spatially or temporally. The only big change to zooplankton community coincides with the hard-coded diapause movement of the two large cope-


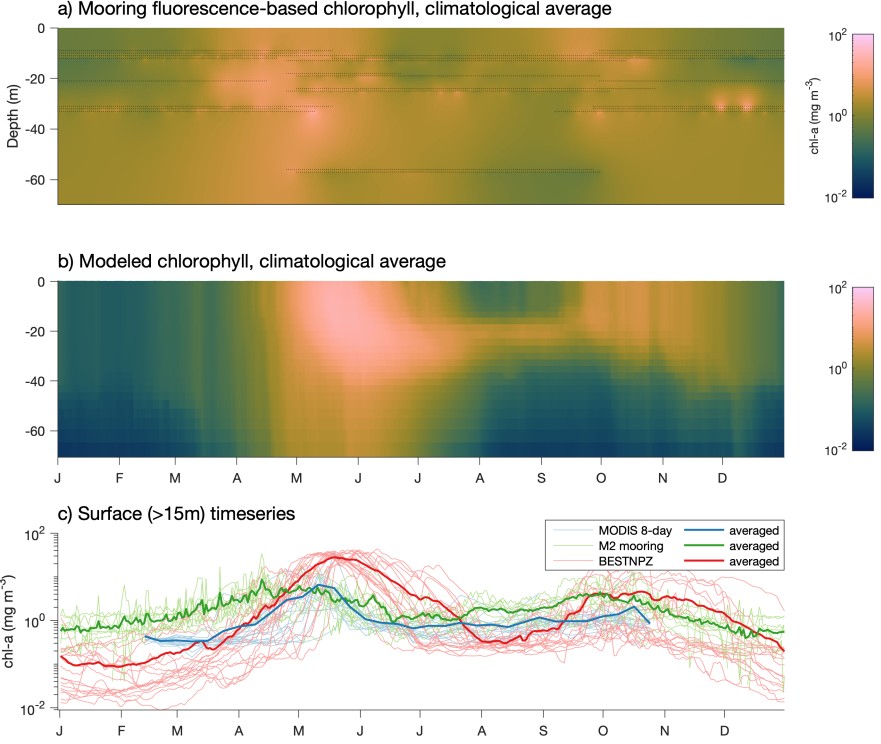

**Figure 11.** a) Fluorescence-derived chlorophyll measurements at the M2 mooring location. Black dots indicate individual measurements, while the shaded values indicate estimates using a spring model interpolant (John D'Errico, 2016). b) Modeled chlorophyll extracted at the M2 mooring location over the same time period as the mooring data (2004-2017). c) Surface-only climatological time series at the M2 location from mooring-based measurements, via satellite (within a 1-deg box around the M2 location derived from MODIS Aqua OCI-algorithm 4-km 8-day-average images), and in the BESTNPZ model.

pod groups; because these groups cease grazing when they enter diapause, their populations quickly drop during the diapause period. Off-shore large copepods (NCaO) die if they encounter the ocean floor prior to reaching their prescribed overwintering depth of $400\,\mathrm{m}$, and this effectively keeps this functional group constrained to deep water locations. Lacking any similar depth-based restrictions on their process rates, the remaining large zooplankton groups can be found throughout the domain, regardless of whether they are designated as onshore or offshore in name.

The model does capture a gradient in timing of the zooplankton population, with offshore populations being established early in the spring while on-shelf populations do not appear until early summer. However, limited observations suggest that early spring offshore zooplankton increases precede the spring phytoplankton bloom offshore (Hunt et al., 2016; Harvey et al., 2012). This timing difference is not captured by the model; instead, the overwintering population in the model is reduced to negligible amounts and only begins to increase again once the primary productivity in the region reaches sufficient levels.

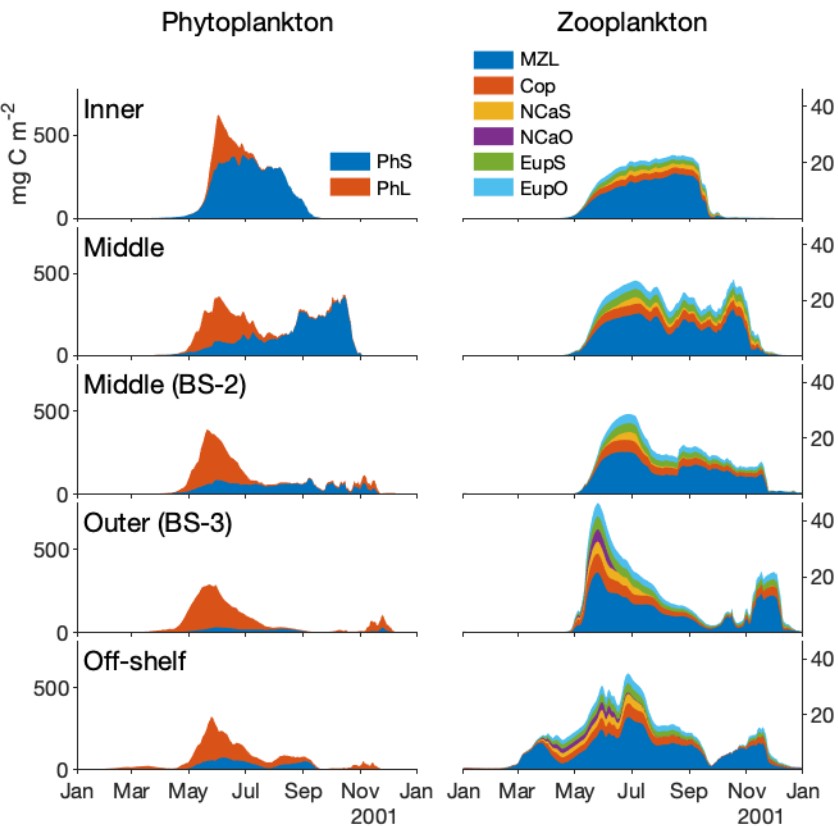

**Figure 12.** Integrated biomass of phytoplankton and zooplankton groups in the southeastern Bering Sea shelf during the final year of the spin up simulation. Colors indicate the fraction of the total value from each functional group. Values were extracted from the model at the five locations indicated by gold dots in Figure 1 (including the M2 mooring location)

.

## 5   Discussion

The Bering10K model correctly reproduces a variety of physical processes known to influence primary production in the Bering Sea. Overall patterns of sea ice cover, including interannual variations in the maximum extent of sea ice and the date of ice retreat, are well-captured by the model. The exact date of sea ice retreat tends to lag observations by approximately two weeks.

5   This lag could, theoretically, lead to subsequent problems in the timing of phytoplankton blooms; in the model's current state, ice algae and ice-associated blooms are poorly resolved regardless of ice melt accuracy or lack thereof, but improving the ice retreat timing should remain a focus of future improvements to the physical model.





The location and extent of the cold pool in the Bering Sea is often used as an index of biophysical variability across the Bering Sea shelf (e.g. Siddon and Zador, 2018), and therefore capturing both spatial and temporal variation in cold pool extent is key for a model to be useful in this region. The Bering10K model performs well on this point, with very high correlation to observations and very small biases. It is encouraging to note that the location of the simulated cold pool in 2016 is offset to the

5 northwest compared to other warm years, replicating the position seen in the observations. During this year, anomalously warm water from the Northeast Pacific (Bond et al., 2015) prevented the cold pool from extending as far southeast as it typically does during the summer months. The ability of the model to capture these anomalous conditions lends promise to its capability to simulate novel conditions that may arise when simulating future conditions.

The Bering10K model shows good replication of both cross-shelf and along-shelf differences in stratification. In the south,

we see a distinct well-mixed inner domain, with a sharp transition to a stratified middle/outer domain occurring near the $50\,\mathrm{m}$ isobath during summer. The model's distinction between the middle and outer domains is less defined than in observations, likely due to the limited 30 vertical layers used in the model combined with bathymetric smoothing. In the northern portions of the eastern shelf, these thermal stratification domains disappear. Salinity is more variable in the north than in the south, driven by both sea ice formation and melt, as well as the large freshwater contribution of the Yukon River. As a result, stratification in

the north is driven more strongly by salinity than in the south.

While the physical dynamics of the model perform well within the Bering Sea itself, the modeling domain south of the Aleutian Islands should be treated with caution, as it is close to the model's open boundary and artifacts associated with boundary conditions are expected.

Despite its strong skill in replicating the underlying physical features of the Bering Sea that are thought to influence primary

production, the Bering10K-BESTNPZ model has limited skills in reproducing observed spatial and temporal patterns of primary production. The degraded performance in the biology realm is due to several interacting deficiencies in model process equations and parameterizations.

Throughout the deep basin, according to observations, iron levels should be low and limit primary production. The model includes only a simple representation of iron, using continuous relaxation to an empirical depth profile. While the low surface

concentration prescribed in our model's basin is consistent with observations, this mechanism of replenishment through relaxation is not one that reflects the true complexity of iron cycling in the ocean. In their observations of phytoplankton growth rates in the Green Belt along the slope, Aguilar-Islas et al. (2007) noted that even in this highly-productive location, the diatoms showed signs of iron stress, and dissolved iron levels remained low here compared to the shelf. They hypothesized that production in this region was maintained at its observed level due to a small but persistent source of iron being mixed from deep water

along the shelf break, rather than a large iron source that fully alleviated iron limitation. The climatological nudging used in our model provides exactly this— a small but persistent source of continuous dissolved iron— throughout the domain, rather than only along the shelf slope. In order to properly capture the HNLC characteristics of the deep basin, a more mechanistic model for iron, with an explicit source near the sediments only rather than throughout the water column, is likely necessary.

Across the eastern shelf, in terms of mean seasonal cycles, modeled phytoplankton biomass and primary production levels

are more in line with observations, and reflect the dominant seasonal pattern observed on the southeast middle shelf of a strong





spring bloom, followed by low summer biomass, then a smaller late fall bloom. However, many of the prevailing hypotheses of energy flow in the Bering Sea focus not on the mean state of the phytoplankton bloom but rather on interannual variability, particularly related to the interplay between temperature, stratification, and nutrient availability during the initial stages of the spring bloom (Stabeno et al., 2001; Coyle et al., 2008; Hunt et al., 2010). This in turn effects the phenological patterns seen in the seasonal production patterns simulated by the model, and could lead to shortcomings in capturing interannual variability.

The first issue is that the spring bloom begins nearly a month later than it should. This is particularly apparent in the north (Figure 10), where observations indicate that phytoplankton blooms should occur both on and under the sea ice. In our model, ice algae biomass is insignificant compared to pelagic phytoplankton biomass, and pelagic production is too strongly limited by both temperature and light for any significant growth to begin during March or April as it should. Currently, the lack of early spring growth also leads to a failure of the model to react to interannual differences in ice extent and retreat timing.

The limit on ice algae biomass in our model is primarily a limitation of the conceptual framework imposed on this state variable. By assuming that the ice algae are confined within the very thin skeletal ice layer, while at the same time allowing a continuous convective exchange (see subsubsection A3.8) between this layer and the surface layer of the water column, modeled ice algae can never grow to much higher concentrations than would be found in an equivalent thin layer of water. This framework does not account for other aggregation types often seen in ice algal communities, such as nets or strands on the underside of the ice surface (Ambrose et al., 2005). Once the ice melts and the ice algae are released into the water, the model framework immediately transfers this pool of biomass to the large phytoplankton group, where it is then subject to the same controls on growth rate as the pelagic-originating phytoplankton.

For pelagic phytoplankton groups, the inability to capture early ice-associated blooms is primarily an inadequacy of the equations and parameters chosen to represent photosynthetic processes. The parameters that define each group's photosynthesis-irradiance curve, as well as those setting the maximum potential light- and nutrient-replete growth rates (a function of temperature), originated from a comparable model for the Gulf of Alaska (Hinckley et al., 2009). However, the phytoplankton community of the Bering Sea includes many Arctic species that are physiologically adapted to grow in both lower temperatures and under a wider range of light levels experienced near and under the sea ice. For example, ice-associated blooms occur in very thin upper layers of the water column left behind by ice melt; these layers are typically only $1\,\mathrm{m}$ to $2\,\mathrm{m}$ thick and characterized by temperatures of around $-1.7\,^{\circ}\mathrm{C}$ (Hunt et al., 2010). At this temperature, the model equations require approximately $2\,\mathrm{W\,m^{-2}}$ surface irradiance to balance respiration and non-predatory mortality costs, even in the absence of any nutrient limitation or grazing losses. But modeled under-ice surface irradiance typically remains below this level when ice of more than approximately $0.5\,\mathrm{m}$ thickness is present. Therefore, the chosen set of equations do not appear appropriate to reproduce the dynamics of under-ice and ice-edge blooms. We also note that while the parameters and equations controlling the maximum potential growth rate of phytoplankton (Equation A16) produce reasonable rates within the temperature ranges seen in the hindcast period in this geographical domain, they increase exponentially above this range, well outpacing respiration rate increases; a temperature increase of $5\,^{\circ}\mathrm{C}$ to $10\,^{\circ}\mathrm{C}$ would push these rates well beyond the physiological limits of phytoplankton division rates.





Another key problem seen for phytoplankton across the shelf is the absence of any strong macronutrient limitation following the initial spring bloom. Observations indicate that ammonium can reach high quantities (up to $15\,\mathrm{mg\,m^{-3}}$) following the initial bloom due to both phytoplankton decomposition and benthic processes (Whitledge et al., 1986; Aguilar-Islas et al., 2007). However, this ammonium is typically concentrated in the deeper shelf water in the observations, while the model tends to accumulate material in a subsurface layer, with continuous high turnover of ammonium in surface waters of the shelf from spring through fall. This likely indicates that decomposition processes in the model are proceeding too quickly, particularly in the slow-sinking detrital group whose coupled remineralization rates and sinking rates result in this material only reaching about $50\,\mathrm{m}$ in depth before being completely converted to ammonium. Small phytoplankton mortality and excretion as well as excretion and egestion by microzooplankton are the primary sources for the slow-sinking detritus group. The model also produces a very robust zooplankton community that persists from the start of the spring bloom until well into the winter months (mid-December), and whose byproducts of respiration, excretion, and egestion feed into this rapid, continuous cycle of regenerated primary production (see Supplement). In general, the process equations concerning the sinking and remineralization of organic material, both in the water column and on the seafloor, are very simplistic (see subsubsection A3.7 for details). A single timescale for remineralization is used for all detrital groups, with no distinction between the lability of different pools of organic material. There is also no mechanism available to account for aggregation of material, which could lead to faster sinking speeds and lower remineralization rates of organic matter. Because of the broad, shallow nature of the eastern Bering Sea shelf, remineralization rates play a strong role in determining the concentrations of macronutrients and the strength of benthic-pelagic coupling. These processes should be a focus of future development in this model.

In contrast to the under-resolved detrital pools, the mesozooplankton groups included in the BESTNPZ model appear to be over-resolved in terms of functional differences capable of being differentiated between with this type of biomass box model. The prey preferences encoded into the feeding behaviors of each group produce only very small variations in their relative contribution to the overall biomass pool. Instead, the five mesozooplankton groups (small copepods, on- and off-shore large copepods, and on- and off-shore euphausiids) effectively function as a single herbivorous/microzooplanktivorous zooplankton functional group. The structure of the model also leads to plankton dynamics that are constrained primarily by the balance of instantaneous production and loss rates. While appropriate for simulating phytoplankton bloom dynamics, this style of model is not well-suited for capturing the dynamics of zooplankton life stages (for example, the necessity of overwinter survival in order to spawn a new generation in the spring), or the nuanced differences between the survival strategies of various species. Mesozooplankton populations drop to a negligible level during the winter due to the absence of sufficient prey to balance continued respiratory and non-predatory losses, and overwintering success (or lack thereof) has almost no effect on the resulting zooplankton populations in summer. In order to truly capture the gradients in relative success of different copepod and euphausiid groups throughout the domain, a model that better captures winter survival strategies (e.g. a life-stage-resolving model or another means of introducing latency between feeding, respiration, and non-predatory mortality) is likely necessary.





# 6 Conclusions

Overall, the BESTNPZ model coupled to the Bering10K regional ocean model demonstrates considerable skill in replicating observed horizontal and vertical patterns of water movement, mixing, and stratification, as well as the temperature and salinity signatures of various water masses throughout the Bering Sea. However, its ability to replicate large scale patterns in nutrient

cycling, primary production, and zooplankton community composition, particularly with respect to the interannual variations that are important in a fisheries management context, is limited. In its current form, the Bering10K model can offer key insights into the physical processes that may affect higher trophic level species directly. In particular, it offers a useful supplement to examine physical features in areas and at times of the year that are difficult or impossible to survey due to sea ice cover or harsh weather. However, we caution that the use of the biological state variable output should be limited until the model is better able

to capture observed characteristics of the Bering Sea phytoplankton and zooplankton communities.

*Code availability.* Source code for the Bering10K Regional Ocean Modeling System domain, including the BESTNPZ biological model, are available on Github at https://github.com/beringnpz/roms-bering-sea, DOI: zenodo.3376314

*Video supplement.* Supplementary material, including additional figures and animation, can be viewed at https://beringnpz.github.io/roms-bering-sea/gmdval_supplement. The code for this website is included in the primary roms-bering-sea repository on the gh-pages branch and

is archived under DOI: 10.5281/zenodo.3376317

# Appendix A: Documentation for the Bering Sea Ecosystem Study Nutrient Phytoplankton Zooplankton Model (BESTNPZ)

## A1  Summary and notation

This section provides a mathematical overview of processes through which biological state variables exchange material with

each other in the BESTNPZ model.

The BESTNPZ model assumes a model geometry that includes $N$ water column layers, a single benthic layer of unspecified depth, and a skeletal ice layer with a constant thickness $h_{sice}$. The skeletal ice layer refers to the base of an ice sheet; this very thin layer is characterized by a looser crystal structure than the more solid ice overlying it, and is the site of the most rapid algal growth in ice (Arrigo et al., 1993; Jin et al., 2006). Within this geometry, BESTNPZ tracks the concentration of 19 biological

state variables (Table A1).

Exchange of material between these state variables, and across vertical layers within a single state variable, results from a variety of processes. In the code, and in this description, these processes are divided into three types.



**Table A1.** Biological state variables in the BESTNPZ model.

| Index | Variable | Description | Units |
|---|---|---|---|
| 1 | $NO3$ | nitrate | $\mathrm{mmol\,N\,m^{-3}}$ |
| 2 | $NH4$ | ammonium | $\mathrm{mmol\,N\,m^{-3}}$ |
| 3 | $PhS$ | small phytoplankton (cells less than $10\,\mu$m diameter) | $\mathrm{mg\,C\,m^{-3}}$ |
| 4 | $PhL$ | large phytoplankton (bloom forming diatoms) | $\mathrm{mg\,C\,m^{-3}}$ |
| 5 | $MZL$ | microzooplankton | $\mathrm{mg\,C\,m^{-3}}$ |
| 6 | $Cop$ | small-bodied copepods (e.g. *Pseudocalanus* spp.) | $\mathrm{mg\,C\,m^{-3}}$ |
| 7 | $NCaS$ | on-shelf large-bodied copepods (primarily *Calanus marshallae*) | $\mathrm{mg\,C\,m^{-3}}$ |
| 8 | $EupS$ | on-shelf euphausiids (primarily *Thysanoessa raschii*) | $\mathrm{mg\,C\,m^{-3}}$ |
| 9 | $NCaO$ | off-shelf large-bodied copepods (primarily *Neocalanus* spp.) | $\mathrm{mg\,C\,m^{-3}}$ |
| 10 | $EupO$ | off-shelf euphasiids (primarily *Thysanoessa inermis*) | $\mathrm{mg\,C\,m^{-3}}$ |
| 11 | $Det$ | slow-sinking detritus | $\mathrm{mg\,C\,m^{-3}}$ |
| 12 | $DetF$ | fast-sinking detritus | $\mathrm{mg\,C\,m^{-3}}$ |
| 13 | $Jel$ | jellyfish (*Chrysaora melanaster*) | $\mathrm{mg\,C\,m^{-3}}$ |
| 14 | $Fe$ | iron | $\mathrm{\mu mol\,Fe\,m^{-3}}$ |
| 15 | $Ben$ | benthic infauna (bivalves, amphipods, polychaetes, etc.) | $\mathrm{mg\,C\,m^{-2}}$ |
| 16 | $DetBen$ | benthic detritus | $\mathrm{mg\,C\,m^{-2}}$ |
| 17 | $IcePhL$ | ice algae | $\mathrm{mg\,C\,m^{-3}}$ |
| 18 | $IceNO3$ | ice nitrate | $\mathrm{mmol\,N\,m^{-3}}$ |
| 19 | $IceNH4$ | ice ammonium | $\mathrm{mmol\,N\,m^{-3}}$ |

The first type of process, described in subsection A2, includes redistribution of state variables due to movement of the water or ice in which they reside. The majority of these calculations (e.g. advection and diffusion of water and ice) take place outside the biological module, and follow the default ROMS behavior for biological tracer variables. The one exception is the exchange of NO3 and IceNO3, NH4 and IceNH4, and PhL and IcePhL due to the formation or loss of ice in a grid cell.

5 The second process type we term source-minus-sink processes (subsection A3); these processes take place within a single depth layer and involve transfer of biomass from one state variable to another. For notation, each source-minus-sink flux process is represented in this document as a function of the source and sink state variables, respectively. For example, $Abc(\mathrm{X,Y})$ is the flux rate of material from group X to group Y via the $Abc$ process.

The final category (subsection A4) is vertical movement, where the concentration of state variables is redistributed within 10 the water column due to sinking or rising movement of the state variables within the water (note that this is separate from vertical advection of tracers due to movement of the water itself.)





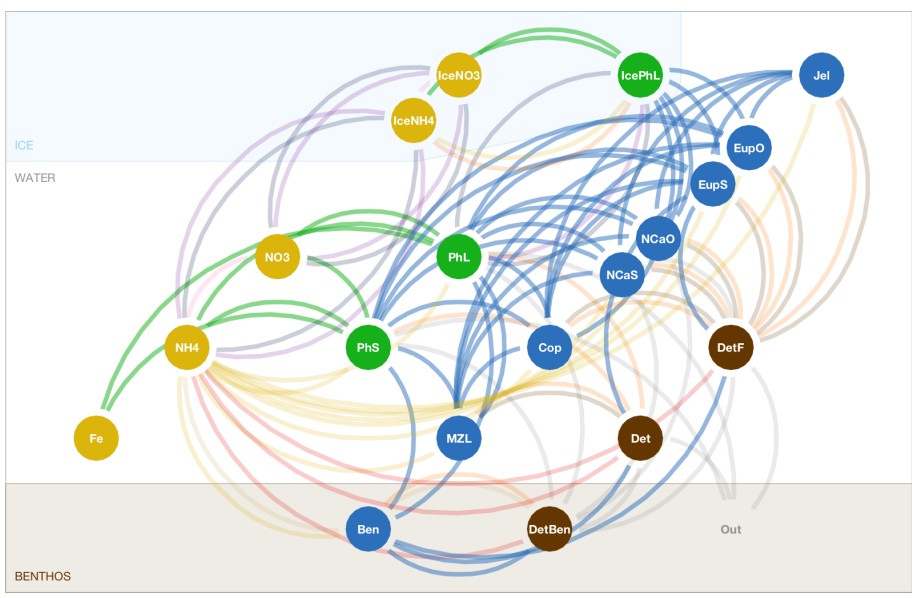

**Figure A1.** Schematic of the BESTNPZ model. Edges (lines) represent fluxes between state variables (gold = nutrient, green = producer, blue = consumer, brown = detritus), and curve clockwise from source node to sink node. Edge colors indicate process type: green = primary production, blue = grazing and predation, brown = egestion, gold = respiration, red = remineralization, pink = nitrification, orange = non-predatory mortality, tan = excretion, purple = convective exchange, gray = sinking to seafloor, navy = freezing/melting of ice

The three types of processes are calculated sequentially in the code, such that changes due to ice loss and formation are calculated first, followed by the total rate of change due to source-minus-sink processes, and finally redistribution due to vertical movement.

Several of the equations in this section rely on state or diagnostic variables that come from the physical model or from the ROMS grid geometry. See Table A3 for a description of these variables and their notation in this document. Additional parameters derived from biological input parameters are listed in Table A4.

Table A3: Notation and description for variables deriving from the physical model.

| Variable | Name | Units | Details |
|---|---|---|---|
| $z$ | depth | m | relative to mean sea level (positive above mean sea level, negative below) |
| $\zeta$ | free surface height | m | relative to mean sea level |
| $h$ | bathymetry | m | depth of the ocean floor in a given grid cell, measured from mean sea level and expressed as a positive number |
| $h_k$ | thickness of depth layer $k$ | m | varies as a function of $h$ and $\zeta$, $k = 1$ corresponds to bottom layer and $k = N$ to the top layer |





| | | | | | |
|---|---|---|---|---|---|
| $h_{ice}$ | thickness of sea ice | m | | | |
| $a_{ice}$ | fraction of grid cell covered by sea ice | $--$ | | | |
| $T$ | temperature | °C | | | |
| $\epsilon$ | machine epsilon | $--$ | small value used to avoid 0 problems | | |
| $c_I$ | light conversion factor | $\mathrm{E\,m^{-2}\,d^{-1}\,W^{-1}\,m^2}$ | Thimijan and Heins (1983) provide the conversion factor of $4.57\,\mathrm{\mu E\,s^{-1}\,m^{-2}}$ per 1 $\mathrm{W\,m^{-2}}$ (i.e. $0.394\,848\,\mathrm{E\,m^{-2}\,d^{-1}\,W^{-1}\,m^2}$) for the 400-700 nm band assuming a light source of "sun and sky, daylight". | | |
| $I_0$ | surface irradiance | $\mathrm{E\,m^{-2}\,d^{-1}}$ | converted from surface heat flux to photon flux assuming seawater density = $1025\,\mathrm{kg\,m^{-3}}$, heat capacity = $3985\,\mathrm{J\,kg^{-1}\,°C^{-1}}$, and absorption wavelengths appropriate to chlorophyll (see $c_I$, above). Note that this represents below-ice irradiance when ice is present. | | |

Table A4: Notation and description for input parameters applicable to all biomass rate of change processes.

| Variable | Description | Units | Relevant input parameter(s) | | |
|---|---|---|---|---|---|
| | | | Group | Parameter | Value |
| $h_{sice}$ | thickness of skeletal ice layer | m | | aidx | 0.02 |
| $\xi$ | N:C ratio | $\mathrm{mmol\,N\,(mg\,C)^{-1}}$ | | xi | 0.0126 |
| $FeC$ | Fe:C ratio | $\mathrm{\mu mol\,Fe\,(mg\,C)^{-1}}$ | | FeC | 0.000\,166\,7 |

To express the concentration of a biological state variable X, we use the notation [X], with units corresponding to those in Table A1. All intermediate fluxes are expressed in terms of $\mathrm{mgC}$ for simplicity. Conversions between units assume constant stoichiometry for all living and detrital groups.

Because many of the variable names used in these equations involve multi-letter and mixed-case notations, we've chosen to
5   use dot notation for all instances of multiplication in this document. Please note that this indicates simple element-by-element multiplication in the BESTNPZ code, not a dot product.

## A2  Ice formation and loss

Although the primary ROMS sea ice model tracks ice presence in terms of fraction grid cell coverage, the BESTNPZ biological module uses a simpler scheme where ice presence is treated as a binary condition. When the ice thickness ($h_{ice}$) in a grid cell
10   is greater than the prescribed thickness of the skeletal ice layer ($h_{sice}$) and the grid cell has at least 50% ice cover (as indicated by the $a_{ice}$ variable), we assume the entirety of that grid cell now supports the ice-related biological processes in a thin layer of skeletal ice covering the entire grid cell and located just above the free surface of that grid cell.





If sea ice appears in a grid cell between the previous time step and the current time step, the large phytoplankton, nitrate, and ammonium in the top layer are redistributed such that the top water column layer and the skeletal ice layer receive equal concentrations of each tracer by volume.

$$[\text{IcePhL}]_{ice} = \frac{[\text{PhL}]_N \cdot h_N}{h_{sice} + h_N} \tag{A1}$$

$$[\text{PhL}]_N = \frac{[\text{PhL}]_N \cdot h_N}{h_{sice} + h_N} \tag{A2}$$

$$[\text{IceNO3}]_{ice} = \frac{[\text{NO3}]_N \cdot h_N}{h_{sice} + h_N} \tag{A3}$$

$$[\text{NO3}]_N = \frac{[\text{NO3}]_N \cdot h_N}{h_{sice} + h_N} \tag{A4}$$

$$[\text{IceNH4}]_{ice} = \frac{[\text{NH4}]_N \cdot h_N}{h_{sice} + h_N} \tag{A5}$$

$$[\text{NH4}]_N = \frac{[\text{NH4}]_N \cdot h_N}{h_{sice} + h_N} \tag{A6}$$

Likewise, when ice disappears between the previous step and the current one, all material in the skeletal ice layer is moved to the top layer of the water.

$$[\text{PhL}]_N = \frac{[\text{PhL}]_N \cdot h_N + [\text{IcePhL}_{ice}] \cdot h_{sice}}{h_N} \tag{A7}$$

$$[\text{NO3}]_N = \frac{[\text{NO3}]_N \cdot h_N + [\text{IceNO3}_{ice}] \cdot h_{sice}}{h_N} \tag{A8}$$

$$[\text{NH4}]_N = \frac{[\text{NH4}]_N \cdot h_N + [\text{IceNH4}_{ice}] \cdot h_{sice}}{h_N} \tag{A9}$$

$$[\text{IcePhL}]_{ice} = 0 \tag{A10}$$

$$[\text{IceNO3}]_{ice} = 0 \tag{A11}$$

$$[\text{IceNH4}]_{ice} = 0 \tag{A12}$$

### A3 Source-minus-sink processes

Unless otherwise specified, all processes detailed in this section are specific to a single layer. We have used the subscript $k$ when defining each layer-specific flux rate; $k$ can be either the index of a water column layer, or $sice$ to indicate the skeletal ice layer. This notation distinguishes between rates that are specific to a single layer, and that use volumetric units ($\text{mg C m}^{-3}\,\text{d}^{-1}$) versus those that exchange material between layers and that are expressed in total flux across a boundary ($\text{mg C m}^{-2}\,\text{d}^{-1}$). To avoid clutter, we have chosen not to apply these subscripts to any remaining layer-dependent variables (such as state variable concentrations), but these are to be assumed for all pelagic and ice layer components.





### A3.1 Light attenuation in water

The model assumes that radiation is attenuated with depth as follows:

$$I_z = f_{PAR} \cdot I_0 \cdot \exp(-K_{PAR} \cdot (\zeta - z)) \tag{A13}$$

where $f_{PAR}$ is the fraction of surface light that is photosynthetically available, $I_0$ is the surface irradiance, and $K_{PAR}$ is
the light attenuation coefficient for photosynthetically active radiation (i.e. 400-700 nm). Incoming radiation is supplied by the
physical model and converted to photon flux.

The attenuation coefficient is itself the sum of attenuation from clear water, chlorophyll, and other sediment and organic
material:

$$K_{PAR} = K_w + K_A \cdot \left( \frac{[\text{PhL}]}{ccr_L} + \frac{[\text{PhS}]}{ccr_S} \right)^{K_B} + K_C + K_{D1} \cdot h^{K_{D2}} \tag{A14}$$

The first two terms in Equation A14 derive from Morel (1988)'s analysis of light attenuation in Case I waters. The final terms
(i.e. the $K_D$ portion plus $K_C$ constant) add additional attenuation based on the depth of the water column; this approximates
the assumption that sediment and organic material is higher near the coastline than in open water. The power law formula was
chosen based on a fit to satellite-derived inherent optical properties across the Bering Sea domain (see Appendix B for further
details).

The $K_{PAR}$ parameter is also used to calculate shortwave radiation decay in the physical model. By default, ROMS uses
the equations of Paulson and Simpson (1977), which considers the differing attenuation length scales for blue-green light
versus shorter- and longer-length wavelengths that are primarily attenuated in the upper $5\,\text{m}$ to $10\,\text{m}$ of the water column.
When coupling to the BESTNPZ model, we modify this equation to substitute our custom PAR attenuation length scale for the
blue-green portion of the spectrum:

$$f_{swdk} = (1 - a_{frac}) \cdot \exp(-K_{PAR} \cdot (\zeta - z)) + a_{frac} \cdot \exp\left( \frac{-(\zeta - z)}{a_{\mu 1}} \right) \tag{A15}$$

The values of $a_{frac}$ and $a_{\mu 1}$ correspond to $R$ and $\zeta_2$ in Paulson and Simpson (1977), with our $K_{PAR}$ replacing Paulson and
Simpson (1977)'s $\zeta_1$; we use the parameter values for Case I waters.

Table A5: Notation and description for input parameters related to light attenuation.

| Variable | Description | Units | Relevant input parameter(s) | | |
|---|---|---|---|---|---|
| | | | Group | Parameter | Value |
| $f_{PAR}$ | PAR fraction (fraction of shortwave that's in the 400-700 nm band) | –– | | PARfrac | 0.42 |





| $ccr_S$ | C:chl ratio | $\mathrm{mg\,C\,(mg\,chla)^{-1}}$ | PhS | ccr | 65 |
| $ccr_L$ | C:chl ratio | $\mathrm{mg\,C\,(mg\,chla)^{-1}}$ | PhL | ccrPhL | 25 |
| $a_{frac}$ | a unitless coefficient that determines switch between deep water and shallow water attenuation | —— | | a_frac hardcoded parameter | 0.58 |
| $a_{\mu 1}$ | attenuation length scale for deeper water | $\mathrm{m^{-1}}$ | | a_mu1 hardcoded parameter | 0.35 |
| $K_W$ | attenuation coefficient for clear water | $\mathrm{m^{-1}}$ | | k_ext | 0.034 |
| $K_A$ | factor, attenuation coefficient for chlorophyll | $\mathrm{m^{-1}}$ | | k_chlA | 0.0518 |
| $K_B$ | exponent, attenuation coefficient for chlorophyll | —— | | k_chlB | 0.428 |
| $K_C$ | attenuation coefficient for other material (CDOM, sediment, etc.) | $\mathrm{m^{-1}}$ | | k_chlC | 0.0363 |
| $K_{D1}$ | factor, depth-based attenuation coefficient | $\mathrm{m^{-1}}$ | | k_sed1 | 2.833 |
| $K_{D2}$ | exponent, depth-based attenuation coefficient | —— | | k_sed2 | $-1.079$ |

**A3.2 Gross primary production**

Primary production for both small and large phytoplankton is governed by the same set of equations. The maximum photosynthetic growth rate per unit chlorophyll $(\mathrm{mg\,C\,(mg\,chla)^{-1}\,d^{-1}})$ is a function of temperature, and defined in terms of each group's doubling rate $D_i$ and doubling rate exponent $D_p$ (Frost, 1987). The maximum uptake rate is calculated in both carbon-specific and chlorophyll-specific units:

$$P_{max} = \left(2^{\left(D_i \cdot 10^{(D_p \cdot T)}\right)} - 1\right) \tag{A16}$$

$$P_{max}^* = P_{max} \cdot ccr \tag{A17}$$

This rate is moderated by light and nutrient limitation. Light limitation uses a hyperbolic tangent function, after Jassby and Platt (1976),

$$Lim_I = \tanh\left(\frac{\alpha \cdot I_z}{P_{max}^*}\right) \tag{A18}$$

Nutrient limitation is based on the availability of nitrate, iron, and ammonium. Nitrate and ammonium limitation terms follow Frost and Franzen (1992), with nitrate uptake inhibited by ammonium when the latter is high relative to its half-saturation parameter:

$$Lim_{NO3} = \frac{[\mathrm{NO3}]}{(k_1 + [\mathrm{NO3}])\left(1 + \frac{[\mathrm{NH4}]}{k_2}\right)} \tag{A19}$$

$$Lim_{NH4} = \frac{[\mathrm{NH4}]}{k_2 + [\mathrm{NH4}]} \tag{A20}$$




Iron limitation follows a similar Michaelis-Menton form, but with an additional term to enforce saturation at a critical threshold value:

$$Lim_{Fe} = \min\left(1.0, \epsilon + \frac{[\text{Fe}]}{k_{Fe} + [\text{Fe}]} \cdot \frac{k_{Fe} + Fe_{Crit}}{Fe_{Crit}}\right) \tag{A21}$$

Nitrate uptake is controlled by the mimimum limitation factor between light, nitrate, and iron, while ammonium uptake is
limited by light or ammonium:

$$Gpp(\text{NO3}, \text{X})_k = P_{max} \cdot [\text{X}] \cdot \min(Lim_{NO3}, Lim_{Fe}, Lim_I) \tag{A22}$$
$$Gpp(\text{NH4}, \text{X})_k = P_{max} \cdot [\text{X}] \cdot \min(Lim_{NH4}, Lim_I) \tag{A23}$$

Primary production also occurs in the ice layer when ice is present. In the ice layer, production is a function of light, nutrient limitation, brine salinity, and temperature, following Jin et al. (2006). Light limitation uses the following photosynthesis-
irradiance curve; unlike the pelagic production, this one includes strong photoinhibition at higher light levels:

$$Lim_{Iice} = \left(1 - \exp\left(-\alpha_{Ib} \cdot \frac{I_0 \cdot f_{PAR}}{c_I}\right)\right) \cdot \exp\left(-\beta_I \cdot \frac{I_0 \cdot f_{PAR}}{c_I}\right) \tag{A24}$$

where $\frac{I_0 \cdot f_{PAR}}{c_I}$ is the photosynthetically active radiation converted to $\text{W m}^{-2}$.

As in the water column, nitrate limitation uses Michaelis-Menten uptake dynamics with ammonium inhibition (with $f_r$ denoting the $f$-ratio between new (nitrate) and regenerated (ammonium) production):

$$Lim_{Nice} = \frac{[\text{IceNO3}]}{k_1 + [\text{IceNO3}]} \cdot \exp(-\psi \cdot [\text{IceNH4}]) + \frac{[\text{IceNH4}]}{k_2 + [\text{IceNH4}]} \tag{A25}$$

$$f_r = \frac{\frac{[\text{IceNO3}]}{k_1 + [\text{IceNO3}]} \cdot \exp(-\psi \cdot [\text{IceNH4}])}{Lim_{Nice}} \tag{A26}$$

Brine salinity ($S_b$) is not tracked explicitly by the ice model, so instead it is estimated based on a piecewise polynomial fit to ice temperature ($T_i$, tracked by the the ice model), following Arrigo et al. (1993):

$$S_b = c_0 + c_1 \cdot T_i + c_2 \cdot T_i^2 + c_3 \cdot T_i^3 \tag{A27}$$

|  | $c_0$ | $c_1$ | $c_2$ | $c_3$ |
|---|---|---|---|---|
| $T_i \geq$ -22.9 | -3.9921 | -22.7 | -1.0015 | -0.019956 |
| -44.0 < $T_i$ < -22.9 | 206.24 | -1.8907 | -0.060868 | -0.0010247 |
| $T_i \leq$ -44.0 | -4442.1 | -277.86 | -5.501 | -0.03669 |





The effect of salinity on ice algae growth rate is also a polynomial fit (Arrigo and Sullivan, 1992):

$$\xi_{sb} = 1.1 \times 10^{-2} + 3.012 \times 10^{-2} \cdot S_b$$
$$+ 1.0342 \times 10^{-3} \cdot S_b^2$$
$$- 4.6033 \times 10^{-5} \cdot S_b^3$$
$$+ 4.926 \times 10^{-7} \cdot S_b^4$$
$$- 1.659 \times 10^{-9} \cdot S_b^5 \tag{A28}$$

When running in climatological ice mode (`CLIM_ICE_1D`), where no explicit ice temperature is modeled, $\xi_{sb} = 1.0$.

The final primary production calculation is then:

$$Gpp(\text{IceNO3}, \text{IcePhL})_{ice} = \mu_0 \cdot \exp(0.0633 \cdot T_{k=N}) \cdot \xi_{sb} \cdot$$
$$\min(Lim_{Iice}, Lim_{Nice}) \cdot [\text{IcePhL}] \cdot f_r \tag{A29}$$

$$Gpp(\text{IceNH4}, \text{IcePhL})_{ice} = \mu_0 \cdot \exp(0.0633 \cdot T_{k=N}) \cdot \xi_{sb} \cdot$$
$$\min(Lim_{Iice}, Lim_{Nice}) \cdot [\text{IcePhL}] \cdot (1 - f_r) \tag{A30}$$

In this case, $T_{k=N}$ is the temperature of the top water layer, used to approximate the temperature of the ice itself.

Table A6: Notation and description for input parameters related to gross primary production.

| Variable | Description | Units | Relevant input parameter(s) | | |
|---|---|---|---|---|---|
| | | | Group | Parameter | Value |
| $\alpha$ | photosynthetic efficiency | $\mathrm{mg\,C\,(mg\,chla)^{-1}\,E^{-1}\,m^2}$ | PhS | alphaPhS | 5.6 |
| | | | PhL | alphaPhL | 2.2 |
| $K_1$ | half-saturation constant for nitrate uptake | $\mathrm{mmol\,N\,m^{-3}}$ | PhS | k1PhS | 1 |
| | | | PhL | k1PhL | 2 |
| | | | IcePhL | ksnut1 | 1 |
| $K_2$ | half-saturation constant for ammonium uptake | $\mathrm{mmol\,N\,m^{-3}}$ | PhS | k2PhS | 0.5 |
| | | | PhL | k2PhL | 2 |
| | | | IcePhL | ksnut2 | 4 |
| $\psi$ | ammonium inhibition constant | $\mathrm{m^3\,(mmol\,N)^{-1}}$ | IcePhL | inhib | 1.46 |
| $D_i$ | doubling rate parameter | $\mathrm{d^{-1}}$ | PhS | DiS | 0.5 |
| | | | PhL | DiL | 1 |
| $D_p$ | doubling rate parameter | $\mathrm{^\circ C^{-1}}$ | PhS | DpS | 0.0275 |
| | | | PhL | DpL | 0.0275 |
| $k_{Fe}$ | half-saturation constant for iron uptake | $\mathrm{\mu mol\,Fe\,m^{-3}}$ | PhS | kfePhS | 0.3 |
| | | | PhL | kfePhL | 1 |
| $Fe_{Crit}$ | iron concentration below which growth is limited | $\mathrm{\mu mol\,Fe\,m^{-3}}$ | PhS | FeCritPS | 2 |





| | | | | PhL | FeCritPL | 2 |
|---|---|---|---|---|---|---|
| $\alpha_{Ib}$ | photosynthetic efficiency/maximal | | W m$^{-2}$ | IcePhL | alphaIb | 0.08 |
| | photosynthetic rate | | | | | |
| $\beta_I$ | light inhibition/maximal photosynthetic rate | | W m$^{-2}$ | IcePhL | betaI | 0.018 |
| $\mu_0$ | maximum growth rate at 0 deg C | | d$^{-1}$ | IcePhL | mu0 | 2.4 |

### A3.3 Grazing and predation

Pelagic grazing and predation fluxes are a function of a grazer's or predator's maximum ingestion rate ($e_Y$), its total prey availability, prey-specific feeding preferences ($fp_{XY}$), and the water temperature, using the multiple resource Holling Type 3 functional response of Ryabchenko et al. (1997):

$$Gra(\mathrm{X},\mathrm{Y})_k = Q_Y^{\left(\frac{T-Q_{TY}}{10}\right)} \cdot e_Y \cdot [\mathrm{Y}] \cdot \frac{fp_{XY} \cdot [\mathrm{X}]^2}{f_Y + \sum_Z \left(fp_{ZY} \cdot [\mathrm{Z}]^2\right)} \tag{A31}$$

where Y refers to the predator group, X is a specific prey group, and Z refers to the set of all prey groups of that predator. Note that some of the pelagic groups can graze on ice algae; when preyed upon, the ice algae concentration is adjusted as though it were located in the surface layer of the water:

$$[\mathrm{IcePhL}]_k = \begin{cases} [\mathrm{IcePhL}]_{ice} \cdot \frac{h_{sice}}{h_k}, & k = N \\ 0, & \text{otherwise} \end{cases} \tag{A32}$$

The maximum ingestion rates, $e_Y$, are constant for all groups except large-bodied copepods (NCaS and NCaO). These groups can be parameterized to perform seasonal diapause, and during periods of downward migration, their ingestion rates are dropped to $e_Y = 0\,\mathrm{d}^{-1}$. See subsection A4 for a description of the diapause time-of-year calculation.

Benthic processes in BESTNPZ are based on a greatly-simplified version of the European Regional Seas Ecosystem model (ERSEM) (Ebenhöh et al., 1995). Benthic infauna graze on pelagic detritus and phytoplankton located within a certain distance of the bottom (currently hard-coded to $dw = 1.0$m). The feeding fluxes are defined as follows:

$$[\mathrm{X}]_{ben} = \int_{-h}^{-h+dw} [\mathrm{X}]dz \tag{A33}$$

$$F_X = \frac{(fp_{XB} \cdot [\mathrm{X}]_{ben})^2}{fp_{XB} \cdot [\mathrm{X}]_{ben} + L_P} \tag{A34}$$

$$Gra(\mathrm{X},\mathrm{Ben}) = Q_B^{\left(\frac{T-Q_{TB}}{10}\right)} \cdot e_{Ben} \cdot [\mathrm{Ben}] \cdot \frac{F_X}{\sum_Z F_Z + f_{PB}} \tag{A35}$$





As in the pelagic grazing equation, X refers to a single pelagic prey group (PhS, PhL, Det, or DetF), and Z refers to the full set of these four groups. A weight factor, $w_{k,X}$ is calculated to distribute these losses proportionately across the water column (see subsubsection A3.9):

$$w_{k,X} = \frac{\int\limits_{z_{lo,k}}^{\min(z_{hi,k}, -h+dw)} [\mathrm{X}]dz}{[\mathrm{X}]_{ben}} \tag{A36}$$

5   where $z_{lo,k}$ and $z_{hi,k}$ are the lower and upper depth limits of layer $k$.

Benthic infauna also graze on benthic detritus, following the same equation but with different parameters for prey threshold and half-saturation values:

$$F_X = \frac{(fp_{XB} \cdot [\mathrm{X}])^2}{fp_{XB} \cdot [\mathrm{X}] + L_D} \tag{A37}$$

10   $$Gra(\mathrm{X}, \mathrm{Ben}) = Q_B^{\left(\frac{T - Q_{TB}}{10}\right)} \cdot e_{Ben} \cdot [\mathrm{Ben}] \cdot \frac{F_X}{F_X + f_{DB}} \tag{A38}$$

Here, X refers to a single prey group, DetBen.

Note that the water column grazing fluxes are in units of $\mathrm{mg\,C\,m^{-3}\,d^{-1}}$ while the benthic feeding fluxes are in $\mathrm{mg\,C\,m^{-2}\,d^{-1}}$.

Table A7: Notation and description for input parameters related to grazing and predation.

| Variable | Description | Units | Relevant input parameter(s) | | |
| --- | --- | --- | --- | --- | --- |
| | | | Group | Parameter | Value |
| $fp_{XY}$ | grazing preference of predator Y on prey X | $--$ | PhS → MZL | fpPhSMZL | 1 |
| | | | PhL → MZL | fpPhLMZL | 0.2 |
| | | | PhS → Cop | fpPhSCop | 0.8 |
| | | | PhL → Cop | fpPhLCop | 0.7 |
| | | | MZL → Cop | fpMZLCop | 0.5 |
| | | | IcePhL → Cop | fpPhLCop | 0.7 |
| | | | PhS → NCaS | fpPhSNCa | 0.1 |
| | | | PhL → NCaS | fpPhLNCa | 1 |
| | | | MZL → NCaS | fpMZLNCa | 1 |
| | | | IcePhL → NCaS | fpPhLNCa | 1 |
| | | | PhS → NCaO | fpPhSNCa | 0.1 |
| | | | PhL → NCaO | fpPhLNCa | 1 |
| | | | MZL → NCaO | fpMZLNCa | 1 |
| | | | IcePhL → NCaO | fpPhLNCa | 1 |
| | | | PhS → EupS | fpPhSEup | 1 |
| | | | PhL → EupS | fpPhLEup | 1 |





| | | | | | |
|---|---|---|---|---|---|
| | | | MZL → EupS | fpMZLEup | 1 |
| | | | Cop → EupS | fpCopEup | 0.2 |
| | | | Det → EupS | fpDetEup | 0.4 |
| | | | DetF → EupS | fpDetEup | 0.4 |
| | | | IcePhL → EupS | fpPhLEup | 1 |
| | | | PhS → EupO | fpPhSEup | 1 |
| | | | PhL → EupO | fpPhLEup | 1 |
| | | | MZL → EupO | fpMZLEup | 1 |
| | | | Cop → EupO | fpCopEup | 0.2 |
| | | | Det → EupO | fpDetEupO | 0 |
| | | | DetF → EupO | fpDetEupO | 0 |
| | | | IcePhL → EupO | fpPhLEup | 1 |
| | | | Cop → Jel | fpCopJel | 1 |
| | | | NCaS → Jel | fpNCaJel | 1 |
| | | | EupS → Jel | fpEupJel | 1 |
| | | | NCaO → Jel | fpNCaJel | 1 |
| | | | EupO → Jel | fpEupJel | 1 |
| | | | PhS → Ben | prefPS | 0.1 |
| | | | PhL → Ben | prefPL | 1 |
| | | | Det → Ben | prefD | 1 |
| | | | DetF → Ben | prefD | 1 |
| | | | DetBen → Ben | prefD | 1 |
| $e_Y$ | maximum specific ingestion rate | mg Cprey (mg Cpred)$^{-1}$ d$^{-1}$ | MZL | eMZL | 0.4 |
| | | | Cop | eCop | 0.4 |
| | | | NCaS | eNCa | 0.3 |
| | | | NCaO | eNCa | 0.3 |
| | | | EupS | eEup | 0.3 |
| | | | EupO | eEup | 0.3 |
| | | | Jel | eJel | 0.069 |
| | | | Ben | Rup | 0.05 |
| $f_Y$ | half-saturation constant for grazing | mg C m$^{-3}$ | MZL | fMZL | 20 |
| | | | Cop | fCop | 30 |
| | | | NCaS | fNCa | 30 |
| | | | NCaO | fNCa | 30 |
| | | | EupS | fEup | 40 |
| | | | EupO | fEup | 40 |
| | | mg C m$^{-2}$ | Ben (pelagic food) | KupP | 10 |
| | | | Ben (benthic food) | KupD | 2000 |
| $Q_Y$ | Q10 (rate change factor per 10 deg) for growth rate | – – | MZL | Q10MLZ | 2 |
| | | | Cop | Q10Cop | 1.7 |
| | | | NCaS | Q10NCa | 1.6 |
| | | | NCaO | Q10NCa | 1.6 |
| | | | EupS | Q10Eup | 1.5 |





| Variable | Description | Units | Group | Parameter | Value |
|---|---|---|---|---|---|
| | | | EupO | Q10Eup | 1.5 |
| | | | Jel | Q10Jele | 2.4 |
| | | | Ben | q10r | 1.5 |
| $Q_{TY}$ | reference temperature for growth rate | °C | MZL | Q10MZLT | 5 |
| | | | Cop | Q10CopT | 5 |
| | | | NCaS | Q10NCaT | 5 |
| | | | NCaO | Q10NCaT | 5 |
| | | | EupS | Q10EupT | 5 |
| | | | EupO | Q10EupT | 5 |
| | | | Jel | Q10JelTe | 10 |
| | | | Ben | T0benr | 5 |
| $L_P$ | threshold for benthos grazing | mg C m$^{-2}$ | Ben (pelagic food) | LupP | 1 |
| $L_D$ | | | Ben (benthic food) | LupD | 292 |

## A3.4 Egestion and excretion

Egestion fluxes associated with grazing and predation in the water column are a simple fraction of total prey eaten:

$$Ege(\mathrm{Y}, \mathrm{Det[F]})_k = (1 - \gamma_Y) \cdot \sum_Z Gra(Z, Y)_k \tag{A39}$$

Egestion fluxes from the microzooplankton group (MZL) go to the slow-sinking detrital pool (Det); all other egestion fluxes

5  go to the fast-sinking detrital pool (DetF).

Infauna egestion and excretion is a bit more complex; it is proportional to the prey eaten, with differing rates for detrital vs phytoplankton prey. The flux is split evenly, with half going to benthic detritus (DetBen) and half to NH4.

$$Exc(\mathrm{Ben}, \mathrm{DetBen}) = 0.5 \cdot \left( ex_D \cdot \sum_{X=\mathrm{det}} Gra(\mathrm{X}, \mathrm{Ben}) + ex_P \cdot \sum_{X=\mathrm{phyto}} Gra(\mathrm{X}, \mathrm{Ben}) \right) \tag{A40}$$

$$Exc(\mathrm{Ben}, \mathrm{NH4}) = 0.5 \cdot \left( ex_D \cdot \sum_{X=\mathrm{det}} Gra(\mathrm{X}, \mathrm{Ben}) + ex_P \cdot \sum_{X=\mathrm{phyto}} Gra(\mathrm{X}, \mathrm{Ben}) \right) \tag{A41}$$

10  As with benthic grazing, these benthic excretion fluxes are in units of mg C m$^{-2}$ d$^{-1}$. The flux to NH4 is assumed to return to the bottom water column layer, and is converted to a volumetric flux based on the thickness of that layer (see subsubsection A3.9).

Table A8: Notation and description for input parameters related to egestion and excretion.

| Variable | Description | Units | Relevant input parameter(s) | | |
|---|---|---|---|---|---|
| | | | Group | Parameter | Value |




| $\gamma_Y$ | growth efficiency | $--$ | MZL | gammaMZL | 0.7 |
|---|---|---|---|---|---|
| | | | Cop | gammaCop | 0.7 |
| | | | NCaS | gammaNCa | 0.7 |
| | | | NCaO | gammaNCa | 0.7 |
| | | | EupS (live prey) | gammaEup | 0.7 |
| | | | EupS (detrital prey) | hardcoded | 0.3 |
| | | | EupO (live prey) | gammaEup | 0.7 |
| | | | EupO (detrital prey) | hardcoded | 0.3 |
| | | | Jel | gammaJel | 1 |
| $ex_P$ | excretion fraction (1 - growth efficiency) | $--$ | Ben (living prey) | eex | 0.3 |
| $ex_D$ | | | Ben (detrital prey) | eexD | 0.5 |

### A3.5 Respiration

All pelagic producers and consumers except jellyfish respire following the temperature-dependent formulation of Arhonditsis and Brett (2005). The phytoplankton and microzooplankton groups maintain a constant basal metabolic rate ($bm$):

$$Res(\mathrm{X}, \mathrm{NH4})_k = \exp(k_{tb} \cdot (T - T_{ref})) \cdot bm \cdot [\mathrm{X}] \tag{A42}$$

5 The larger zooplankton groups substitute a basal metabolic rate that includes a starvation response when prey is scarce:

$$Res(\mathrm{X}, \mathrm{NH4})_k = \exp(k_{tb} \cdot (T - T_{ref})) \cdot B_{met} \cdot [\mathrm{X}] \tag{A43}$$

where

$$B_{met} = \begin{cases} bm \cdot \left( \dfrac{\sum\limits_{Z}\left(fp_{ZY} \cdot [\mathrm{Z}]^2\right)}{0.01} \right) & \sum\limits_{Z}\left(fp_{ZY} \cdot [\mathrm{Z}]^2\right) < 0.01 \\ bm & \text{otherwise} \end{cases} \tag{A44}$$

(The summation relates to the total available prey; see subsubsection A3.3 for details.)

10 The large copepod groups (NCaS and NCaO) also include a diapause adjustment, such that their basal metabolic rate $bm$ is reduced to 10% of the $bm$ parameter value during periods of downward migration. See subsection A4 for details of the time-of-year calculation for diapause.

Jellyfish respiration also follows a temperature-dependent formula, after Uye and Shimauchi (2005):

$$Res(\mathrm{Jel}, \mathrm{NH4})_k = Q_r^{\left(\frac{T - Q_{Tr}}{10}\right)} \cdot bm \cdot [\mathrm{Jel}] \tag{A45}$$

15 Infaunal respiration includes terms for both basal metabolism and active metabolism proportional to grazing:





$$Res(\text{Ben}, \text{NH4}) = Q_B^{\left(\frac{T-Q_{TB}}{10}\right)} \cdot bm \cdot [\text{Ben}] +$$
$$am \cdot \left( (1 - ex_D) \cdot \sum_{X=\text{det}} Gra(X, \text{Ben}) + \right.$$
$$\left. (1 - ex_P) \cdot \sum_{X=\text{phyto}} Gra(X, \text{Ben}) \right) \tag{A46}$$

Finally, ice algae respiration uses a metabolic rate linearly proportional to its maximum growth rate, after Jin et al. (2006):

$$5 \quad Res(\text{IcePhl}, \text{IceNH4})_{ice} = r \cdot \mu_0 \cdot \exp(0.0633 \cdot T_{k=N}) \cdot [\text{IcePhL}] \tag{A47}$$

As with ice algae production, the surface water temperature is used as a proxy for ice temperature when calculating the temperature component of this rate.

Table A9: Notation and description for input parameters related to respiration. (See Table A7 for pelgagic prey preferences and infauna Q-10 parameters, Table A8 for infauna excretion fraction parameters, and Table A6 for ice algae growth rate parameter.)

| Variable | Description | Units | Group | Parameter | Value |
|---|---|---|---|---|---|
| $bm$ | basal metabolic rate | d$^{-1}$ | PhS | respPhS | 0.02 |
| | | | PhL | respPhL | 0.02 |
| | | | MZL | respMZL | 0.08 |
| | | | Cop | respCop | 0.04 |
| | | | NCaS | respNCa | 0.03 |
| | | | NCaO | respNCa | 0.03 |
| | | | EupS | respEup | 0.02 |
| | | | EupO | respEup | 0.02 |
| | | | Jel | respJel | 0.02 |
| | | | Ben | Rres | 0.0027 |
| $am$ | active metabolic rate | d$^{-1}$ | Ben | Qres | 0.25 |
| $k_{tb}$ | temperature coefficient for respiration | °C$^{-1}$ | PhS | KtBm_PhS | 0.03 |
| | | | PhL | KtBm_PhL | 0.03 |
| | | | MZL | KtBm_MZL | 0.069 |
| | | | Cop | ktbmC | 0.05 |
| | | | NCaS | ktbmN | 0.05 |
| | | | NCaO | ktbmN | 0.05 |
| | | | EupS | ktbmE | 0.069 |
| | | | EupO | ktbmE | 0.069 |
| $T_{ref}$ | reference temperature for respiration | °C | PhS | TmaxPhS | 10 |





| | | | | | |
|---|---|---|---|---|---|
| | | | PhL | TmaxPhL | 10 |
| | | | MZL | TmaxMZL | 8 |
| | | | Cop | TrefC | 15 |
| | | | NCaS | TrefN | 5 |
| | | | NCaO | TrefN | 5 |
| | | | EupS | TrefE | 5 |
| | | | EupO | TrefE | 5 |
| $Q_r$ | Q10 for respiration rate | −− | Jel | Q10Jelr | 2.8 |
| $Q_{Tr}$ | reference temperature for Q10 respiration | °C | Jel | Q10JelTr | 10 |
| $r$ | respiration rate as a fraction of maximum growth rate | −− | IcePhL | R0i | 0.05 |

## A3.6 Mortality and senescence

Non-predatory mortality losses for phytoplankton groups are formulated as a linear closure term:

$$Mor(\text{X}, \text{Det})_k = m_L \cdot [\text{X}]\tag{A48}$$

Microzooplankton losses have the option of following either a linear closure as above (`MZLM0LIN` flag defined), or a
quadratic closure:

$$Mor(\text{X}, \text{Det})_k = m_Q \cdot [\text{X}]^2\tag{A49}$$

Note that when switching between the linear and quadratic formulations, the relevant input parameter for the `MZL` group
switches between `mMZL` and `mpredMZL`.

All larger zooplankton groups use a temperature-mediated quadratic closure term:

$$Mor(\text{X}, \text{DetF})_k = Q_Y^{\left(\frac{T - Q_{TY}}{10}\right)} \cdot m_Q \cdot [\text{X}]^2\tag{A50}$$

Non-predatory mortality fluxes from the phytoplankton and microzooplankton groups go to the slow-sinking detritus, while
all other non-predatory mortality losses go to the fast-sinking detritus.

The benthic infauna group includes both a linear and quadratic mortality function, the former to represent senescence and
the latter as a predation closure term.

$$Mor(\text{Ben}, \text{DetBen}) = Q_B^{\left(\frac{T - Q_{TB}}{10}\right)} \cdot \left(m_L \cdot [\text{Ben}] + m_Q \cdot [\text{Ben}]^2\right)\tag{A51}$$

Finally, ice algae use a linear mortality rate with a temperature dependence, following Jin et al. (2006):

$$Mor(\text{IcePhl}, \text{IceNH4})_{ice} = \exp(rg \cdot T_{k=N}) \cdot m_{L0} \cdot [\text{IcePhL}]\tag{A52}$$




Table A10: Notation and description for input parameters related to non-predatory mortality. (See Table A7 for Q-10 parameters.)

| Variable | Description | Units | Relevant input parameter(s) | | |
|---|---|---|---|---|---|
| | | | Group | Parameter | Value |
| $m_L$ | linear mortality rate | $d^{-1}$ | PhS | mPhS | 0.01 |
| | | | PhL | mPhL | 0.01 |
| | | | Ben | rmort | 0.0021 |
| $m_Q$ | quadratic mortality rate | $(mg\,C)^{-1}\,d^{-1}$ | MZL | mpredMZL | 0.01 |
| | | | Cop | mpredCop | 0.05 |
| | | | NCaS | mpredNCa | 0.05 |
| | | | NCaO | mpredNCa | 0.05 |
| | | | EupS | mpredEup | 0.05 |
| | | | EupO | mpredEup | 0.05 |
| | | | Jel | mpredJel | 0.006 |
| | | | Ben | BenPred | $1 \times 10^{-6}$ |
| $m_{L0}$ | mortality rate at 0 deg C | $d^{-1}$ | IcePhL | rg0 | 0.01 |
| $r_g$ | temperature coefficient for mortality | $°C^{-1}$ | IcePhL | rg | 0.03 |

### A3.7 Remineralization and nitrification

Detrital remineralization is proportional to temperature and the nitrogen content of detritus, after Kawamiya et al. (2000):

$$Rem(\mathrm{X}, \mathrm{NH4})_k = (P_{v0} \cdot \exp(P_{vT} \cdot T) \cdot [\mathrm{X}] \cdot \xi)/\xi \tag{A53}$$

5  The conversion from nitrogen content back to carbon content is done to maintain unit consistency with the other between-group fluxes, using the assumption that all living and detrital groups maintain identical C:N:Fe stoichiometry. Both fast- and slow-sinking detritus use the same parameters for this process.

Nitrification rate in the water column is also influenced by temperature (Arhonditsis and Brett, 2005):

$$Nit(\mathrm{NH4}, \mathrm{NO3})_k = \left( n_0 \cdot \exp(-k_{tntr} \cdot (T - T_{opt})^2) \cdot [\mathrm{NH4}] \cdot \frac{[\mathrm{NH4}]}{k_{Nit} + [\mathrm{NH4}]} \right)/\xi \tag{A54}$$

10  Nitrification in the ice is a simple linear function of ammonium concentration (Jin et al., 2006):

$$Nit(\mathrm{IceNH4}, \mathrm{IceNO3})_{ice} = (N_{nit} \cdot [\mathrm{IceNH4}])/\xi \tag{A55}$$

As with remineralization, the final nitrification rate values are converted to carbon units simply for bookkeeping purposes; they will be converted back to nitrogen units when used in the final rate of change equations (see subsubsection A3.9.)



Table A11: Notation and description for input parameters related to remineralization and nitrification.

| Variable | Description | Units | Relevant input parameter(s) | | |
| --- | --- | --- | --- | --- | --- |
| | | | Group | Parameter | Value |
| $P_{v0}$ | PON remineralization rate at 0 deg C | $\mathrm{d}^{-1}$ | | Pv0 | 0 |
| $P_{vT}$ | temperature coefficient for remineralization | $\mathrm{°C}^{-1}$ | | PvT | 0.069 |
| $n_0$ | nitrification rate at 0 deg C | $\mathrm{d}^{-1}$ | | Nitr0 | 0.0107 |
| $k_{tntr}$ | temperature coefficient for nitrification | $\mathrm{°C}^{-1}$ | | ktntr | 0.002 |
| $T_{opt}$ | optimal temperature for nitrification | $\mathrm{°C}$ | | ToptNit | 20 |
| $k_{Nit}$ | half-saturation constant for nitrification | $\mathrm{mmol\,N\,m}^{-3}$ | | KNH4Nit | 0.057 |
| $N_{Nit}$ | ice nitrification rate | $\mathrm{d}^{-1}$ | | annit | 0.0149 |

## A3.8 Ice interface convective exchange

As an ice sheet grows, dense brine is released from the skeletal ice layer at its base and replaced with seawater; this results in a convective exchange of water and nutrients between the ice and surface water. The BESTNPZ model follows Jin et al. (2006) and sets the rate of this exchange using a polynomial function of ice growth rate:

$$
5 \quad T_{wi} = \begin{cases} 720 \cdot 86400 \cdot \left( 4.9 \times 10^{-6} \cdot \left( -\frac{dH}{dt} \right) - 1.39 \times 10^{-5} \cdot \left( -\frac{dH}{dt} \right)^2 \right) & \frac{dH}{dt} \leq 0 \\ 72 \cdot 86400 \cdot \left( 9.667 \times 10^{-11} + 4.49 \times 10^{-6} \cdot \left( \frac{dH}{dt} \right) - 1.39 \times 10^{-5} \cdot \left( \frac{dH}{dt} \right)^2 \right) & \frac{dH}{dt} > 0 \end{cases} \tag{A56}
$$

where $\frac{dH}{dt}$ is the rate of change of ice thickness $(\mathrm{m\,s}^{-1})$ between the current time step and the previous one. The resulting exchange rate, $T_{wi}$, is expressed in $\mathrm{m\,d}^{-1}$.

The exchange in nutrients then becomes a function of the difference in concentrations in the surface layer of water versus the ice layer. Phytoplankton can be washed out of the skeletal ice layer but not in, so the exchange of ice algae and large phytoplankton assumes a concentration of 0 in the surface water:

$$
Twi(\mathrm{IceNO3}, \mathrm{NO3}) = T_{wi} \cdot ([\mathrm{IceNO3}] - [\mathrm{NO3}])/\xi \tag{A57}
$$

$$
Twi(\mathrm{IceNH4}, \mathrm{NH4}) = T_{wi} \cdot ([\mathrm{IceNH4}] - [\mathrm{NO3}])/\xi \tag{A58}
$$

$$
Twi(\mathrm{IcePhL}, \mathrm{PhL}) = T_{wi} \cdot [\mathrm{IcePhL}] \tag{A59}
$$

This equation results in a rate of exchange of material across the boundary $(\mathrm{mgC/m^2/d})$ that can have either a positive value (net movement from ice to water) or a negative value (net movement from water to ice). As in previous sections, the nutrient transport is converted to carbon units here purely for bookkeeping purposes.





### A3.9 Total rate of change

The total rate of change for each state variable due to source-minus-sink processes is calculated as a sum of the rates detailed in the previous sections (subsubsection A3.2 – subsubsection A3.8). Recall that in the previous sections, all flux rates taking place within the pelagic water column layers, or within the ice layer, were expressed in volumetric units ($\mathrm{mg\,C\,m^{-3}\,d^{-1}}$), while all

processes in the benthos or across the water-ice or water-benthos boundaries were expressed in per-area units ($\mathrm{mg\,C\,m^{-2}\,d^{-1}}$). Terms displayed in blue apply only in the top layer ($k = N$), while terms displayed in brown apply only to the bottom layer ($k = 1$).

$$\frac{d}{dt}\mathrm{NO3}_k = \left( Nit(\mathrm{NH4},\mathrm{NO3})_k - \sum_{\substack{X\in(PhS,\\PhL)}} Gpp(\mathrm{NO3},\mathrm{X})_k + \frac{Twi(\mathrm{IceNO3},\mathrm{NO3})}{h_k} \right) \cdot \xi \tag{A60}$$

$$\frac{d}{dt}\mathrm{NH4}_k = \left( \sum_{\substack{X\in(PhS,\\PhL,MZL,\\Cop,NCaS,\\NCaO,EupS,\\EupO,Jel)}} Res(\mathrm{X},\mathrm{NH4})_k + \sum_{\substack{X\in(Det,\\DetF)}} Rem(\mathrm{X},\mathrm{NH4})_k - \sum_{\substack{X\in(PhS,\\PhL)}} Gpp(\mathrm{NH4},\mathrm{X})_k \right.$$

$$\left. - Nit(\mathrm{NH4},\mathrm{NO3})_k + \frac{Exc(\mathrm{Ben},\mathrm{NH4})_k}{h_k} + \frac{Res(\mathrm{Ben},\mathrm{NH4})_k}{h_k} + \frac{Twi(\mathrm{IceNH4},\mathrm{NH4})_k}{h_k} \right) \cdot \xi \tag{A61}$$

$$\frac{d}{dt}\mathrm{PhS}_k = \sum_{\substack{X\in(NO3,\\NH4)}} Gpp(\mathrm{X},\mathrm{PhS})_k - \sum_{\substack{X\in(MZL,\\Cop,NCaS,\\NCaO,EupS,\\EupO)}} Gra(\mathrm{PhS},\mathrm{X})_k - Mor(\mathrm{PhS},\mathrm{Det})_k - Res(\mathrm{PhS},\mathrm{NH4})_k$$

$$- \frac{Gra(\mathrm{PhS},\mathrm{Ben}) \cdot w_{k,PhS}}{h_k} \tag{A62}$$

$$\frac{d}{dt}\mathrm{PhL}_k = \sum_{\substack{X\in(NO3,\\NH4)}} Gpp(\mathrm{X},\mathrm{PhL})_k - \sum_{\substack{X\in(MZL,\\Cop,NCaS,\\NCaO,EupS,\\EupO)}} Gra(\mathrm{PhL},\mathrm{X})_k - Mor(\mathrm{PhL},\mathrm{Det})_k - Res(\mathrm{PhL},\mathrm{NH4})_k$$

$$- \frac{Gra(\mathrm{PhL},\mathrm{Ben}) \cdot w_{k,PhL}}{h_k} + \frac{Twi(\mathrm{IcePhL},\mathrm{PhL})}{h_k} \tag{A63}$$

$$\frac{d}{dt}\mathrm{MZL}_k = \sum_{\substack{X\in(PhS,\\PhL)}} Gra(\mathrm{X},\mathrm{MZL})_k - Ege(\mathrm{MZL},\mathrm{Det})_k - \sum_{\substack{X\in(Cop,\\NCaS,NCaO,\\EupS,EupO)}} Gra(\mathrm{MZL},\mathrm{X})_k$$

$$- Mor(\mathrm{MZL},\mathrm{Det})_k - Res(\mathrm{MZL},\mathrm{NH4})_k \tag{A64}$$



$$\frac{d}{dt}\text{Cop}_k = \sum_{\substack{X \in (PhS, \\ PhL, MZL, \\ IcePhL)}} Gra(\text{X}, \text{Cop})_k - Ege(\text{Cop}, \text{DetF})_k - \sum_{\substack{X \in (EupS, \\ EupO, Jel)}} Gra(\text{Cop}, \text{X})_k$$

$$- Mor(\text{Cop}, \text{DetF})_k - Res(\text{Cop}, \text{NH4})_k \tag{A65}$$

$$\frac{d}{dt}\text{NCaS}_k = \sum_{\substack{X \in (PhS, \\ PhL, MZL, \\ IcePhL)}} Gra(\text{X}, \text{NCaS})_k - Ege(\text{NCaS}, \text{DetF})_k - Gra(\text{NCaS}, \text{Jel})_k$$

$$- Mor(\text{NCaS}, \text{DetF})_k - Res(\text{NCaS}, \text{NH4})_k \tag{A66}$$

$$\frac{d}{dt}\text{EupS}_k = \sum_{\substack{X \in (PhS, \\ PhL, MZL, \\ Cop, IcePhL)}} Gra(\text{X}, \text{EupS})_k - Ege(\text{EupS}, \text{DetF})_k - Gra(\text{EupS}, \text{Jel})_k$$

$$- Mor(\text{EupS}, \text{DetF})_k - Res(\text{EupS}, \text{NH4})_k \tag{A67}$$

$$\frac{d}{dt}\text{NCaO}_k = \sum_{\substack{X \in (PhS, \\ PhL, MZL, \\ IcePhL)}} Gra(\text{X}, \text{NCaO})_k - Ege(\text{NCaO}, \text{DetF})_k - Gra(\text{NCaO}, \text{Jel})_k$$

$$- Mor(\text{NCaO}, \text{DetF})_k - Res(\text{NCaO}, \text{NH4})_k \tag{A68}$$

$$\frac{d}{dt}\text{EupO}_k = \sum_{\substack{X \in (PhS, \\ PhL, MZL, \\ Cop, IcePhL)}} Gra(\text{X}, \text{EupO})_k - Ege(\text{EupO}, \text{DetF})_k - Gra(\text{EupO}, \text{Jel})_k$$

$$- Mor(\text{EupO}, \text{DetF})_k - Res(\text{EupO}, \text{NH4})_k \tag{A69}$$

$$\frac{d}{dt}\text{Det}_k = Ege(\text{MZL}, \text{Det})_k + \sum_{\substack{X \in (PhS, \\ PhL, MZL)}} Mor(\text{X}, \text{Det})_k - \sum_{\substack{X \in (EupS, \\ EupO)}} Gra(\text{Det}, \text{X})_k - Rem(\text{Det}, \text{NH4})_k$$

$$- \frac{Gra(\text{Det}, \text{Ben}) \cdot w_{k, Det}}{h_k} \tag{A70}$$

$$\frac{d}{dt}\text{DetF}_k = \sum_{\substack{X \in (Cop, \\ NCaS, NCaO, \\ EupS, EupO, \\ Jel)}} Ege(\text{X}, \text{DetF})_k + \sum_{\substack{X \in (Cop, \\ NCaS, NCaO, \\ EupS, EupO, \\ Jel)}} Mor(\text{X}, \text{DetF})_k - \sum_{\substack{X \in (EupS, \\ EupO)}} Gra(\text{DetF}, \text{X})_k$$

$$- Rem(\text{DetF}, \text{NH4})_k - \frac{Gra(\text{DetF}, \text{Ben}) \cdot w_{k, DetF}}{h_k} \tag{A71}$$

$$\frac{d}{dt}\text{Jel}_k = \sum_{\substack{X \in (Cop, \\ NCaS, NCaO, \\ EupS, EupO)}} Gra(\text{X}, \text{Jel})_k - Ege(\text{Jel}, \text{DetF})_k$$

$$- Mor(\text{Jel}, \text{DetF})_k - Res(\text{Jel}, \text{NH4})_k \tag{A72}$$





$$\frac{d}{dt}\text{Fe}_k = \left( \sum_{\substack{X \in (PhS, \\ PhL)}} Gpp(\text{NO3}, \text{X})_k \right) \cdot FeC \tag{A73}$$

$$\frac{d}{dt}\text{Ben} = \sum_{\substack{X \in (Det, \\ DetF,PhS, \\ PhL,BenDet)}} Gra(\text{X}, \text{Ben}) - \sum_{\substack{X \in (NH4, \\ BenDet)}} Exc(\text{Ben}, \text{X}) - Mor(\text{Ben}, \text{BenDet}) - Res(\text{Ben}, \text{NH4}) \tag{A74}$$

$$\frac{d}{dt}\text{BenDet} = Exc(\text{Ben}, \text{BenDet}) + Mor(\text{Ben}, \text{BenDet}) - Gra(\text{BenDet}, \text{Ben})$$
$$- Rem(\text{BenDet}, \text{NH4}) \tag{A75}$$

$$\frac{d}{dt}\text{IcePhL} = \sum_{\substack{X \in (IceNO3, \\ IceNH4)}} Gpp(\text{X}, \text{IcePhL})_{ice} - \left( \sum_{\substack{X \in (Cop, \\ NCaS,NCaO, \\ EupS,EupO)}} Gra(\text{IcePhL}, \text{X})_k \right) \cdot \frac{h_k}{h_{sice}}$$
$$- Mor(\text{IcePhL}, \text{IceNH4})_{ice} - Res(\text{IcePhL}, \text{IceNH4})_{ice} - \frac{Twi(\text{IcePhL}, \text{PhL})}{h_{sice}} \tag{A76}$$

$$\frac{d}{dt}\text{IceNO3} = \left( Nit(\text{IceNH4}, \text{IceNO3})_{sice} - Gpp(\text{IceNO3}, \text{IcePhL})_{ice} - \frac{Twi(\text{IceNO3}, \text{NO3})}{h_{sice}} \right) \cdot \xi \tag{A77}$$

$$\frac{d}{dt}\text{IceNH4} = \left( Res(\text{IcePhL}, \text{IceNH4})_{ice} + Mor(\text{IcePhL}, \text{IceNH4})_{ice} - Gpp(\text{IceNH4}, \text{IcePhL})_{ice} \right.$$
$$\left. - Nit(\text{IceNH4}, \text{IceNO3})_{ice} - \frac{Twi(\text{IceNH4}, \text{NH4})}{h_{sice}} \right) \cdot \xi \tag{A78}$$

**A4  Vertical movement and exchanges**

All vertical movement in the BESTNPZ model is calculated using a piecewise parabolic method and weighted non-oscillatory scheme, following the sediment settling code from a ROMS sediment model (Warner et al., 2008). This scheme allows for fast sinking speeds that may cause material to cross multiple layers, without being constrained by CFL criterion.

We have modified this scheme slightly to allow a zero-flux boundary condition to be imposed at a specified depth. We also
allow the use of a vertical velocity rather sinking speed; a negative velocity implies sinking, while a positive velocity indicates rising.

**A4.1  Sinking of phytoplankton and detritus**

Phytoplankton and detrital groups (PhS, PhL, Det, and DetF) are subject to vertical settling, with a constant sinking speed for each state variable.

When material crosses the water/benthic boundary, it is assumed that 20% of the material becomes biologically unavailable (Walsh et al., 1981; Walsh and McRoy, 1986), and 1% is lost to denitrification (pers. comm. with D. Schull via Gibson and Spitz, 2011). The remaining 79% of the flux across the boundary is transferred to the benthic detritus (DetBen.)





## A4.2  Copepod diapause

Copepod diapause is included in the BESTNPZ model by imposing seasonal movement through the water column on both large-bodied copepod groups (NCaS and NCaO), coupled with modifications to their feeding and respiration rates during periods of downward movement. The timing of copepod diapause is specified via input parameters for sinking start ($S_{start}$),

sinking end ($S_{end}$), rising start ($R_{start}$), and rising end ($R_{end}$) day for each group, all specified as days of the year. These four parameters combine to define periods of downward directed movement ($S_{start} \leq t_{doy} \leq S_{end}$), upward directed movement ($R_{start} \leq t_{doy} \leq R_{end}$), and no directed movement ($R_{end} < t_{doy} < S_{start}$ and $S_{end} < t_{doy} < R_{start}$) (during all times, both groups are still subject to passive advection and diffusion).

Earlier versions allowed specification of $S_{start}$, $S_{end}$, $R_{start}$, and $R_{end}$ for the off-shore group (NCaO) only, with the

onshore group automatically lagged by 30 days behind the offshore group. In the current version of the code, the 30-day lag is assumed when all four parameters are set to 0 for the on-shore group (this maintains backward-compatibility with older input files that do not include values for the on-shore group; BESTNPZ sets missing input parameters to zero by default). Diapause can be turned off for either group by setting all four timing parameters to the same non-zero value.

On-shore copepods migrate to 200 m depth during their diapause period. During downward movement, a zero-flux condition

is set at 200 m or at the bottom boundary, whichever is shallower. When migrating upward, a zero-flux condition is applied to the top of the surface layer.

Offshore copepods migrate to 400 m depth for their diapause period. A zero-flux condition is set at 400 m. In shallower waters, any biomass that crosses the bottom boundary is transferred to the benthic detritus (DetBen) group. During upward movement, a zero-flux condition is applied to the top of the surface layer.

## A4.3  Euphausiid diel vertical migration

Diel vertical migration is currently implemented for on-shelf euphausiids through the (still experimental) `EUPDIEL` compilation flag. When defined, sinking and rising velocities are applied such that EupS move at a hard-coded speed of $100 \, \mathrm{m\,d^{-1}}$ toward a target depth, defined as the shallowest depth layer where photosynthetically active radiation ($PAR \cdot I_0$) is less than $0.5 \, \mathrm{E/m^2/d}$. This option was turned off during the simulations detailed in this study.

Table A12: Notation and description for input parameters related to vertical movement.

| Variable | Description | Units | Relevant input parameter(s) | | |
| --- | --- | --- | --- | --- | --- |
| | | | Group | Parameter | Value |
| $w$ | sinking or rising speed | $\mathrm{d^{-1}}$ | PhS | wPhS | 0.05 |
| | | | PhL | wPhL | 1 |
| | | | Det | wDet | 1 |
| | | | DetF | wDetF | 10 |
| | | | NCaS (down) | wNCsink | 11 |
| | | | NCaS (up) | wNCrise | 12 |





| | | | | | |
|---|---|---|---|---|---|
| | | | NCaO (down) | `wNCsink` | 11 |
| | | | NCaO (up) | `wNCrise` | 12 |
| $S_{start}$ | start day of year for downward movement | $day-of-year$ | NCaS | `SinkStartCM` | 0 |
| | | | NCaO | `SinkStart` | 155 |
| $S_{end}$ | end day of year for downward movement | $day-of-year$ | NCaS | `SinkEndCM` | 0 |
| | | | NCaO | `SinkEnd` | 366 |
| $R_{start}$ | start day of year for upward movement | $day-of-year$ | NCaS | `RiseStartCM` | 0 |
| | | | NCaO | `RiseStart` | 0 |
| $R_{end}$ | end day of year for upward movement | $day-of-year$ | NCaS | `RiseEndCM` | 0 |
| | | | NCaO | `RiseEnd` | 60 |

## A5   Analytical relaxation of state variables

Iron concentrations throughout the water column are initialized using a vertical profile that prescribes values above $50\,\mathrm{m}$ and below $300\,\mathrm{m}$, with a linear interpolation between these two depths. The values of the shallow- and deep-water limits are a function of the bottom depth in a grid cell, with higher values in shallow water and lower values in deep water. The primary source of iron in this region is the sediment, and therefore this gradient between shallow and deep water is intended to capture the iron differences between on-shelf and off-shelf regions.

Iron-related biogeochemical processes are not included in this model. Instead, the iron state variable is continuously nudged towards these prescribed vertical profiles on an annual timescale. The nudging process is implemented using the generic ROMS framework for climatological nudging, with the `TNUDG` input parameter controlling the strength of the relaxation calculations.

Table A13: Notation and description for input parameters related to state variable nudging.

| Variable | Description | Units | Relevant input parameter(s) | | |
|---|---|---|---|---|---|
| | | | Group | Parameter | Value |
| | Surface value of iron in shallow water | $\mu\mathrm{mol\,Fe\,m^{-3}}$ | | `Feinlo` | 2 |
| | Below-mixed-layer value of iron in shallow water | $\mu\mathrm{mol\,Fe\,m^{-3}}$ | | `Feinhi` | 4 |
| | Surface value of iron in deep water | $\mu\mathrm{mol\,Fe\,m^{-3}}$ | | `Feofflo` | 0.01 |
| | Below-mixed-layer value of iron in deep water | $\mu\mathrm{mol\,Fe\,m^{-3}}$ | | `Feoffhi` | 2 |
| | Bottom depth corresponding to shallow water values | m | | `Feinh` | 20 |
| | Bottom depth corresponding to deep values | m | | `Feoffh` | 100 |
| | relaxation time interval for iron | d | | `TNUDG(iFe)` | 360 |





**Appendix B:  Light attenuation and limitation in BESTNPZ**

Light attenuation in the Bering10K+BESTNPZ model has undergone a variety of modifications between the Hermann et al. (2013) and Hermann et al. (2016) publications, and between the Hermann et al. (2016) publication and this one. In this section, we clarify the motivations behind these changes and their effects on both biological and physical state variables in the fully-coupled model.

Light attenuation formulations are used in a typical, physics-only implementation of ROMS in order to determine the attenuation of shortwave radiation in the water column and distribute surface heat fluxes appropriately. When a biological module is added, any light attenuation calculations necessary for photosynthesis (typically focused only on the photosynthetically active wavelengths) are coded separately, independent of the shortwave attenuation code in the physical core of ROMS, and with no direct feedback of the biology on the physics.

The Hermann et al. (2013) version of BESTNPZ followed this default setup; the attenuation coefficient of photosynthetically-active radiation for the biological model followed Gibson and Spitz (2011):

$$I_{PAR}(z) = I_{PAR}(0) \exp(-Kz) \tag{B1}$$
$$K = k_{ext} + k_A C^{0.0428} \tag{B2}$$

where $C$ is the concetration of chlorophyll-a ($\mathrm{mg\,chla\,m^{-3}}$), $k_{ext}$ is the clear-water attenuation coefficient, and the $k_A$ term is attenuation due to light absoption by chlorophyll; the $k_{ext}$ and $k_A$ values, as well as the $C$-term exponent, derive from an empirical fit of open-ocean chlorophyll concentrations versus measured attenuation (Morel, 1988).

The physical model relied on the ROMS default formula, which applies separate attenuation length scales to blue-green wavelengths and other wavelengths following Paulson and Simpson (1977):

$$I(z) = I(0)\left((1 - a_{frac})\exp(-z/a_{\mu 1}) + a_{frac}\exp(-z/a_{\mu 2})\right) \tag{B3}$$

The values for $a_{frac}$ (unitless), $a_{\mu 1}$ (m), and $a_{\mu 2}$ (m) are set via an internal lookup table based on one of several water classification types. These water type classes account for the varying chlorophyll, sediment, and CDOM concentrations in different types of environments. Early versions of the Bering10K domain alternated between Class I ($a_{frac} = 0.58$, $a_{\mu 1} = 0.35$, $a_{\mu 2} = 23.0$) and Class III ($a_{frac} = 0.78$, $a_{\mu 1} = 1.4$, $a_{\mu 2} = 7.9$) parameters; the Hermann et al. (2013) study used the Class III option. Note that in this setup, the amount of chlorophyll in the water column assumed by the parameters of Equation B3 is independent from the chlorophyll levels modeled by the coupled biological model.

A more harmonious approach to light attenuation was adopted in the Hermann et al. (2016) version of the model. As one of several adjustments aimed at addressing a warm bias in heating of the water column, the attenuation equation in the physical model replaced the $a_{\mu 2}$ attenuation length scale parameter with the PAR-related length scale ($K^{-1}$) from the biological model; this provided direct feedback of phytoplankton on heat absorption in the water column, and better encompassed the varying




light conditions expected between the low-productivity basin and high-productivity shelf region. A rough approximation of attentuation due to sediment was also added as a function of bottom depth ($h$) in both the biological and physical calculations for light attenuation:

$$K = k_{ext} + k_A C^{0.0428} + 2.0\exp\left(0.05h\right) \tag{B4}$$

Hermann et al. (2016) also increased the value the $k_A$ parameter (from 0.0518 to 0.121), and adjusted the values of $a_{frac}$ and $a_{\mu 1}$ to Class I values (now hard-coded within the source code).

The new biological feedback resulted in only a very small change in water column heat content, and the model warm bias was later addressed through other adjustments to surface boundary conditions. However, the addition of the sediment attenuation term strongly affected the biological calculations. Even in the absence of grazing and under nutrient-replete, peak-irradiance

conditions, the Hermann et al. (2016) set of parameters raises the compensation depths for both large and small phytoplankton to very shallow depths; in the shallowest waters of our domain, primary production is only possible in the upper $4\,\mathrm{m}$ to $5\,\mathrm{m}$ of the water column (Figure B1). While high sediment attenuation is possible in a few specific locations in the Bering10K domain, near the mouths of the Yukon, Anadyr, and Kuskokwim Rivers during peak streamflow, there is no evidence to support such strong sediment shading throughout the entire inner domain.

During early validation of the circa-Hermann et al. (2016) model for this study (prior to any adjustments for bug fixes in the code), it was also noted that light was the key limiting factor in the deep basin region. The sediment attenuation term is negligible in this deep-water region, and the compensation depth reasonable in this location. However, the coarse vertical resolution of the model domain resulted in a surface layer nearly $40\,\mathrm{m}$ thick. The discretization used by this model assumes that all processes are approximately linear within a given layer, and that the layer midpoint can be therefore be used to approximate

characteristics of the layer. In the case of light levels in these deep basin surface layers, this assumption did not hold, and the resulting light levels at the mid-points of the surface layers were too low to support any significant growth of phytoplankton.

For this study, all parameters related to light were revisited. A new sediment attenuation term was derived empirically from satellite inherent optical property measurements from the region (Figure B2). This remains a rough estimate that does not consider the seasonal variability of sediment and detrital matter that contribute to this term, but alleviates the previously excessive

limitation in shallow water. Attenuation-related parameters for clear-water attenuation and chlorophyll-based attenuation were updated to values consistent with literature ($k_{ext} = 0.034$ following Morel et al. (2007) and $k_A = 0.0518$ and $k_B = 0.428$ following Morel (1988)). It was noted that the value used for for photosynthetically active radiation fraction ($PAR_{frac} = 0.5$) in the Gibson and Spitz (2011), Hermann et al. (2013), and Hermann et al. (2016) publications was higher than most field-based and analytical estimates. We adjusted this parameter to a value of 0.42 based on examination of satellite-derived PAR versus

our model's surface boundary condition shortwave radiation values. Finally, the vertical resolution in the physical model was increased from 10 layers to 30 layers to allow for better resolution of mixed layer dynamics and light attenuation in the basin.

The updated light attenuation equations and parameters, coupled with the higher vertical resolution in the newer simulations, better reflect the true mechanisms controlling light levels in the Bering Sea; the bathymetry-following artifacts in phytoplankton



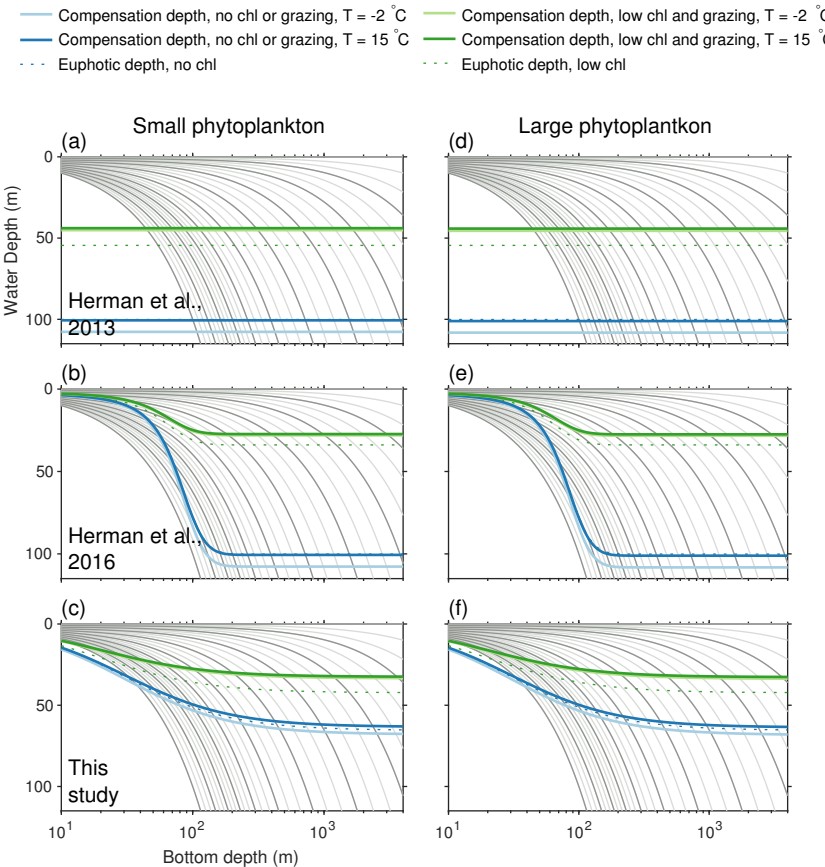

**Figure B1.** Compensation depth (depth where photosynthesis balances phytoplankton loss terms) and euphotic depth (depth of 1% PAR) in three versions of the BESTNPZ model, under low (light-colored) and high (dark-colored) temperature conditions, assuming a surface irradiance of $200\,\mathrm{W\,m^{-2}}$ and nutrient-replete waters. The "no chl or grazing" scenario (blue) assumes losses due to respiration and non-predatory mortality only, with no attenuation due to phytoplankton itself. The "low chl and grazing" scenario (green) assumes attenuation due to $0.5\,\mathrm{mmol\,chl\,m^{-3}}$ throughout the water column and adds a constant grazing loss rate of $0.05\,\mathrm{d^{-1}}$. Grey lines indicate boundaries of the ROMS depth levels in the 10-layer (dark) and 30-layer (light) versions of the model. (Note: values are very nearly identical for small and large phytoplankton due to the proportional scaling of rate parameters between the groups.)

spatial patterns are no longer present (Figure B3). However, the lower light limitation in certain parts of the domain has exposed some previously-overlooked deficiencies in micro- and macronutrient limitation in the BESTNPZ model. Observations suggest that the deep basin should be a low production region, a pattern that is present but for the wrong reasons in the Hermann et al. (2016) version of the model and absent in the updated, more-mechanistically sound version of the model. For some applications,

5   it may be more useful to have the correct gradient between on- and off-shelf populations, even if it is for the incorrect reason, than to have mechanistically-sound light limitation; however, users should be fully aware of this discrepancy before analyzing



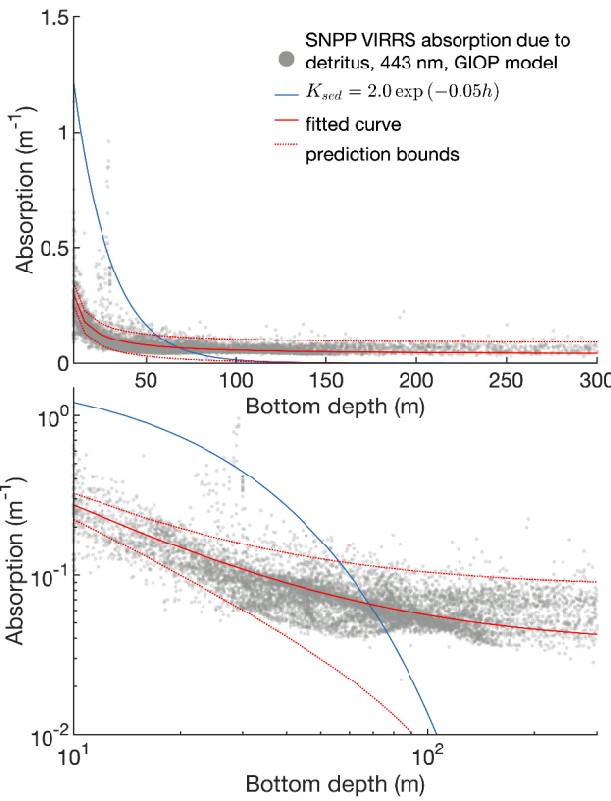

**Figure B2.** Attenuation due to sediment derives from a power law fit of satellite estimations of absorption due to gelbstoff and detrital material (entire mission composite from VIIRS, 2012-2018) versus bottom depth. Subpanels show the same data in linear and logarithmic space.

any biological output from either the earlier Hermann et al. (2016) simulations or this newer set of simulations using the updated code.



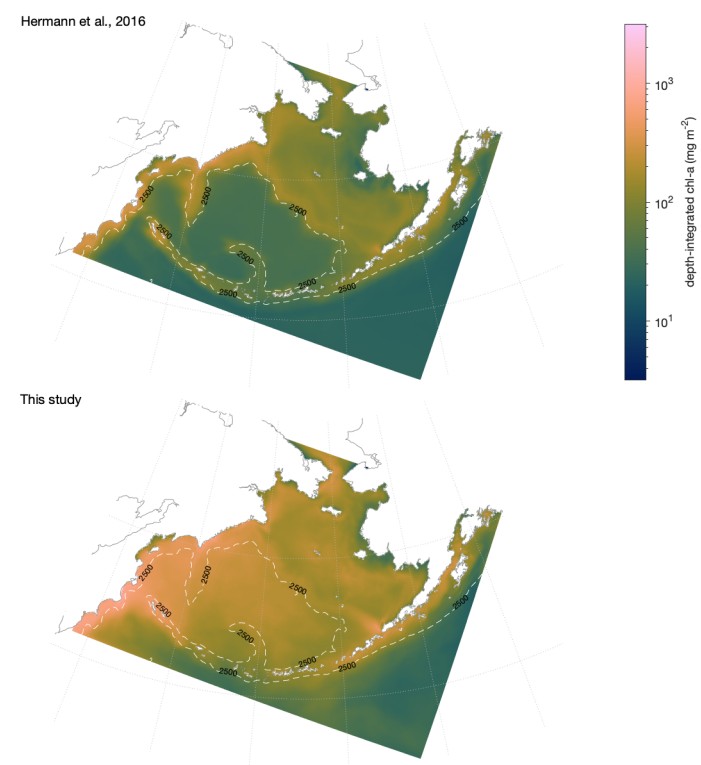

**Figure B3.** Mean depth-integrated chlorophyll in the Hermann et al. (2016) version of BESTNPZ vs this study. The contour line indicates the $2500\,\mathrm{m}$ isobath.

## Appendix C: Coupling BESTNPZ to ROMS: compilation options and output variables

The following appendix provides instructions for setting up input for, compiling, and retrieving output from a simulation of ROMS coupled to the BESTNPZ biological model. At present the BESTNPZ code has only been coupled to the Bering10K domain, and a few quirks of its internal coding prevent it from being coupled to other physical domains in its current form.

5 Therefore, this users' manual should be considered to be specific to the Bering10K+BESTNPZ ROMS model setup.

The primary input parameters specific to the BESTNPZ are described in Appendix A. The remaining setup options relate to the compilation flags and the indices of input and output state variables, described in the following sections.

### C1 Compilation flags

Within the ROMS source code, C preprocessing flags are used to selectively compile the code. Several C preprocessing flags

10 appear throughout the BESTNPZ code; those that currently exist in the master branch of the source code are detailed in the following table.





A few of these compilation options resulted from the gradual addition of new features (such as the addition of benthic and ice state variables) that later became, for all intents and purposes, permanent additions to the BESTNPZ model. These are noted in the table as "recommended to always define." While we have attempted to keep the source code flexible to all compilation options, we rarely test for compilation stability with these undefined, and all recent model validation work assumes these model features are present.

Table C1: Compilation flags related to the BEST_NPZ model.

| Compilation flag | Purpose | Notes |
|---|---|---|
| BENTHIC | Turns on benthic portion of the module. This adds the Ben and BenDet variables along with all associated flux processes. | Advised to always define with BEST_NPZ; not often tested without benthos. |
| BERING_10K | ROMS application flag for Bering 10K domain. Also used in many parts of the ice code as shorthand for "use variables from ice model" (as opposed to analytical one-dimensional ice) | The ice-specific use of this flag is not robust to other model domains with their own unique application flags. |
| CLIM_ICE_1D | Use analytical calculations for a seasonal ice cycle (typically used when no full ice model is coupled to the physical model, as in one-dimensional simulations) | |
| CORRECT_TEMP_BIAS | Subtracts 1.94°C from water temperature for biological rate calculations | Legacy... not sure when or where this bias-correction was needed. |
| DEPTHLIMITER | Switch to turn on on/off-shelf enforcers for NCa and Eup groups | Provided for consistency with ggibson branch, but not recommended |
| DIAPAUSE | Turn on copepod diapause (vertical movement with accompanying reductions in feeding and respiration) | Recommended to always define. Behavior can now be turned off for individual simulations through use of input parameters without the need to recompile. |
| DIURNAL_SRFLUX | Sets day length (a now-unused internal parameter) to 24 hours rather than a latitude-and-declination calculation | |
| EUPDIEL | Turn on euphausiid diel vertical migration | Still experimental |





| FEAST | Turns on the FEAST upper trophic level model | Not documented here; requires a large number of additional input variables and parameters. |
|---|---|---|
| FEAST_NOEXCHANGE | | |
| GPPMID | Calculate gross primary production at midpoint of layer, rather than integrating over the layer. | Usually defined. The integrate-over-layer option was marginally better in deep water when using a coarse 10-layer depth resolution. However, increasing the vertical resolution and sticking with the typical midpoint calculations proved to be a much better solution. |
| ICE_BIO | Turn on ice biology. This adds the IcePhL, IceNO3, and IceNH4 variables and all related fluxes | Advised to always define with BEST_NPZ; not often tested without ice biology. Also, must define either CLIM_ICE_1D or BERING_10K to work. |
| IRON_LIMIT | Turn on iron limitation. This adds the Fe variable and all associated fluxes. | Advised to always define with BEST_NPZ; not often tested without iron limitation. |
| JELLY | Turn on jellyfish. This adds the Jel variable and all associated fluxes. | Advised to always define with BEST_NPZ; not often tested without jellyfish. |
| LINEAR_CONTINUATION | An option in the sinking code subfunction (used for particle sinking and diapause) related to the WENO scheme. Inherited from sediment sinking code. | |
| MATLABCOMPILE | Modify biology_tile subroutine for standalone compilation | This flag should never be defined when running a full ROMS simulation; intended for my Matlab-based unit-testing environment (allows the file to be compiled as a Matlab mex-Fortran file). |
| MZLM0LIN | Switch to linear form for MZL non-predatory mortality (quadratic otherwise) | |
| NEUMANN | An option in the sinking code related to the WENO scheme | |





| | | |
|---|---|---|
| PI_CONSTANT | Use a constant value (provided as an input parameter) for the $\alpha$ parameter when calculating the photosynthesis-irradiance curve, as opposed to setting $\alpha$ as a function of irradiance as in the ggibson branch. | Recommended for now... variable-$\alpha$ equation is a bit questionable. |
| SPINUPBIO | Run the model in biological spinup mode, starting with only deep nitrate (at 40 mmol N m$^{-3}$ below 300 m). | Only relevant if ANA_BIOLOGY is defined. |
| STATIONARY | Calculate 3D stationary diagnostic variables. | |
| STATIONARY2 | Calculate 2D stationary diagnostic variables. | Currently a placeholder, but available if we need any 2D diagnostics |
| fixedPRED | | Legacy; do not define. |

Flags not specific to BEST_NPZ but found in the bestnpz.h source code

| | | |
|---|---|---|
| ASSUMED_SHAPE | | |
| DISTRIBUTE | Internal switch to run in parallel | |
| EW_PERIODIC | Use east-west periodic boundary conditions | |
| MASKING | Use land/sea masking | |
| NS_PERIODIC | Use north-south periodic boundary conditions | |
| PROFILE | Use time profiling | |
| TS_MPDATA | Use MPDATA finite difference solver for 3D advective time-stepping | |





## C2    Output variables and input parameter indices

The BESTNPZ module allows for a large number of variables to be added to the various ROMS output files (history, average, and station files).

Regardless of compilation options, the biological tracer variables are saved and available as output variables. Water column tracers are specified via the `H/Sout(idTvar)` input parameters. The exact tracer variables available, and their positions within the `idTvar` array, depend on whether the `ICE_BIO`, `BENTHIC`, `IRON_LIMIT`, and `JELLY` compilation flags are defined; see Table C2 for details. Benthic tracers can be specified for output through the `H/Sout(idBvar)` input; `iBen = 1` and `iDetBen = 2` within this array. The ice variables are turned on and off with the variable-specific input parameters `H/Sout(idIcePhL)`, `H/Sout(idIceNO3)`, and `H/Sout(idIceNH4)`.

**Table C2.** Indices in the `idTvar` array. Note that `Hout(idTvar)` appears separately in the ocean.in file and BPARNAM file, with the latter including only biological active tracers. The `Sout(idTvar)` input appears only once, in the STANAME file, and biological tracer indices begin at NAT+1; NAT is here assumed to be 2, for temperature and salinity. In all cases, note the skip between `iPhL` and `iMZL` due to the now-deprecated small microzooplankton group.

| Index | Hout (in BPARNAM file) | | | | Sout | | | |
|-------|------|-------|------------|------|------|-------|------------|------|
|       | none | JELLY | IRON_LIMIT | both | none | JELLY | IRON_LIMIT | both |
| iNO3  | 1 | 1 | 1 | 1 | 3 | 3 | 3 | 3 |
| iNH4  | 2 | 2 | 2 | 2 | 4 | 4 | 4 | 4 |
| iPhS  | 3 | 3 | 3 | 3 | 5 | 5 | 5 | 5 |
| iPhL  | 4 | 4 | 4 | 4 | 6 | 6 | 6 | 6 |
| iMZL  | 6 | 6 | 6 | 6 | 8 | 8 | 8 | 8 |
| iCop  | 7 | 7 | 7 | 7 | 9 | 9 | 9 | 9 |
| iNCaS | 8 | 8 | 8 | 8 | 10 | 10 | 10 | 10 |
| iEupS | 9 | 9 | 9 | 9 | 11 | 11 | 11 | 11 |
| iNCaO | 10 | 10 | 10 | 10 | 12 | 12 | 12 | 12 |
| iEupO | 11 | 11 | 11 | 11 | 13 | 13 | 13 | 13 |
| iDet  | 12 | 12 | 12 | 12 | 14 | 14 | 14 | 14 |
| iDetF | 13 | 13 | 13 | 13 | 15 | 15 | 15 | 15 |
| iJel  | – | 14 | – | 14 | – | 16 | – | 16 |
| iFe   | – | – | 14 | 15 | – | – | 16 | 17 |

If the `STATIONARY` flag is defined, many more intermediate diagnostic variables are saved internally and available for output. These are controlled by the `idTSvar` index array to `Hout` and `Sout`. Note that when running with a subdivided timestep (`BioIter > 1`), these diagnostic variables will reflect the values calculated during the final subdivision only.





Older versions of the code included a second compilation flag, `STATIONARY2` (with the corresponding index array `idTS2var`), to define two-dimensional stationary diagnostics. In my rewrite, I did not end up requiring any 2D diagnostics. However, the code structure is still in place for this if we decided to use it in the future.

See Table C4 for a comprehensive description of all output variables associated with the BESTNPZ model.

Table C4: Output variables available in the BEST_NPZ and FEAST models.

| Index | Short name | Long name | Variable type |
|---|---|---|---|
| `idTvar(iNO3)` | NO3 | Nitrate concentration | 3D RHO-variable |
| `idTvar(iNH4)` | NH4 | Ammonium concentration | 3D RHO-variable |
| `idTvar(iPhS)` | PhS | Small phytoplankton concentration | 3D RHO-variable |
| `idTvar(iPhL)` | PhL | Large phytoplankton concentration | 3D RHO-variable |
| `idTvar(iMZL)` | MZL | Microzooplankton concentration | 3D RHO-variable |
| `idTvar(iCop)` | Cop | Small copepod concentration | 3D RHO-variable |
| `idTvar(iNCaS)` | NCaS | On-shelf large copepod concentration | 3D RHO-variable |
| `idTvar(iEupS)` | EupS | On-shelf euphausiid concentration | 3D RHO-variable |
| `idTvar(iNCaO)` | NCaO | Offshore large copepod concentration | 3D RHO-variable |
| `idTvar(iEupO)` | EupO | Offshore euphausiid concentration | 3D RHO-variable |
| `idTvar(iDet)` | Det | Slow-sinking detritus concentration | 3D RHO-variable |
| `idTvar(iDetF)` | DetF | Fast-sinking detritus concentration | 3D RHO-variable |
| `idTvar(iJel)` | Jel | Jellyfish concentration | 3D RHO-variable |
| `idBvar(iBen)` | Ben | Benthic infauna concentration | 2D RHO-variable |
| `idBvar(iDetBen)` | DetBen | Benthic detritus concentration | 2D RHO-variable |
| `idIcePhL` | IcePhL | Ice algae concentration | 2D RHO-variable |
| `idIceNO3` | IceNO3 | Ice nitrate concentration | 2D RHO-variable |
| `idIceNH4` | IceNH4 | Ice ammonium concentration | 2D RHO-variable |
| `idTSvar(i3Stat1)` | LightLimS | PhS Light limitation | 3D RHO-variable |
| `idTSvar(i3Stat2)` | LightLimL | PhL Light limitation | 3D RHO-variable |
| `idTSvar(i3Stat3)` | NOLimS | PhS NO3 limitation | 3D RHO-variable |
| `idTSvar(i3Stat4)` | NOLimL | PhL NO3 limitation | 3D RHO-variable |
| `idTSvar(i3Stat5)` | NHLimS | PhS NH4 limitation | 3D RHO-variable |
| `idTSvar(i3Stat6)` | NHLimL | PhL NH4 limitation | 3D RHO-variable |
| `idTSvar(i3Stat7)` | IronLimS | PhS Iron limitation | 3D RHO-variable |
| `idTSvar(i3Stat8)` | IronLimL | PhL Iron limitation | 3D RHO-variable |
| `idTSvar(i3Stat9)` | Gpp_NO3_PhS | gross primary production flux from NO3 to PhS | 3D RHO-variable |
| `idTSvar(i3Stat10)` | Gpp_NO3_PhL | gross primary production flux from NO3 to PhL | 3D RHO-variable |
| `idTSvar(i3Stat11)` | Gpp_NH4_PhS | gross primary production flux from NH4 to PhS | 3D RHO-variable |
| `idTSvar(i3Stat12)` | Gpp_NH4_PhL | gross primary production flux from NH4 to PhL | 3D RHO-variable |
| `idTSvar(i3Stat13)` | Gra_PhS_MZL | grazing/predation flux from PhS to MZL | 3D RHO-variable |
| `idTSvar(i3Stat14)` | Gra_PhL_MZL | grazing/predation flux from PhL to MZL | 3D RHO-variable |
| `idTSvar(i3Stat15)` | Ege_MZL_Det | egestion flux from MZL to Det | 3D RHO-variable |
| `idTSvar(i3Stat16)` | Gra_PhS_Cop | grazing/predation flux from PhS to Cop | 3D RHO-variable |
| `idTSvar(i3Stat17)` | Gra_PhL_Cop | grazing/predation flux from PhL to Cop | 3D RHO-variable |





| | | | |
|---|---|---|---|
| `idTSvar(i3Stat18)` | Gra_MZL_Cop | grazing/predation flux from MZL to Cop | 3D RHO-variable |
| `idTSvar(i3Stat19)` | Gra_IPhL_Cop | grazing/predation flux from IcePhL to Cop | 3D RHO-variable |
| `idTSvar(i3Stat20)` | Ege_Cop_DetF | egestion flux from Cop to DetF | 3D RHO-variable |
| `idTSvar(i3Stat21)` | Gra_PhS_NCaS | grazing/predation flux from PhS to NCaS | 3D RHO-variable |
| `idTSvar(i3Stat22)` | Gra_PhL_NCaS | grazing/predation flux from PhL to NCaS | 3D RHO-variable |
| `idTSvar(i3Stat23)` | Gra_MZL_NCaS | grazing/predation flux from MZL to NCaS | 3D RHO-variable |
| `idTSvar(i3Stat24)` | Gra_IPhL_NCaS | grazing/predation flux from IcePhL to NCaS | 3D RHO-variable |
| `idTSvar(i3Stat25)` | Ege_NCaS_DetF | egestion flux from NCaS to DetF | 3D RHO-variable |
| `idTSvar(i3Stat26)` | Gra_PhS_NCaO | grazing/predation flux from PhS to NCaO | 3D RHO-variable |
| `idTSvar(i3Stat27)` | Gra_PhL_NCaO | grazing/predation flux from PhL to NCaO | 3D RHO-variable |
| `idTSvar(i3Stat28)` | Gra_MZL_NCaO | grazing/predation flux from MZL to NCaO | 3D RHO-variable |
| `idTSvar(i3Stat29)` | Gra_IPhL_NCaO | grazing/predation flux from IcePhL to NCaO | 3D RHO-variable |
| `idTSvar(i3Stat30)` | Ege_NCaO_DetF | egestion flux from NCaO to DetF | 3D RHO-variable |
| `idTSvar(i3Stat31)` | Gra_PhS_EupS | grazing/predation flux from PhS to EupS | 3D RHO-variable |
| `idTSvar(i3Stat32)` | Gra_PhL_EupS | grazing/predation flux from PhL to EupS | 3D RHO-variable |
| `idTSvar(i3Stat33)` | Gra_MZL_EupS | grazing/predation flux from MZL to EupS | 3D RHO-variable |
| `idTSvar(i3Stat34)` | Gra_Cop_EupS | grazing/predation flux from Cop to EupS | 3D RHO-variable |
| `idTSvar(i3Stat35)` | Gra_IPhL_EupS | grazing/predation flux from IcePhL to EupS | 3D RHO-variable |
| `idTSvar(i3Stat36)` | Gra_Det_EupS | grazing/predation flux from Det to EupS | 3D RHO-variable |
| `idTSvar(i3Stat37)` | Gra_DetF_EupS | grazing/predation flux from DetF to EupS | 3D RHO-variable |
| `idTSvar(i3Stat38)` | Ege_EupS_DetF | egestion flux from EupS to DetF | 3D RHO-variable |
| `idTSvar(i3Stat39)` | Gra_PhS_EupO | grazing/predation flux from PhS to EupO | 3D RHO-variable |
| `idTSvar(i3Stat40)` | Gra_PhL_EupO | grazing/predation flux from PhL to EupO | 3D RHO-variable |
| `idTSvar(i3Stat41)` | Gra_MZL_EupO | grazing/predation flux from MZL to EupO | 3D RHO-variable |
| `idTSvar(i3Stat42)` | Gra_Cop_EupO | grazing/predation flux from Cop to EupO | 3D RHO-variable |
| `idTSvar(i3Stat43)` | Gra_IPhL_EupO | grazing/predation flux from IcePhL to EupO | 3D RHO-variable |
| `idTSvar(i3Stat44)` | Gra_Det_EupO | grazing/predation flux from Det to EupO | 3D RHO-variable |
| `idTSvar(i3Stat45)` | Gra_DetF_EupO | grazing/predation flux from DetF to EupO | 3D RHO-variable |
| `idTSvar(i3Stat46)` | Ege_EupO_DetF | egestion flux from EupO to DetF | 3D RHO-variable |
| `idTSvar(i3Stat47)` | Gra_Cop_Jel | grazing/predation flux from Cop to Jel | 3D RHO-variable |
| `idTSvar(i3Stat48)` | Gra_EupS_Jel | grazing/predation flux from EupS to Jel | 3D RHO-variable |
| `idTSvar(i3Stat49)` | Gra_EupO_Jel | grazing/predation flux from EupO to Jel | 3D RHO-variable |
| `idTSvar(i3Stat50)` | Gra_NCaS_Jel | grazing/predation flux from NCaS to Jel | 3D RHO-variable |
| `idTSvar(i3Stat51)` | Gra_NCaO_Jel | grazing/predation flux from NCaO to Jel | 3D RHO-variable |
| `idTSvar(i3Stat52)` | Ege_Jel_DetF | egestion flux from Jel to DetF | 3D RHO-variable |
| `idTSvar(i3Stat53)` | Mor_PhS_Det | other mortality flux from PhS to Det | 3D RHO-variable |
| `idTSvar(i3Stat54)` | Mor_PhL_Det | other mortality flux from PhL to Det | 3D RHO-variable |
| `idTSvar(i3Stat55)` | Mor_MZL_Det | other mortality flux from MZL to Det | 3D RHO-variable |
| `idTSvar(i3Stat56)` | Mor_Cop_DetF | other mortality flux from Cop to DetF | 3D RHO-variable |
| `idTSvar(i3Stat57)` | Mor_NCaS_DetF | other mortality flux from NCaS to DetF | 3D RHO-variable |
| `idTSvar(i3Stat58)` | Mor_EupS_DetF | other mortality flux from EupS to DetF | 3D RHO-variable |
| `idTSvar(i3Stat59)` | Mor_NCaO_DetF | other mortality flux from NCaO to DetF | 3D RHO-variable |
| `idTSvar(i3Stat60)` | Mor_EupO_DetF | other mortality flux from EupO to DetF | 3D RHO-variable |
| `idTSvar(i3Stat61)` | Mor_Jel_DetF | other mortality flux from Jel to DetF | 3D RHO-variable |





| | | | |
|---|---|---|---|
| idTSvar(i3Stat62) | Res_PhS_NH4 | respiration flux from PhS to NH4 | 3D RHO-variable |
| idTSvar(i3Stat63) | Res_PhL_NH4 | respiration flux from PhL to NH4 | 3D RHO-variable |
| idTSvar(i3Stat64) | Res_MZL_NH4 | respiration flux from MZL to NH4 | 3D RHO-variable |
| idTSvar(i3Stat65) | Res_Cop_NH4 | respiration flux from Cop to NH4 | 3D RHO-variable |
| idTSvar(i3Stat66) | Res_NCaS_NH4 | respiration flux from NCaS to NH4 | 3D RHO-variable |
| idTSvar(i3Stat67) | Res_NCaO_NH4 | respiration flux from NCaO to NH4 | 3D RHO-variable |
| idTSvar(i3Stat68) | Res_EupS_NH4 | respiration flux from EupS to NH4 | 3D RHO-variable |
| idTSvar(i3Stat69) | Res_EupO_NH4 | respiration flux from EupO to NH4 | 3D RHO-variable |
| idTSvar(i3Stat70) | Res_Jel_NH4 | respiration flux from Jel to NH4 | 3D RHO-variable |
| idTSvar(i3Stat71) | Rem_Det_NH4 | remineralization flux from Det to NH4 | 3D RHO-variable |
| idTSvar(i3Stat72) | Rem_DetF_NH4 | remineralization flux from DetF to NH4 | 3D RHO-variable |
| idTSvar(i3Stat73) | Nit_NH4_NO3 | nitrification flux from NH4 to NO3 | 3D RHO-variable |
| idTSvar(i3Stat74) | Gra_Det_Ben | grazing/predation flux from Det to Ben | 3D RHO-variable |
| idTSvar(i3Stat75) | Gra_DetF_Ben | grazing/predation flux from DetF to Ben | 3D RHO-variable |
| idTSvar(i3Stat76) | Gra_PhS_Ben | grazing/predation flux from PhS to Ben | 3D RHO-variable |
| idTSvar(i3Stat77) | Gra_PhL_Ben | grazing/predation flux from PhL to Ben | 3D RHO-variable |
| idTSvar(i3Stat78) | Gra_DetBen_Ben | grazing/predation flux from DetBen to Ben | 3D RHO-variable |
| idTSvar(i3Stat79) | Exc_Ben_NH4 | excretion flux from Ben to NH4 | 3D RHO-variable |
| idTSvar(i3Stat80) | Exc_Ben_DetBen | excretion flux from Ben to DetBen | 3D RHO-variable |
| idTSvar(i3Stat81) | Res_Ben_NH4 | respiration flux from Ben to NH4 | 3D RHO-variable |
| idTSvar(i3Stat82) | Mor_Ben_DetBen | other mortality flux from Ben to DetBen | 3D RHO-variable |
| idTSvar(i3Stat83) | Rem_DetBen_NH4 | remineralization flux from DetBen to NH4 | 3D RHO-variable |
| idTSvar(i3Stat84) | Gpp_INO3_IPhL | gross primary production flux from IceNO3 to IcePhL | 3D RHO-variable |
| idTSvar(i3Stat85) | Gpp_INH4_IPhL | gross primary production flux from IceNH4 to IcePhL | 3D RHO-variable |
| idTSvar(i3Stat86) | Res_IPhL_INH4 | respiration flux from IcePhL to IceNH4 | 3D RHO-variable |
| idTSvar(i3Stat87) | Mor_IPhL_INH4 | other mortality flux from IcePhL to IceNH4 | 3D RHO-variable |
| idTSvar(i3Stat88) | Nit_INH4_INO3 | nitrification flux from IceNH4 to IceNO3 | 3D RHO-variable |
| idTSvar(i3Stat89) | Twi_IPhL_PhL | ice/water exchange flux from IcePhL to PhL | 3D RHO-variable |
| idTSvar(i3Stat90) | Twi_INO3_NO3 | ice/water exchange flux from IceNO3 to NO3 | 3D RHO-variable |
| idTSvar(i3Stat91) | Twi_INH4_NH4 | ice/water exchange flux from IceNH4 to NH4 | 3D RHO-variable |
| idTSvar(i3Stat92) | Ver_PhS_DetBen | sinking-to-bottom flux from PhS to DetBen | 3D RHO-variable |
| idTSvar(i3Stat93) | Ver_PhS_Out | sinking-to-bottom flux from PhS to Out | 3D RHO-variable |
| idTSvar(i3Stat94) | Ver_PhL_DetBen | sinking-to-bottom flux from PhL to DetBen | 3D RHO-variable |
| idTSvar(i3Stat95) | Ver_PhL_Out | sinking-to-bottom flux from PhL to Out | 3D RHO-variable |
| idTSvar(i3Stat96) | Ver_Det_DetBen | sinking-to-bottom flux from Det to DetBen | 3D RHO-variable |
| idTSvar(i3Stat97) | Ver_Det_Out | sinking-to-bottom flux from Det to Out | 3D RHO-variable |
| idTSvar(i3Stat98) | Ver_DetF_DetBen | sinking-to-bottom flux from DetF to DetBen | 3D RHO-variable |
| idTSvar(i3Stat99) | Ver_DetF_Out | sinking-to-bottom flux from DetF to Out | 3D RHO-variable |
| idTSvar(i3Stat100) | Ver_NCaO_DetBen | sinking-to-bottom flux from NCaO to DetBen | 3D RHO-variable |
| idTSvar(i3Stat101) | Ver_NCaS_DetF | sinking-to-bottom flux from NCaS to DetF | 3D RHO-variable |
| idTSvar(i3Stat102) | Ver_NCaS_DetBen | sinking-to-bottom flux from NCaS to DetBen | 3D RHO-variable |
| idTSvar(i3Stat103) | Frz_PhL_IPhL | freezing(+)/melting(-) flux from PhL to IcePhL | 3D RHO-variable |
| idTSvar(i3Stat104) | Frz_NO3_INO3 | freezing(+)/melting(-) flux from NO3 to IceNO3 | 3D RHO-variable |
| idTSvar(i3Stat105) | Frz_NH4_INH4 | freezing(+)/melting(-) flux from NH4 to IceNH4 | 3D RHO-variable |





| | | | |
|---|---|---|---|
| idTSvar(i3Stat106) | prod_PhS | PhS net production rate | 3D RHO-variable |
| idTSvar(i3Stat107) | prod_PhL | PhL net production rate | 3D RHO-variable |
| idTSvar(i3Stat108) | prod_MZL | MZL net production rate | 3D RHO-variable |
| idTSvar(i3Stat109) | prod_Cop | Cop net production rate | 3D RHO-variable |
| idTSvar(i3Stat110) | prod_NCaS | NCaS net production rate | 3D RHO-variable |
| idTSvar(i3Stat111) | prod_EupS | EupS net production rate | 3D RHO-variable |
| idTSvar(i3Stat112) | prod_NCaO | NCaO net production rate | 3D RHO-variable |
| idTSvar(i3Stat113) | prod_EupO | EupO net production rate | 3D RHO-variable |
| idTSvar(i3Stat114) | prod_Jel | Jel net production rate | 3D RHO-variable |
| idTSvar(i3Stat115) | prod_Ben | Ben net production rate | 3D RHO-variable |
| idTSvar(i3Stat116) | prod_IcePhL | IcePhL net production rate | 3D RHO-variable |
| idTSvar(i3Stat117) | onExit_NO3 | NO3 biomass tracker | 3D RHO-variable |
| idTSvar(i3Stat118) | onExit_NH4 | NH4 biomass tracker | 3D RHO-variable |
| idTSvar(i3Stat119) | onExit_PhS | PhS biomass tracker | 3D RHO-variable |
| idTSvar(i3Stat120) | onExit_PhL | PhL biomass tracker | 3D RHO-variable |
| idTSvar(i3Stat121) | onExit_MZL | MZL biomass tracker | 3D RHO-variable |
| idTSvar(i3Stat122) | onExit_Cop | Cop biomass tracker | 3D RHO-variable |
| idTSvar(i3Stat123) | onExit_NCaS | NCaS biomass tracker | 3D RHO-variable |
| idTSvar(i3Stat124) | onExit_EupS | EupS biomass tracker | 3D RHO-variable |
| idTSvar(i3Stat125) | onExit_NCaO | NCaO biomass tracker | 3D RHO-variable |
| idTSvar(i3Stat126) | onExit_EupO | EupO biomass tracker | 3D RHO-variable |
| idTSvar(i3Stat127) | onExit_Det | Det biomass tracker | 3D RHO-variable |
| idTSvar(i3Stat128) | onExit_DetF | DetF biomass tracker | 3D RHO-variable |
| idTSvar(i3Stat129) | onExit_Jel | Jel biomass tracker | 3D RHO-variable |
| idTSvar(i3Stat130) | onExit_Fe | Fe biomass tracker | 3D RHO-variable |
| idTSvar(i3Stat131) | advdiff_NO3 | NO3 rate of change due to advection and diffusion | 3D RHO-variable |
| idTSvar(i3Stat132) | advdiff_NH4 | NH4 rate of change due to advection and diffusion | 3D RHO-variable |
| idTSvar(i3Stat133) | advdiff_PhS | PhS rate of change due to advection and diffusion | 3D RHO-variable |
| idTSvar(i3Stat134) | advdiff_PhL | PhL rate of change due to advection and diffusion | 3D RHO-variable |
| idTSvar(i3Stat135) | advdiff_MZL | MZL rate of change due to advection and diffusion | 3D RHO-variable |
| idTSvar(i3Stat136) | advdiff_Cop | Cop rate of change due to advection and diffusion | 3D RHO-variable |
| idTSvar(i3Stat137) | advdiff_NCaS | NCaS rate of change due to advection and diffusion | 3D RHO-variable |
| idTSvar(i3Stat138) | advdiff_EupS | EupS rate of change due to advection and diffusion | 3D RHO-variable |
| idTSvar(i3Stat139) | advdiff_NCaO | NCaO rate of change due to advection and diffusion | 3D RHO-variable |
| idTSvar(i3Stat140) | advdiff_EupO | EupO rate of change due to advection and diffusion | 3D RHO-variable |
| idTSvar(i3Stat141) | advdiff_Det | Det rate of change due to advection and diffusion | 3D RHO-variable |
| idTSvar(i3Stat142) | advdiff_DetF | DetF rate of change due to advection and diffusion | 3D RHO-variable |
| idTSvar(i3Stat143) | advdiff_Jel | Jel rate of change due to advection and diffusion | 3D RHO-variable |
| idTSvar(i3Stat144) | advdiff_Fe | Fe rate of change due to advection and diffusion | 3D RHO-variable |
| idTSvar(i3Stat145) | feastpred_Cop | Cop cumulative daily predation loss to fish | 3D RHO-variable |
| idTSvar(i3Stat146) | feastpred_NCaS | NCaS cumulative daily predation loss to fish | 3D RHO-variable |
| idTSvar(i3Stat147) | feastpred_NCaO | NCaO cumulative daily predation loss to fish | 3D RHO-variable |
| idTSvar(i3Stat148) | feastpred_EupS | EupS cumulative daily predation loss to fish | 3D RHO-variable |





| `idTSvar(i3Stat149)` | feastpred_EupO | EupO cumulative daily predation loss to fish | 3D RHO-variable |
| --- | --- | --- | --- |

*Author contributions.* K. Kearney wrote text, prepared figures, and ran and analyzed simulations for this paper; Hermann, Cheng, Ortiz, and Aydin provided edits and review during manuscript preparation. A. Hermann and W. Cheng are responsible for initial development and analysis of the physical model (Bering10K) used in these simulations. A. Hermann, I. Ortiz, and K. Aydin secured funding to support development of the Bering10K and BESTNPZ models.

*Acknowledgements.* This publication is funded by the Joint Institute for the Study of the Atmosphere and Ocean (JISAO) under NOAA Cooperative Agreement NA15OAR4320063, Contribution No. 2019-1014.





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
