# Peer review of "A coupled pelagic-benthic-sympagic biogeochemical model for the Bering Sea: documentation and validation of the BESTNPZ model (v2019.08.23) within a high-resolution regional ocean model"

_Geoscientific Model Development, 2019_

## Referee Comment (RC1) · Anonymous Referee #1 · 12 Nov 2019

This manuscript presents a thorough description and validation of a coupled physical-biogeochemical model for the Bering Sea. The physical model is the Bering10K and the biogeochemical counterpart is the BESTNPZ. The latter aims at characterizing the complex ecosystem of the Bering Sea by representing variables and processes not only in the water column, but also in the benthos and the sympagic sea ice environment. Both models have been used in previous publications, although both have been modified since the last published papers. The manuscript explains in detail the changes

(and the reason behind these changes) of the Bering10K and BESTNPZ since the latest publications and presents a thorough validation of several aspects of the coupled model. Overall, the authors conclude that the model skill is quite good in terms of the physics (e.g., patterns of water movement, mixing, stratification, water masses distribution), but there is still room for improvement in the ecosystem module (e.g. limited ability to replicate large scale patterns in nutrient cycling, primary production and zooplankton community composition).

General comment:

I found the manuscript well written and thoughtfully organized. Model performance is well assessed with a suit of metrics and figures; furthermore, technical details and model equations are well documented. I think that the clear identification of previous issues in the model and the explanation of their fixes (along with the honest description of the things that are still suboptimal, unnecessary and/or still not working well) will be very useful for fellow modellers encountering similar issues. Below, I make a list of some minor comments, which are mostly intended for clarification. Therefore, I recommend this paper for publication in Geoscientific Model Development.

Comments:

P5.L30: It'd be useful to state which publications had this nudging, since it is a major issue.

P7.L11: why did the approach change from adjusting CORE to adjusting CFSR? It'd be interesting to know.

P8.L19: Usually, modellers sub-sample the model to compare against observations. However, in this case, I understand that you have to interpolate the observations to look at the ice edge location. It would be useful to know how much your results depend on the interpolation method chosen and also how much interpolation is needed (ie, are there just a few data points missing here and there, or are there times when most of

the domain is being interpolated from a few points with satellite data?).

P9.L21: It would be useful to have the three regions shown in a map (a new figure or even in another panel in figs 1 or 2)

P10.L6: Given that you already calculated the Simposon parameter (SI), why not check the location of the front by using this parameter? Some authors have used a critical value of SI to determine the position of the fronts (eg, Bianchi et al 2005), while others have looked at the region of largest change in SI (eg, Wang et al 2004).

References: * Bianchi, A. A., Bianucci, L., Piola, A. R., Pino, D. R., Schloss, I., Poisson, A., and Balestrini, C. F. (2005), Vertical stratification and air-sea CO2 fluxes in the Patagonian shelf, J. Geophys. Res., 110, C07003.

* Wong, L. A., J. C. Chen, and L. X. Dong. (2004), A model of the plume front of the Pearl River Estuary, China and adjacent coastal waters in the winter dry season. Continental Shelf Research 24.16: 1779-1795.

P11.L33: Is there a reason or a reference for the choice of attenuation length scale of 45 m?

P12.L1: "our model-derived estimate of satellite-visible chlorophyll is a rough one": I'd suggest to mention explicitly that, while the model is vertically integrated, the satellite can only measure the surface.

P12.L17: I believe there is a typo in the biomass unit (it should be 10 gC m$^{-3}$ rather than $^{-2}$). Also, is there a reference for this value?

P14.Fig4: I'd suggest to add the location of the observations as small dots in the top panels, so we can se where interpolation is taking place.

P15.Fig6caption: "on the eastern shelf": Should it say "in the model domain" instead?

P16.L5: I'd suggest to mention that "structural fronts" are also known as tidal fronts

P17.1stParagraph: this text needs to point/refer to a figure. Is it Figure 10?

P18.Fig8caption: I'd suggest to mention explicitly here that these lines used the 0.5deg approach. Also, I'm curious: how would they look if you were using a critical SI instead?

P19.L2-4: when comparing the model to the observations at the surface, it would be fair to mention that there are no observations in the top ∼10 m (but we rely on the interpolation).

P25.L20: it feels to me that a word is missing in this sentence to make it clearer, maybe "... capable of being differentiated between EACH OTHER with this type of biomass box model."

P28.FigA1: This is a great way of showing a complex diagram. However, some colours are hard to differentiate (e.g. tan vs gold).

P28.FigA1caption: I'd suggest to mention the circles explicitly, eg: "CIRCLES SHOW STATE VARIABLES (gold = nutrient, green = producer, blue = consumer, brown = detritus). Edges (lines) represent fluxes between state variables and curve clockwise from source node to sink node; edges colors indicate... "

P34.Table6: Usually, alpha values for diatoms are larger than for small phytoplankton. Since you have alphaPhS > alphaPhL, I'm wondering if it is a typo or if there is a reason for this choice of values.

P46.L21: Does the 1% of Ph/Det lost to denitrification lead to a direct flux of NO3 into or NH4 out of the sediments? Since such fluxes are not shown in the equations, it seems that this 1% just adds to the 20% lost out of the DetBen pool. It would be nice to have a clearer explanation about this choice.

P50.L27: typo: "for" is written twice

P55: "MATLABCOMPILE" row: I'd suggest to replace "my" in the third column by "K.Kearney's" or any other way that is appropriate

P58.L2: I'd also suggest to rewrite to avoid "my" and "I"

P58.TableC4: given that all types are "RHO-variable", I'd suggest to mention that in the caption (along with an explanation of what "RHO-variable" means" and leave the column showing only 2D or 3D.

---

## Referee Comment (RC2) · Neil Banas (Referee) · 19 Nov 2019

**GENERAL COMMENTS**

This paper documents and evaluates a biophysical model of the Bering Sea, including not only the present state of the model code but incremental differences among the versions used in published studies. The stated purpose of the paper is to "reveals the model's strengths and weaknesses in reproducing historical patterns" (p.2), motivated

by the fact that "at least a dozen ongoing projects" (p.2) are relying on this model; the assessment also provides a "baseline to which further model improvements can be compared." Model performance is quite variable, but the analysis and discussion are solid and presented in a pleasingly direct style. I think this paper will serve as useful documentation, and an important reference point as those dozen projects work out how best to use the model in an applied context.

There are a few places where the directness regarding the model's limitations lapses, and some places where I think the comparison with phytoplankton and zooplankton observations could be, and deserve to be, made more complete–but overall I think the paper well deserves publication after revisions.

SPECIFIC COMMENTS

(1) Description of errors

The paper's final statement "However, we caution that the use of the biological state variable output should be limited until the model is better able to capture observed characteristics of the Bering Sea phytoplankton and zooplankton communities" is fair and honest. Some equivalent warning belongs in the _abstract_ as well: "ability. . .remains limited" is euphemistic in a way that most of this clear-eyed paper is not. (However, it would be fair and constructive to state the model limitations in positive form, as a recommendation for how best to make use of the model as it stands: e.g., "near-term application should focus on the use of physical model outputs rather than biological model outputs.")

(2) Phytoplankton validation

p.17 l.2-7: I think this account of the bloom phenology, along with the phrase "nuances in spatiotemporal variability" in the next paragraph, underplay the complexity and potential importance of these patterns. Ecosystem effects of variation in bloom timing have been at the centre of conceptual pictures of the Eastern Bering Sea ever

since Hunt et al. 2002. It would be straightforward and more helpful to the reader to summarise the patterns observed in both satellite data (Brown and Arrigo 2013) and mooring records (Sigler et al. 2014). Most notably, at M2, Sigler et al. show that the bloom date varied by about 70 days, April-June, between 1995-2011. It is fair to separate the mean seasonal cycle from interannual variability to discuss separately, but this history of observational and theoretical attention shouldn't be obscured.

Fig. 10, which shows major differences between modelled and satellite chlorophyll month by month, is well-described in the text. Likewise, the text gives a good account of both biases and agreements between modelled and moored chl at M2 (Fig. 11). I suppose the latter comparison is the support for the statement in the Abstract that the model is "able to capture the mean seasonal cycle of primary production observed on the data-rich eastern middle shelf." This is is not completely unfair, but it is a stretch. I think a more precise statement in terms of timing/magnitude/composition would be better.

It is also worth considering to what extent data limitations and not model limitations are responsible for blurring this picture. How does the mean of the annual-max mooring chl compare with the annual max of the composited mooring chl? Because of the huge variance in bloom timing, they might be quite different. Likewise, it would be interesting to superimpose a satellite time series from M2 on the model and mooring records in Fig 11c: if the observational records disagree on bloom timing, then there is an inherent level of fuzziness that one would expect from a data-model comparison.

(3) Zooplankton biomass

I think there are missed opportunities here for comparsion of the model with zooplankton data. The summary of typical biomass numbers by functional group in Sec 3.3 is very helpful, but I notice it doesn't contain any references to the observational work during BEST-BSIERP itself, which might provide more definite points of comparison. For example, Campbell et al. 2016 (Deep-Sea Research 134:157–172) in Table 6 give

mesozooplankton biomass by region/season/year, as well as comparisons with integrated phytoplankton biomass and estimated grazing rates in relation to primary productivity—such that the relation of zooplankton to phytoplankton could be assessed in either absolute, relative, or functional terms. Stoecker et al. 2013 (Deep Sea Res 109:134-44) gives similar numbers for microzooplankton.

As best I can tell from a quick comparison, the overall biomass of microzooplankton in the model is on the right order, but the large-zooplankton biomass in reality is orders of magnitude larger than the micro-, not comparable or smaller as in the model. The authors give 10 g C/mˆ2 as a typical large-zooplankton biomass in Sec 3.3, and from Campbell et al. 2016 Table 6, I would have said 1-2 g C/mˆ2–this difference is not the important one–whereas the model seems to have large zooplankton on the order of 0.01 g C/mˆ2, unless I'm misunderstanding Fig. 12. A biomass bias on that scale raises questions about the overall role the zooplankton play in nutrient and phytobiomass budgets, in the model and in reality. A careful look at the grazing rate parameters might be a good place to start: again, with the BEST/BSIERP observational papers mentioned above, along with others by Stoecker et al. and Sherr et al., as concrete guidance. (I went through this exercise in Banas et al. 2016 (J Geophys Res, 10.1002/2015JC011449) and concluded that microzooplankton max specific grazing was an order of magnitude higher than the value reported in Table A7.) General reviews like Hansen et al. 1997 (Limnol Oceaongr 42:687:704) and Kiorboe and Hirst 2014 (American Naturalist 183, 10.1086/675241) might be a simpler way to get at the same issues.

(4) Design strategies and parameter comparisons

Sec 5 includes a discerning discussion of the design limitations and parameterisation issues that could be degrading the biogeochemical performance of the model. I think the authors have chosen a good set of issues to highlight, such as the approach to ice algae, the photophysiology (p. 24), and the over-resolution of the large zooplankton boxes, especially given the inherent limitations of a stock-flux framework compared

with life-stage-resolving models (p. 25). But what are the next steps in model development? Are there design/parameterisation strategies the authors are planning to take, or already taking, or feel are worth mentioning more generally as good possibilities in these sorts of situations?

In this context I think it's worth pointing out that the authors haven't cited any of the other Bering Sea plankton modelling efforts in the literature, but could: not to turn this into a model intercomparison project and certainly not a competition, but to highlight other design, parameterisation, and validation strategies that could have benefits for the future of the BESTNPZ effort. Maybe the Bering Sea modelling literature is too narrow, and it's subpolar/polar plankton modelling in general that deserves some mining. In the Bering Sea, I can think of two recent and one older NPZ models with independent lineages and pretty good agreement with plankton observations: (1) Jin et al., Geophys Res Lett 34:L06612, 2007; (2) Zhang et al., Deep Sea Res 118:122-135, 2018 (emphasis on Chukchi blooms but unpublished analysis shows it does quite well against against satellite chl in the Bering Sea too); and (3) my own model, Banas et al. 2016 (reference above), further developed by Sloughter et al., J Mar Sys 191:64-75, 2019 through photophysiology process data from BEST/BSIERP. Likewise, there are at least two published Calanus life-history models for the Bering Sea with independent lineages: (4) Coyle and Gibson, J Plankt Res 39:257-270, 2017, and (5) my own, Banas et al., Front Mar Res, 10.3389/fmars.2016.00225, 2016.

I don't mean to push the authors toward intercomparisons or competitions, but the processes we have all concluded we need to attend to are very, very similar. Zhang et al focus on ice algae and under-ice growth conditions, Sloughter et al. focus on photoparameters and bloom timing, and the Calanus IBMs in the Bering Sea focus on (and perhaps disagree on) the constraints on over-winter survival–and these are exactly the issues that the authors of this study highlight as crucial to the performance of BESTNPZ. So I would be surprised if there was nothing to learn or comment on from digging a bit into the similarities and differences among these models and their

parameter values.

MINOR COMMENTS

p.3 "of the larger Northeast Pacific (NEP5) domain" -> "of a larger-domain ROMS model of the Northeast Pacific (NEP5)"

p.16, l.14-17: The tidally averaged currents may be much smaller than tidal velocities, but still crucially important to lateral nutrient supply and the distribution of biomass. A more quantitative comparison with the transport patterns synthesized by Stabeno et al. 2016 (e.g. Fig 12 in that study vs fig 9 in this one: Deep-Sea Research 134:13–29) would be helpful.

Table A7: could the feeding preferences be placed in their own table as a matrix? They would be much easier to read that way.

---

## Author Comment (AC1) · 24 Dec 2019

**Author's Comment: gmd-2019-239**

We thank both reviewers for their constructive comments on this manuscript. Here, we address each comment. Reviewer comments are italicized, with our responses below. Where snippets of changed text appear in our replies, we have underlined new additions. A marked-up version of the manuscript follows our comments; all line and page numbers in our response refer to that marked-up version.

**Referee 1**

*General comment:*

*I found the manuscript well written and thoughtfully organized. Model performance is well assessed with a suit of metrics and figures; furthermore, technical details and model equations are well documented. I think that the clear identification of previous issues in the model and the explanation of their fixes (along with the honest description of the things that are still suboptimal, unnecessary and/or still not working well) will be very useful for fellow modellers encountering similar issues. Below, I make a list of some minor comments, which are mostly intended for clarification. Therefore, I recommend this paper for publication in Geoscientific Model Development.*

*Comments:*

*P5.L30: It'd be useful to state which publications had this nudging, since it is a major issue.*

We added the following text (Page 5, line 30-31) to clarify that all previous publications of the Bering10K + BESTNPZ model included this nudging:

"However, when moved to the three-dimensional Bering10K domain (as was done in all previous publications using BESTNPZ, including Hermann et al. (2013) and Hermann et al. (2016)), this nudging becomes inappropriate…"

*P7.L11: why did the approach change from adjusting CORE to adjusting CFSR? It'd be interesting to know.*

This decision was primarily a practical one. The adjustments to radiation values were done based on simple pattern-matching, with the goal of avoiding any noticeable temperature artifacts at the dataset junction, rather than being based on a specific underlying mechanism. Therefore, the choice of which dataset to adjust is somewhat arbitrary.

In the earlier simulations, the dataset junction was placed between 2004 (the last COREv2 year available) and 2005; because there were far more CORE years than CFSR years, the radiation values from the latter were modified. In subsequent years, hindcast simulations were extended further forward as new output became available from CFSR then the newer CFS operational analysis dataset. Also, 2004 proved to be an inconvenient location for the dataset junction, since the years of 2004-2005 marked a shift from a warm period to a cold period in the Eastern Bering Sea shelf, and the junction was moved to 1994/5. Rather than continuing to apply

adjustments to each new year's data as it became available, we opted to reverse the adjustments, leaving CFSR/CFSv2 as is and modifying CORE radiation values. Additionally, this allows for consistency with other ongoing projects that rely on the unmodified CFSR/CFSv2 reanalysis, operational forecast, and reforecast datasets.

We are currently conducting more detailed comparisons of the Bering10K model's performance using both datasets, independently and unmodified, during the period where they overlap (1979-2004). This should provide more mechanistically sound justifications for any adjustments to either dataset in future simulations. However, for the purposes of the relevant metrics within this paper, this reversal in input radiation adjustments plays a much smaller role than other changes to the model framework (e.g. number of layers, changes to light attenuation).

*P8.L19: Usually, modellers sub-sample the model to compare against observations. However, in this case, I understand that you have to interpolate the observations to look at the ice edge location. It would be useful to know how much your results depend on the interpolation method chosen and also how much interpolation is needed (ie, are there just a few data points missing here and there, or are there times when most of the domain is being interpolated from a few points with satellite data?).*

The satellite sea ice data is provided on a polar stereographic grid with nominal grid resolution ranging from about 6.25—25 km. There are no spatial gaps in this satellite dataset; the interpolation was purely a practical step to keep sea ice edge calculations between the model and satellite data consistent. Because the data was at comparable resolution to the 10-km model, particularly near the sea ice edge, we found that the method of interpolation had a negligible effect on our ice edge calculations; nearest neighbor was chosen as the quickest option.

We have altered the text as follows (Page 8, lines 24-25) to clarify this point:

"For comparison with model output, satellite-derived fraction ice cover was interpolated from its native 6.25 km to 20 km-resolution polar stereographic grid to the Bering10K model grid via a nearest neighbor method."

*P9.L21: It would be useful to have the three regions shown in a map (a new figure or even in another panel in figs 1 or 2)*

We have added contour lines to the Figure 1 map, along with an approximation for where these isobaths correspond to the 3 mixing domains.

*P10.L6: Given that you already calculated the Simposon parameter (SI), why not check the location of the front by using this parameter? Some authors have used a critical value of SI to determine the position of the fronts (eg, Bianchi et al 2005), while others have looked at the region of largest change in SI (eg, Wang et al 2004).*

We experimented with a number of different methods for identifying the fronts separating the inner domain from middle domain and the middle domain from the outer domain. The maximum-gradient contour method had the benefit of allowing us to only choose a single threshold value that could identify all fronts, rather than choosing specific values for each front

(as would be necessary to identify all with a straight SI contour). Also, the stratification index calculation did not scale well in deeper water, due to the coarser resolution in surface waters of the basin. We found the temperature gradient required fewer arbitrary cutoffs and masks in order to be applied to the entire domain. When compared to the SI contours, both methods agreed fairly closely on the location of the inner/middle front. Note that in the figure below, the SI = 100 contour is more or less identifying the edge of the depth mask applied to the SI value calculation, rather than a particular stratification feature.

[Figure]

*P11.L33: Is there a reason or a reference for the choice of attenuation length scale of 45 m?*

The length scale was chosen based on mean global values of K490 (i.e. attenuation coefficient at 490 nm) as estimated from satellite (and following advice on the Ocean Color Forum at https://oceancolor.gsfc.nasa.gov/forum/oceancolor/ regarding estimation of optical depth.)

*P12.L1: "our model-derived estimate of satellite-visible chlorophyll is a rough one": I'd suggest to mention explicitly that, while the model is vertically integrated, the satellite can only measure the surface.*

In this particular case, we compare the satellite data to *optically-weighted* integrated biomass, not simple integrated biomass. This calculation sees the same portion of the modeled water column that the satellite would.

*P12.L17: I believe there is a typo in the biomass unit (it should be 10 gC m^-3 rather than ^-2). Also, is there a reference for this value?*

The units are correct; larger zooplankton species are typically sampled by either net or acoustics, and therefore reported as depth-integrated concentrations. We have added a few citations (Campbell et al., 2016; Hunt et al., 2016) to accompany this estimate, as well as adjusting the number to reflect the range of uncertainty covered by the numbers in these

citations (Page 12, lines 21-22).  The updated range does not alter our paper's conclusions (our mesozooplankton numbers are too high, regardless).

*P14.Fig4: I'd suggest to add the location of the observations as small dots in the top panels, so we can se where interpolation is taking place.*

We have added markers as suggested, and updated the figure caption accordingly.

*P15.Fig6caption: "on the eastern shelf": Should it say "in the model domain" instead?*

The text has been corrected accordingly.

P16.L5: I'd suggest to mention that "structural fronts" are also known as tidal fronts

The text has been revised (Page 15, line 5) to read, "the model does reproduce the structural front, also known as a tidal front, expected between the unstratified inner domain and thermally-stratified middle domain during the summer months."

*P17.1stParagraph: this text needs to point/refer to a figure. Is it Figure 10?*

A reference to Figure 10 was added after the first sentence of this paragraph (Page 16, line 11).

*P18.Fig8caption: I'd suggest to mention explicitly here that these lines used the 0.5deg approach. Also, I'm curious: how would they look if you were using a critical SI instead?*

We altered the caption to read, "Location of structural fronts,  defined as the 0.5 deg C/m contour line of maximum vertical temperature gradient ."  See figure above for a comparison of critical SI contours versus this metric.

*P19.L2-4: when comparing the model to the observations at the surface, it would be fair to mention that there are no observations in the top ~10 m (but we rely on the interpolation).*

We altered the next here (Page 18, line 10) to read, "the bloom begins with a large, diatom-dominated bloom starting in the near-surface waters and then migrating deeper as surface nutrients are depleted."

*P25.L20: it feels to me that a word is missing in this sentence to make it clearer, maybe ". . . capable of being differentiated between EACH OTHER with this type of biomass box model."*

The text was altered (Page 25 line 16) to read, "In contrast to the under-resolved detrital pools, the mesozooplankton groups included in the BESTNPZ model appear to be over-resolved in terms of functional differences capable of being differentiated  from each other with this type of biomass box model."

*P28.FigA1: This is a great way of showing a complex diagram. However, some colours are hard to differentiate (e.g. tan vs gold).*

Given the 11 different flux types, it is difficult to find a set of colors that are clearly distinct from one another. This set was the best found after quite a lot of color experimentation. We did attempt to use similar colors only for fluxes that could not be mistaken for each other based on pathways (e.g. all gold lines target NH4 or IceNH4, while all tan ones target detrital nodes).

*P28.FigA1caption: I'd suggest to mention the circles explicitly, eg: "CIRCLES SHOW STATE VARIABLES (gold = nutrient, green = producer, blue = consumer, brown = detritus). Edges (lines) represent fluxes between state variables and curve clockwise from source node to sink node; edges colors indicate. . . "*

Text changed as suggested.

*P34.Table6: Usually, alpha values for diatoms are larger than for small phytoplankton. Since you have alphaPhS > alphaPhL, I'm wondering if it is a typo or if there is a reason for this choice of values.*

These values originate from the one-dimensional version of the BESTNPZ model as documented by Gibson & Spitz, 2011, which in turn inherited the values from the Gulf of Alaska GOANPZ model (Coyle et al., 2012), who based their parameters on P-E curves fit to data collected in 2003 along the GAK sampling line in the Gulf of Alaska (Strom et al., 2010). Strom et al., 2010 attributed the higher alpha values in the small size class to higher C:chl ratios within this group. Reexamination of these values and their appropriateness for the Bering Sea, as well as the equations governing light harvesting and its plasticity or lack thereof, is a high priority in future model development.

Coyle KO, Cheng W, Hinckley SL, Lessard EJ, Whitledge T, Hermann AJ, Hedstrom K (2012) Model and field observations of effects of circulation on the timing and magnitude of nitrate utilization and production on the northern Gulf of Alaska shelf. Prog Oceanogr 103:16–41

Strom SL, Macri EL, Fredrickson KA (2010) Light limitation of summer primary production in the coastal Gulf of Alaska: Physiological and environmental causes. Mar Ecol Prog Ser 402:45–57

*P46.L21: Does the 1% of Ph/Det lost to denitrification lead to a direct flux of NO3 into or NH4 out of the sediments? Since such fluxes are not shown in the equations, it seems that this 1% just adds to the 20% lost out of the DetBen pool. It would be nice to have a clearer explanation about this choice.*

The reviewer is correct that this flux is denitrification in name only, and is treated as a loss from the system. We added the following text (Page 48, lines 17-19) to clarify: "Note that the ``denitrification'' flux is not tracked explicitly, but simply subtracted from the flux reaching benthic detritus; the biomass associated with both burial and denitrification is lost from the system."

*P50.L27: typo: "for" is written twice*

Fixed typo (Page 52, line 27)

*P55: "MATLABCOMPILE" row: I'd suggest to replace "my" in the third column by "K.Kearney's" or any other way that is appropriate*

Text changed as suggested (Page 57, though change not marked by latexdiff markup due to use of external .tex files for tables).

*P58.L2: I'd also suggest to rewrite to avoid "my" and "I"*

We rephrased this paragraph to use more passive phrasing (page 59, line 14 to Page 60, line 2): "More recent versions of the code no longer include 2D diagnostic variables. However, the code structure is still in place for this if it becomes necessary in the future."

*P58.TableC4: given that all types are "RHO-variable", I'd suggest to mention that in the caption (along with an explanation of what "RHO-variable" means" and leave the column showing only 2D or 3D.*

We added text to the caption (Page 60, though change not marked by latexdiff markup due to use of external .tex files for tables) to explain that RHO-variable refers to a variable located in the center of each grid cell (as opposed to on an edge or at the corner, as is also possible in the ROMS framework).

**Referee 2**

*(1) Description of errors*
*The paper's final statement "However, we caution that the use of the biological state variable output should be limited until the model is better able to capture observed char- acterics of the Bering Sea phytoplankton and zooplankton communities" is fair and honest. Some equivalent warning belongs in the _abstract_ as well: "ability. . .remains limited" is euphemistic in a way that most of this clear-eyed paper is not. (However, it would be fair and constructive to state the model limitations in positive form, as a recommendation for how best to make use of the model as it stands: e.g., "near-term application should focus on the use of physical model outputs rather than biological model outputs.")*

We altered the final line of the abstract as suggested to more clearly highlight areas of skill versus areas of continued development in the model (Page 1, lines 13-15). A few additional changes were made to the abstract (Page 1, line 10) to clarify some ambiguous phrasing.

*(2) Phytoplankton validation*
*p.17 l.2-7: I think this account of the bloom phenology, along with the phrase "nuances in spatiotemporal variability" in the next paragraph, underplay the complexity and potential importance of these patterns. Ecosystem effects of variation in bloom timing have been at the centre of conceptual pictures of the Eastern Bering Sea ever since Hunt et al. 2002. It would be straightforward and more helpful to the reader to summarise the patterns observed in both satellite data (Brown and Arrigo 2013) and mooring records (Sigler et al. 2014). Most notably, at M2, Sigler et al. show that the bloom date varied by about 70 days, April-June, between 1995-2011. It is fair to separate the mean seasonal cycle from interannual variability to discuss separately, but this history of observational and theoretical attention shouldn't be obscured.*

*Fig. 10, which shows major differences between modelled and satellite chlorophyll month by month, is well-described in the text. Likewise, the text gives a good account of both biases and agreements between modelled and moored chl at M2 (Fig. 11). I suppose the latter comparison is the support for the statement in the Abstract that the model is "able to capture the mean seasonal cycle of primary production observed on the data-rich eastern middle shelf." This is is not completely unfair, but it is a stretch. I think a more precise statement in terms of timing/magnitude/composition would be better.*

We altered the text in a few locations to better explain the central role of interannual variability in the predominant theories of Bering Sea ecosystem function.  First, in the methods section (Page 11, lines 27-29) we highlighted the central role that interannual variability in the spring bloom has played in most theories of ecosystem function:

"This ice edge bloom occurs during years where ice lingers later over the shelf, protecting the underlying water from wind mixing and setting up stronger stratification; earlier-melting ice leads to more wind mixing and a later spring bloom.  Variations in spring bloom timing, its correlation or lack thereof with ice melt date, and the impact of this timing on community composition and energy transfer to higher trophic levels form the backbone of most prevailing theories of ecosystem function in the southeastern Bering Sea (Hunt et al., 2010; Sigler et al., 2016)

Within the results text (Page 17, lines 3-5), we added the following sentence to provide a bit more detail regarding the bloom date mismatches in the M2 data versus our model: "Measurements at the M2 mooring location suggest that peak spring bloom date varies widely, from mid-April to early June (Sigler et al., 2014); in the model, peak bloom timing is constricted to a much narrower window from early to late May."

Finally, we altered the text in the Discussion section (Page 23, line 31 to Page 24, line 2) to more bluntly state the importance that phenological patterns play in the current understanding of the EBS ecosystem:

"In particular, the timing of the spring bloom, and its correlation or lack thereof with ice retreat timing, form the basis for many theories of energy transfer within the EBS ecosystem.  Given the key role that phenological variability plays in the predominant theories of energy transfer, shortcomings in the model's ability to capture the processes leading to such variability raise concerns about its potential ability to predict either current or future changes in primary and secondary production."

*It is also worth considering to what extent data limitations and not model limitations are responsible for blurring this picture. How does the mean of the annual-max mooring chl compare with the annual max of the composited mooring chl? Because of the huge variance in bloom timing, they might be quite different. Likewise, it would be interesting to superimpose a satellite time series from M2 on the model and mooring records in Fig 11c: if the observational records disagree on bloom timing, then there is an inherent level of fuzziness that one would expect from a data-model comparison.*

We did intend for Fig. 11, panel c to indicate the range of values measured by both satellite (blue lines) and in-situ mooring measurements (green lines). The mooring measurements offer the benefit of being able to measure under-ice and subsurface values, but are also subject to much higher variability resulting from measuring patchy blooms at a single point. Satellites, while less subject to small-scale noise, are unable to see through ice or cloud cover, limiting their ability to quantify early-bloom dynamics. We found this combination of noisiness in the former dataset and missing-data in the latter made it difficult to quantify agreement (or lack thereof) between the two datasets in terms of bloom timing; therefore, we instead focused on the climatological averages for our comparison.

*(3) Zooplankton biomass*
*I think there are missed opportunities here for comparsion of the model with zooplankton data. The summary of typical biomass numbers by functional group in Sec 3.3 is very helpful, but I notice it doesn't contain any references to the observational work during BEST-BSIERP itself, which might provide more definite points of comparison. For example, Campbell et al. 2016 (Deep-Sea Research 134:157–172) in Table 6 give mesozooplankton biomass by region/season/year, as well as comparisons with integrated phytoplankton biomass and estimated grazing rates in relation to primary productivity such that the relation of zooplankton to phytoplankton could be assessed in either absolute, relative, or functional terms. Stoecker et al. 2013 (Deep Sea Res 109:134-44) gives similar numbers for microzooplankton.*

*As best I can tell from a quick comparison, the overall biomass of microzooplankton in the model is on the right order, but the large-zooplankton biomass in reality is orders of magnitude larger than the micro-, not comparable or smaller as in the model. The authors give 10 g C/m^2 as a typical large-zooplankton biomass in Sec 3.3, and from Campbell et al. 2016 Table 6, I would have said 1-2 g C/m^2–this difference is not the important one–whereas the model seems to have large zooplankton on the order of 0.01 g C/m^2, unless I'm misunderstanding Fig. 12. A biomass bias on that scale raises questions about the overall role the zooplankton play in nutrient and phytobiomass budgets, in the model and in reality. A careful look at the grazing rate parameters might be a good place to start: again, with the BEST/BSIERP observational papers mentioned above, along with others by Stoecker et al. and Sherr et al., as con- crete guidance. (I went through this exercise in Banas et al. 2016 (J Geophys Res, 10.1002/2015JC011449) and concluded that microzooplankton max specific grazing was an order of magnitude higher than the value reported in Table A7.) General reviews like Hansen et al. 1997 (Limnol Oceaongr 42:687:704) and Kiorboe and Hirst 2014 (American Naturalist 183, 10.1086/675241) might be a simpler way to get at the same issues.*

First, we apologize for a crucial typo in Fig. 12; the values plotted were in mmol N m^-2, rather than in mg C m^-2 as was indicated by the axis label. We have corrected this in the updated version of Fig. 12. The factor of 79.365 between the two sets of units accounts for the apparent disagreement between our text and this figure. Our biomass values are indeed quite high, compared to the 1-10 mg C m^-2 field estimates.

The figure below (data equivalent to Fig 12, M2 subpanels) perhaps shows the comparative biomass of each phytoplankton and zooplankton group better (using both the N-based and C-based units on the left and right axes, respectively). These are the same values shown in Fig 12, but on a log scale and without the cumulative summing across P and Z. We have updated Fig 12

to use line plots on a log scale, and with the axes labeled in both sets of units; this better emphasizes the similar patterns seen across all zooplankton groups while allowing one to see the approximate order of magnitude of each group's biomass. We also corrected the location labels to use the M2 and M3 labels for the stations coinciding with mooring locations, rather than the BS-2 and BS-3 labels that were used in the earlier version (the mooring datasets are referred to by both monikers in different data publications, but the former are more common and are used in this rest of the paper).

[Figure]

Our decision to avoid an in-depth quantitative comparison between the modeled results and EBS zooplankton sampling data, including that from the BEST/BSIERP program, was motivated by a combination of practical limitations and expected return on investment from such an evaluation. Of practical note, we were concerned about the already extremely lengthy state of this manuscript. Given the range of methodologies and data issues across survey methods, we decided that calibration of data and comparison (including documentation of calibration methods for the purposes of comparison) would be beyond the scope of the current manuscript. A quantitative comparison between this model and the EBS sampling data (including BEST/BSIERP data) is ongoing, under the lead of David Kimmel at AFSC, but we decided that such a comparison was beyond the scope of documentation and validation. In addition, given the already-identified issues regarding primary production, mesozooplankton biomass order-of-magnitude, and mesozooplankton community dynamics demonstrated by the model, we decided an in-depth quantitative assessment would add little to the overall conclusion that this model's skill at reproducing patterns in secondary production is limited.

*(4) Design strategies and parameter comparisons*
*Sec 5 includes a discerning discussion of the design limitations and parameterisation issues that could be degrading the biogeochemical performance of the model. I think the authors have chosen a good set of issues to highlight, such as the approach to ice algae, the photophysiology (p. 24), and the over-resolution of the large zooplankton boxes, especially given the inherent limitations of a stock-flux framework compared with life-stage-resolving models (p. 25). But what are the next steps in model development? Are there design/parameterisation strategies the authors are planning to take, or already taking, or feel are worth mentioning more generally as good possibilities in these sorts of situations?*

*In this context I think it's worth pointing out that the authors haven't cited any of the other Bering Sea plankton modelling efforts in the literature, but could: not to turn this into a model intercomparison project and certainly not a competition, but to highlight other design, parameterisation, and validation strategies that could have benefits for the future of the BESTNPZ effort. Maybe the Bering Sea modelling literature is too narrow, and it's subpolar/polar plankton modelling in general that deserves some mining. In the Bering Sea, I can think of two recent and one older NPZ models with independent lineages and pretty good agreement with plankton observations: (1) Jin et al., Geophys Res Lett 34:L06612, 2007; (2) Zhang et al., Deep Sea Res 118:122-135, 2018 (emphasis on Chukchi blooms but unpublished analysis shows it does quite well against against satellite chl in the Bering Sea too); and (3) my own model, Banas et al. 2016 (reference above), further developed by Sloughter et al., J Mar Sys 191:64- 75, 2019 through photophysiology process data from BEST/BSIERP. Likewise, there are at least two published Calanus life-history models for the Bering Sea with independent lineages: (4) Coyle and Gibson, J Plankt Res 39:257-270, 2017, and (5) my own, Banas et al., Front Mar Res, 10.3389/fmars.2016.00225, 2016.*

*I don't mean to push the authors toward intercomparisons or competitions, but the processes we have all concluded we need to attend to are very, very similar. Zhang et al focus on ice algae and under-ice growth conditions, Sloughter et al. focus on photoparameters and bloom timing, and the Calanus IBMs in the Bering Sea focus on (and perhaps disagree on) the constraints on over-winter survival–and these are exactly the issues that the authors of this study highlight as crucial to the performance of BESTNPZ. So I would be surprised if there was nothing to learn or comment on from digging a bit into the similarities and differences among these models and their parameter values.*

A new paragraph was added to the end of the Discussion section ()Page 25 line 31 to Page 26 line 16), detailing a few of the improvements that are being considered for the future of this model:

"Given the deficiencies identified in this evaluation, future work will comprehensively reevaluate each component of the existing model.  More accurate simulation of under-ice and near-ice phytoplankton blooms may be addressed by allowing seasonal plasticity in the parameters defining the photosynthesis-irradiance curve for each phytoplankton group; when used in a simple NPZD-style model, this type of equation has been shown to better capture the magnitude and timing of Bering Sea blooms than constant parameters (Sloughter et al., 2019). For sea ice algae, Tedesco and Vichi (2014) note that models using a fixed-thickness skeletal ice layer tend to underestimate production in first-year ice; they suggest that varying the width of the sea ice layer in which algae is found as a function of sea ice permeability can help overcome this issue with minimal additional model complexity required.  Issues related to excessive regenerated production on the eastern shelf may be addressed by more closely examining the detrital functional groups within the model, and the remineralization timescales associated with each; the use of a single remineralization timescale for all detrital groups is out of step with most modern biogeochemical models (e.g. Moore et al., 2002; Aumont and Bopp, 2006; Dunne et al., 2012) and allowing for parameters that reflect the varying lability of different detrital pools may better capture the nutrient dynamics both on and off the shelf.  Improving the EBS nutrient budget may also require a more complex representation of the benthic component of the ecosystem; the benthic module from a mature shelf model such as ERSEM (Butenschön et

al., 2016) may offer a blueprint for future development related to benthic functional groups. Finally, we intend to reconsider the number of functional groups used to represent the planktonic consumers within this ecosystem.  Banas et al. (2016) demonstrated that a much simpler 6-box model was capable of capturing spring bloom dynamics representative of the M2 mooring location.  However, Friedrichs et al. (2007) cautioned that though simple models are typically able to be tuned to better simulate the ecosystem dynamics of a single location, their portability is more limited than their more complex counterparts.  Given the rapidly-changing conditions in the Bering Sea, and the wide range of applications for which this model was designed (ranging from hindcast-based process studies to long-term climate-change forecasts), we must carefully consider the tradeoffs of parsimony versus complexity."

*MINOR COMMENTS*
*p.3 "of the larger Northeast Pacific (NEP5) domain" -> "of a larger-domain ROMS model of the Northeast Pacific (NEP5)"*

Text changed as suggested (Page 3, line 24).

*p.16, l.14-17: The tidally averaged currents may be much smaller than tidal velocities, but still crucially important to lateral nutrient supply and the distribution of biomass. A more quantitative comparison with the transport patterns synthesized by Stabeno et al. 2016 (e.g. Fig 12 in that study vs fig 9 in this one: Deep-Sea Research 134:13–29) would be helpful.*

We added the following paragraph (page 15 line 16 to Page 16, line 7), which provides a quantitative comparison of model flow compared to the Stabeno et al., 2016 drifter-based measurements:

"Water entering the Bering Sea from the Gulf of Alaska through Unimak Pass moves alongside and onto the eastern shelf and travels northward; it takes approximately 7-8 months to reach the northern shelf region (i.e. 60 N) along the 100-m isobath, in line with drifter-derived measurement of this flow (Stabeno et al., 2016). Further north, modeled velocities are slightly slower than seen in the observations, with water taking approximately 13-15 months to reach the Bering Strait from Unimak Pass in the model compared to 9-13 months in the observations; this may reflect a weak Anadyr Current in this region, or alternatively be the result of missing flow from off-shelf through submarine canyons that are not well-resolved by the modeled bathymetry.  Overall, flow within the modeled Bering Sea reproduces the important circulation patterns within this region."

*Table A7: could the feeding preferences be placed in their own table as a matrix? They would be much easier to read that way.*

An additional table (Table A8) was added rearranging the feeding preference parameters as a matrix.